# Dynamic blebbing and absence of organelle transfer during mouse oocyte formation

Eishi Aizawa [1,2 ✉], So Shimamoto[1], Eriko Kajikawa[3], Junko Hara [4], Takaya Abe [4], Hiroki Shibuya [3] & Tomoya S Kitajima [1 ✉]

## Abstract

**Oocyte formation in mammals is a tightly regulated process essential for female fertility, yet the underlying mechanisms remain poorly understood. In this study, we establish an ex vivo culture system that faithfully recapitulates in vivo development and enables long-term live imaging of mouse fetal ovaries. Using high resolution imaging, we capture the dynamic behaviors of germ cells during the development from oogonia to nascent oocytes. We identify pronounced blebbing activity during the mitosis-to-meiosis transition. This behavior is regulated by meiotic initiation signals, underscoring its potential developmental relevance, although its precise role remains unclear. A prevailing model suggests that oocyte formation involves organelle transfer from neighboring germ cells during cyst breakdown. However, through photoconversion-based tracking, we observe no detectable transfer of mitochondria or centrosomes, as organelles remain confined to individual cells. These findings point to alternative mechanisms for cytoplasmic enrichment in oocytes. Our study provides new insights into mammalian oocyte formation and establishes a powerful platform for analyzing germ cell dynamics in real time.**

**Subject Categories** Cell Cycle; Development; Methods & Resources

## Introduction

Germ cells are fundamental to the continuity of life, ensuring the transmission of genetic information across generations. Among germ cells, oocytes play a vital role, as they contribute genetic material while also providing cytoplasmic components and organelles essential for early embryonic development. Despite their biological importance, the mechanisms underlying the formation of oocytes remain largely elusive.

In mammals, germ cells originate as primordial germ cells (PGCs), which arise early in embryogenesis, around embryonic day (E) 7.25 (Saitou, 2021). These PGCs migrate to the developing gonads, where their fate is determined by local signaling cues. Around E11.5, bone morphogenetic protein (BMP) signaling in the gonadal environment directs these cells toward oogenic fate, prompting their differentiation into female-type germ cells known as oogonia (Miyauchi et al, 2017; Wu et al, 2016). Oogonia proliferate through mitotic divisions before entering meiosis, typically around E13.5–E14.5, and progressing to oocyte development (Spiller and Bowles, 2022).

A critical aspect of this developmental transition is the shift from mitotic proliferation to meiotic entry. This mitosis-to-meiosis transition is tightly regulated by extrinsic signals, notably the BMP and retinoic acid (RA) signaling (Bowles et al, 2006; Koubova et al, 2006; Miyauchi et al, 2017). BMP signaling confers meiotic competence to oogonia, while RA signaling induces the expression of *Stra8*, a transcription factor essential for initiating meiosis (Anderson et al, 2008). A recent study demonstrated that BMP signaling and STRA8 function cooperatively to ensure proper progression through the mitosis-to-meiosis transition (Cheung et al, 2025). In contrast to male germ cells, which enter a prolonged mitotic arrest, female germ cells proceed through a pre-meiotic S-phase and initiate meiosis during fetal development (Miles et al, 2010). However, the cellular dynamics accompanying the mitosis-to-meiosis transition remain poorly characterized, underscoring the need for further investigation.

Upon completing mitotic divisions, oogonia form germ cell cysts through incomplete cytokinesis, resulting in clusters of interconnected cells via intercellular bridges (Pepling and Spradling, 1998). A recent study showed that the balance between germ cell motility and the stability of intercellular bridges regulates the size of cysts (Levy et al, 2024). Within these cysts, a subset of germ cells undergoes meiotic progression, while others are eliminated during cyst breakdown, a process that separates interconnected germ cells into individual oocytes. Between E14.5 and postnatal day (PD) 5, approximately 80% of germ cells are lost through this process (Niu and Spradling, 2022). Following cyst breakdown, the surviving oocytes are individually enclosed by pregranulosa cells, forming primordial follicles beginning around PD1 (O'Connell and Pepling, 2021). These follicles then remain dormant for extended periods, over a year in mice and

[1]Laboratory for Chromosome Segregation, RIKEN Center for Biosystems Dynamics Research, Kobe, Japan. [2]Department of Molecular and Cellular Biology, Harvard University, Cambridge, MA, USA. [3]Laboratory for Gametogenesis, RIKEN Center for Biosystems Dynamics Research, Kobe, Japan. [4]Laboratory for Animal Resources and Genetic Engineering, RIKEN Center for Biosystems Dynamics Research, Kobe, Japan. ✉E-mail: eaizawa@fas.harvard.edu; tomoya.kitajima@riken.jp

approximately 50 years in humans, ensuring a sustained ovarian reserve critical for lifelong fertility (Zhang et al, 2014).

The mechanisms underlying oocyte formation have been investigated across various species. In insects and lower vertebrates, precursor germ cells undergo synchronous mitotic divisions, giving rise to germline cysts interconnected by intercellular bridges (Buning, 1994; Matova and Cooley, 2001). In *Drosophila*, germline cysts are formed where 16 cyst cells are connected by incomplete cell divisions (Spradling et al, 2022). The intercellular bridges between these cyst cells facilitate the transfer of organelle-enriched cytoplasm from surrounding cells to the dominant oocyte, ensuring its development into the mature egg (Cox and Spradling, 2003; Mahowald and Strassheim, 1970). In mammals, organelle and cytoplasm transfer within cysts is believed to follow a similar pattern but with more complexity (Ikami et al, 2023; Lei and Spradling, 2013, 2016; Niu and Spradling, 2022). During cyst breakdown, a subset of germ cells undergo selection, ultimately contributing to the establishment of a finite ovarian reserve. The interaction between cyst germ cells during cyst breakdown is thought to involve intercellular bridges, facilitating the transfer of organelles and cytoplasm between cyst cells. Using a cell labeling strategy in mice, Lei and Spradling traced the cyst development and reported that organelle-enriched cytoplasm is transferred from nursing germ cells to dominant germ cells presumably via plasma membrane gaps between cyst germ cells (Lei and Spradling, 2013, 2016). This transfer has also been proposed to facilitate oocyte selection and contribute to the formation of an organelle-enriched structure, known as the Balbiani body. Single-cell RNA sequencing by Niu and Spradling has explored the molecular mechanisms regulating cytoplasmic exchange in these cysts, providing insights into the regulation of oocyte formation (Niu and Spradling, 2022). Collectively, these studies suggest that oocyte formation through cyst breakdown follows a conserved mechanism in mammals, akin to that observed in *Drosophila*.

However, while organelle and cytoplasm transfer are widely accepted as a key process in oocyte formation, studies of this transfer primarily employ static observations, which are limited in capturing the dynamic nature of this process (Ikami et al, 2023; Lei and Spradling, 2013, 2016). Additionally, the necessity of cyst structures in mammalian oocyte formation has been questioned by the *Tex14* knockout mouse model, in which females lacking intercellular bridges remain fertile (Greenbaum et al, 2009; Ikami et al, 2021). These findings suggest that organelle and cytoplasm transfer in the cyst may not be essential for oocyte formation. Furthermore, a recent study reported an alternative mechanism of oocyte formation, involving autophagy-driven oocyte competition, in which dominant oocytes engulf debris from sacrificed oocytes (Zhang et al, 2026). While this mechanism presents a new perspective on oocyte formation, its validity and underlying mechanisms remain to be clarified.

In this study, we established an ex vivo culture system that enables live imaging of germ cell development in mouse gonads or ovaries, allowing real-time observation of oocyte formation. This novel application led to the discovery of unique blebbing behavior in germ cells during the mitosis-to-meiosis transition. Furthermore, we employed high-resolution 3D time-lapse tracking to investigate organelle transfer during oocyte formation. Unexpectedly, we found no detectable mitochondrial or centrosome transfer between germ cells, challenging the previous model of organelle transfer from nursing germ cells to dominant oocytes. Our findings highlight the utility of this ex vivo live imaging system for studying dynamic cellular processes and offer new insights into the mechanisms underlying oocyte formation.

# Results

## Establishment of an ex vivo culture system for mouse female germ cell development

To investigate the development of mouse female germ cells, we established an ex vivo culture system that enables continuous monitoring of developmental processes (Fig. 1A). Building upon a previously reported system for in vitro oocyte generation from pluripotent stem cells (Aizawa et al, 2023), we adapted it to culture fetal female gonads or ovaries on coverslips within a vessel, facilitating long-term live imaging. For clarity, we refer to the developing reproductive organ as a 'gonad' until E13.5, before morphological sex differentiation is complete, and as an 'ovary' from E14.5 onward. The system was first applied to E12.5 female gonads, a stage at which sex can be reliably distinguished morphologically and which precedes the mitosis-to-meiosis transition and oocyte formation (Fig. 1B). This ex vivo system allowed us to track gonadal cell development for up to 36 days (Fig. 1C,D). In this study, cultured samples are labeled by their duration ex vivo (e.g., E12.5 + 7 d refers to an E12.5 gonad cultured for 7 days). By day 21, distinguishable follicles emerged, with further expansion observed by day 35 (Fig. 1C). Dissection of gonads cultured for 36 days confirmed the presence of germinal vesicles (GVs) in oocytes, a defining characteristic of growing or fully-grown oocytes (Fig. 1D). These observations suggest the validity of our ex vivo culture system in monitoring female germ cell development and recapitulating the early stages of folliculogenesis.

## Comparison of ex vivo and in vivo germ cell development

To determine whether germ cell development in our ex vivo culture system recapitulates in vivo progression, we first examined chromosome axis formation and synapsis during meiotic prophase I. Immunostaining revealed comparable formation of SYCP3-positive chromosome axes and SYCP1 localization along these axes in both E17.5 ovaries and E12.5 + 5 d cultured gonads (Fig. 1E). Extensive colocalization of SYCP3 and SYCP1 signals is consistent with synaptonemal complex assembly and progression to the pachytene stage under ex vivo conditions.

We next assessed crossover designation by analyzing MLH1 localization. In both E18.5 ovaries and E12.5 + 6 d cultured gonads, MLH1 foci were observed along each SYCP3-positive chromosome axis with comparable distributions (Fig. 1F), consistent with pachytene progression and establishment of crossover sites. Together, these findings suggest that meiotic prophase I progression is comparable under ex vivo and in vivo conditions.

To further evaluate developmental equivalence at the transcriptional level, we isolated germ cells from in vivo and ex vivo samples by fluorescence-activated cell sorting (FACS). Germ cells were identified using Stella-ECFP reporter mice, in which the germline-specific gene *Dppa3* (also known as *Stella*) drives expression of a fusion protein consisting of STELLA and ECFP (Ohinata et al, 2008). We then quantified the expression of representative genes

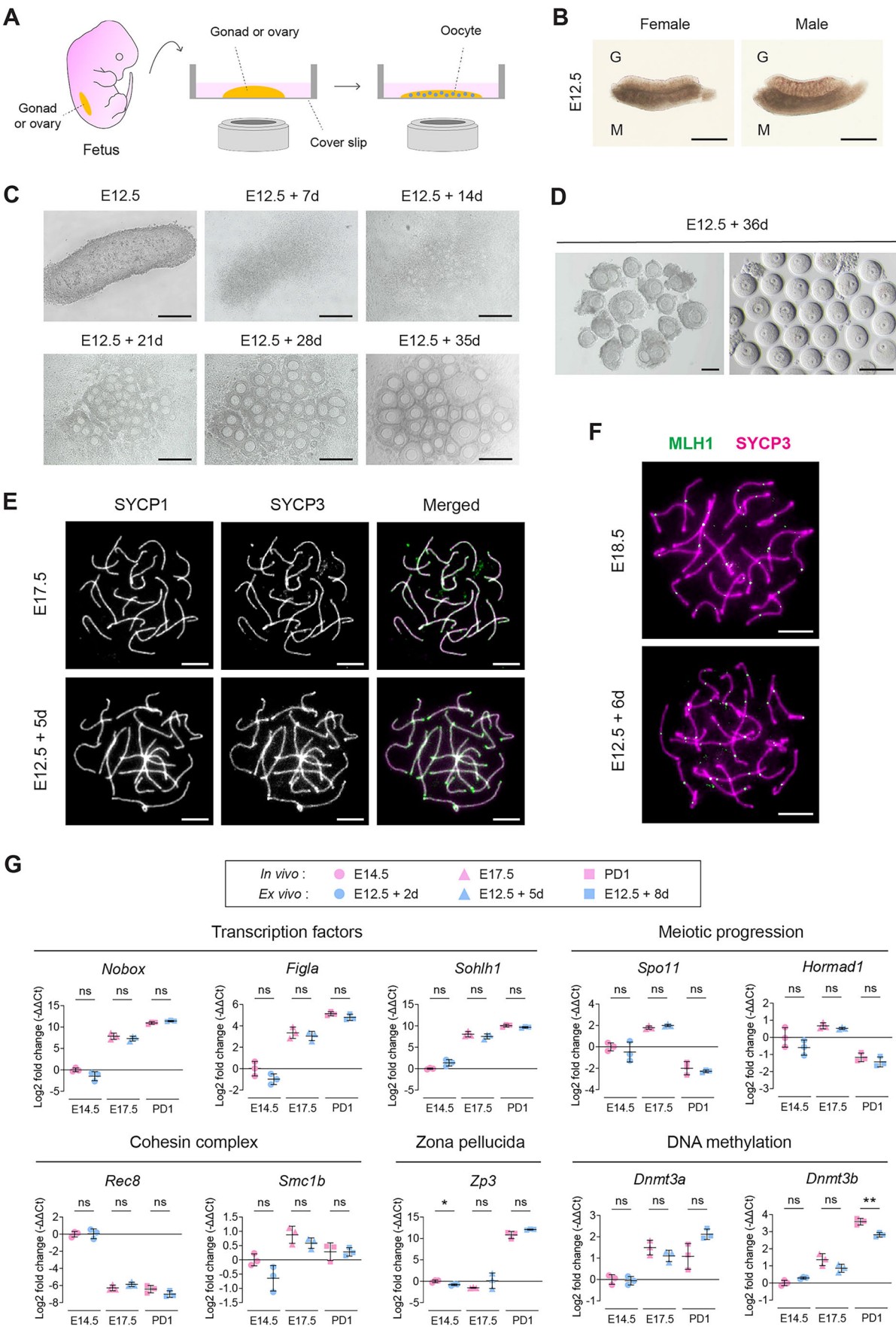

◀ **Figure 1. Establishment and validation of an ex vivo culture system for mouse female germ cell development.**

(A) A schematic illustration of the ex vivo culture system for visualizing the development of mouse female germ cells. A fetal gonad or ovary is cultured on a coverslip within the culture vessel, enabling microscopic observation of female germ cell development. (B) Representative images of female and male gonads at E12.5. The male gonad exhibits distinct testis cord structures, whereas the female gonad lacks cord formation and displays a uniformly unstructured morphology. M, mesonephros; G, gonad. Scale bar, 500 μm. (C) Representative ex vivo development of an E12.5 gonad cultured within the vessel. Bright field images were captured at days 0, 7, 14, 21, 28 and 35. Follicles became distinguishable after 21 days of culture and further expanded by day 35. Scale bar, 200 μm. (D) Bright field images of follicles (left) and oocytes (right) harvested from the E12.5 gonad shown in (C) and cultured ex vivo for 36 days. Germinal vesicles (GV) were present in each oocyte. Scale bar, 100 μm. (E) Representative chromosome spreads of oocytes derived from E17.5 ovaries and E12.5 gonads cultured ex vivo for 5 days. Immunostaining was performed using antibodies against SYCP1 and SYCP3. In merged images, SYCP1 is displayed in magenta and SYCP3 in green. Scale bar, 5 μm. (F) Representative chromosome spreads of oocytes derived from E18.5 ovaries and E12.5 gonads cultured ex vivo for 6 days. Immunostaining was performed using antibodies against MLH1 and SYCP3. Images are maximum-intensity Z-projections generated from three optical sections for each marker acquired at 200 nm Z-intervals. Scale bar, 5 μm. (G) Quantitative RT-PCR analysis of stage-specific gene expression in developing female germ cells in vivo and ex vivo. Relative expression levels of transcription factors (*Nobox*, *Figla*, *Sohlh1*), meiotic progression genes (*Spo11*, *Hormad1*), cohesin complex components (*Rec8*, *Smc1b*), zona pellucida gene (*Zp3*), and DNA methylation regulators (*Dnmt3a*, *Dnmt3b*) were measured in Stella-ECFP-positive germ cells isolated by FACS from in vivo ovaries (E14.5, E17.5, PD1) and ex vivo cultured gonads (E12.5 + 2 d, + 5 d, + 8 d). Ct values were normalized to the housekeeping gene *Rplp0*. Data are presented as −ΔΔCt (log2 fold change) relative to the mean ΔCt value of E14.5 samples. Bars shown mean ± SD from three biological replicates, each analyzed with two technical replicates. Statistical analyses were performed on ΔCt values using a two-tailed Welch's *t* test. ns, non-significant; *$P$ = 0.0215; **$P$ = 0.0067. Source data are available online for this figure.

associated with oocyte formation, including transcription factors (*Nobox*, *Figla*, and *Sohlh1*), meiotic genes (*Spo11* and *Hormad1*), cohesin components (*Rec8* and *Smc1b*), the zona pellucida gene (*Zp3*), and DNA methyltransferases (*Dnmt3a* and *Dnmt3b*) (Hamazaki et al, 2021; La Salle et al, 2004; Niu and Spradling, 2020) (Fig. 1G). No significant differences were detected between ex vivo and in vivo samples at corresponding developmental stages (E14.5, E17.5, and PD1), with the exception of *Zp3* at E14.5 and *Dnmt3b* at PD1. The difference in *Zp3* expression at E14.5 occurred at a stage when its transcript levels were minimal relative to PD1, suggesting that this variation is unlikely to reflect a biologically meaningful deviation in developmental progression. Although *Dnmt3b* expression at PD1 differed between conditions, this observation is consistent with a previous report indicating that epigenetic regulation during in vitro oocyte differentiation is particularly vulnerable to misregulation (Aizawa et al, 2023). Importantly, in both ex vivo and in vivo samples, *Dnmt3b* expression increased markedly at PD1 compared with E14.5 and E17.5, indicating that the overall developmental trajectory was maintained. Together, these results support the conclusion that ex vivo germ cells largely recapitulate in vivo transcriptional progression.

Overall, these cytological and transcriptional analyses demonstrate that germ cells cultured ex vivo progress through meiotic prophase and early oocyte differentiation in a manner highly comparable to in vivo development.

## Characterization of germ and somatic cell morphology in ex vivo culture

We next characterized germ and somatic cell populations within the ex vivo culture system. To track germ cell dynamics, we used Stella-ECFP reporter mice. The membrane-specific fluorescent dye, PlasMem Bright Red or Green, and Hoechst 33342 were also used to delineate individual cell boundaries and nuclei, respectively. Imaging of E12.5 gonads at daily intervals over an 8-day culture period captured morphological transitions in germ cells (Fig. 2A). To comprehensively assess cell morphology during oocyte formation, we generated 3D images of cells each day throughout the 8-day culture period (Figs. 2B and EV1A). These 3D reconstructions captured the morphological changes of germ and somatic

cells, illustrating the transition of germ cells from cyst-enclosed structures to individually spherical oocytes. Quantitative analysis of 3D morphology confirmed a rapid increase in germ cell volume after 6 days of culture, whereas somatic cell volume remained relatively unchanged (Fig. 2C). We also measured the distance between the nuclear center and the cell surface center as a proxy for cell polarity (Fig. 2D). While somatic cells maintained a consistently short distance without significant changes across stages, germ cells exhibited a significant increase, particularly after day 6 (Fig. 2E). These results corroborate that germ cells acquire cell polarity following meiotic entry (Elkouby et al, 2016).

## Morphometric strategy enabling marker-independent germ cell identification

To enhance the identification of germ and somatic cells in cultured tissues, we incorporated a morphometric analysis based on 2D images of E12.5 gonads cultured for 8 days (Fig. EV1B). Throughout this period, we observed a transient decrease in Stella-ECFP signal between days 5 and 7, followed by recovery on day 8, consistent with previous reports describing transient downregulation of *Stella* expression (Aizawa et al, 2023). Quantitative analysis revealed that Stella-ECFP intensity, when coupled with cell circularity, effectively distinguished the germ cell population from the somatic cell population (Fig. 2F). A threshold Stella-ECFP relative intensity of 2.5 was sufficient to identify germ cells. Furthermore, cell circularity alone proved to be a reliable distinguishing factor, with a threshold of 0.88 effectively separating germ cells from somatic cells between days 4 and 8 of the culture (Fig. 2G). In contrast, cell area was not a reliable parameter for distinguishing between germ and somatic cells (Fig. 2H).

To further validate this approach, we analyzed nuclear morphology and histone intensity using Stella-ECFP and H2B-mCherry double-reporter mice (Fig. EV2A). Germ cells were again identified using a Stella-ECFP relative intensity threshold of 2.5. Quantitative assessment of nuclear characteristics demonstrated that H2B-mCherry intensity and nuclear circularity effectively distinguished these cell populations after 3 days of culture (Fig. EV2B). An H2B-mCherry intensity threshold of 0.63 identified germ cells (≤ 0.63) and somatic cells (> 0.63) with over 98.8% accuracy between E12.5 + 3 d and E12.5 + 6 d (Fig. EV2C).

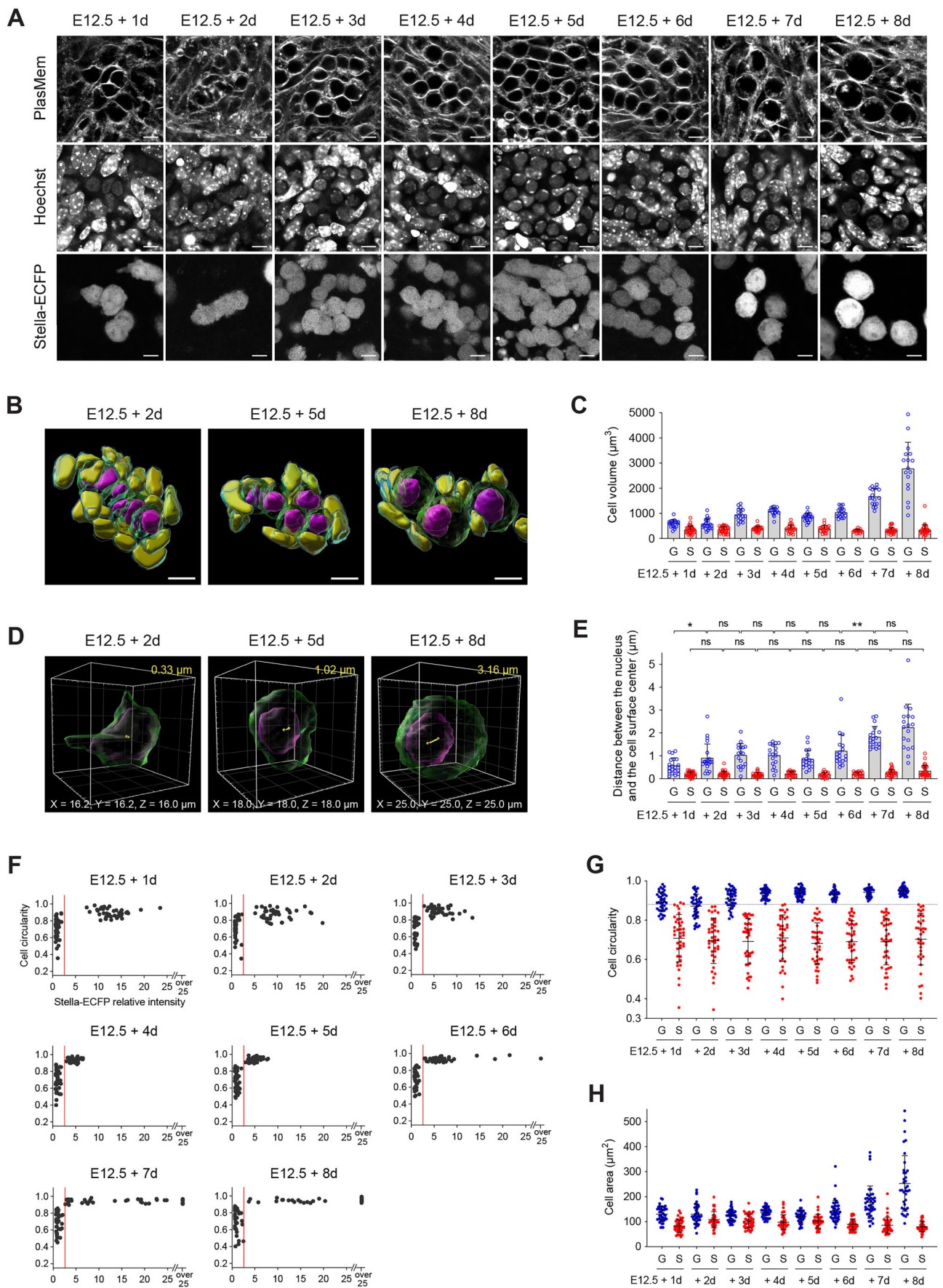

**Figure 2.   Morphometric characterization of germ and somatic cells during ex vivo culture.**

(A) Representative images of cells in E12.5 gonads expressing Stella-ECFP during an 8-day ex vivo culture. Gonads were stained with Hoechst 33342 and PlasMem Bright Red prior to imaging. Confocal z-sections were acquired at 1 μm intervals, and representative 2D sections are shown. Scale bar, 10 μm. (B) Representative three-dimensional reconstructed images of cells in E12.5 gonads expressing Stella-ECFP at 3, 5, and 8 days of ex vivo culture. Gonads were stained with Hoechst 33342 and PlasMem Bright Red. Germ cell membranes (green), germ cell nuclei (magenta), somatic cell membranes (cyan), and somatic cell nuclei (yellow) are shown. Confocal z-stacks were acquired at 1 μm intervals, and orthogonal 3D reconstructions were generated. Identical images are shown in Fig. EV1A. Scale bar, 10 μm. (C) Quantitative analysis of cell volume in cells from E12.5 gonads expressing Stella-ECFP during the 8-day ex vivo culture. Bars represent mean values + standard deviations. Volumes of germ cells (blue) and somatic cells (red) are plotted. Sample sizes (germ cells; somatic cells) were as follows: E12.5 + 1 d (19; 34), E12.5 + 2 d (21; 34), E12.5 + 3 d (18; 26), E12.5 + 4 d (17; 23), E12.5 + 5 d (18; 21), E12.5 + 6 d (17; 20), E12.5 + 7 d (18; 31), and E12.5 + 8 d (17; 33). G, germ cell; S, somatic cell. (D) Representative measurements of the distance between the nucleus and the cell surface center in germ cells from E12.5 gonads after 3, 5, and 8 days of ex vivo culture. Three-dimensional images were reconstructed from confocal z-stacks of gonads labeled with Stella-ECFP, Hoechst 33342, and PlasMem Bright Red. Germ cell membranes (green) and nuclei (magenta) are shown. Yellow spots indicate the centers of the cell membrane and the nucleus; the measured distance is shown in yellow. (E) Quantitative analysis of the distance between the nucleus and the cell surface center in cells from E12.5 gonads expressing Stella-ECFP during the 8-day ex vivo culture. Bars represent mean values + standard deviations. Distances for germ cells (blue) and somatic cells (red) are plotted. Sample sizes (germ cells; somatic cells) were as follows: E12.5 + 1 d (19; 35), E12.5 + 2 d (21; 36), E12.5 + 3 d (18; 26), E12.5 + 4 d (18; 23), E12.5 + 5 d (18; 21), E12.5 + 6 d (18; 20), E12.5 + 7 d (18; 31), and E12.5 + 8 d (18; 34). Statistical analysis was performed using a $t$ test with Welch's correction. ns, non-significant; *$P = 0.0472$; **$P = 0.0045$. G germ cell, S somatic cell. (F) Quantitative analysis of Stella-ECFP intensity and cell circularity in cells observed from E12.5 gonads expressing Stella-ECFP during an 8-day ex vivo culture. Two distinct cell populations were identified: one with low Stella-ECFP intensity and low circularity, corresponding to somatic cells, and the other with high Stella-ECFP intensity and high circularity, representing germ cells. The mean Stella-ECFP intensity of the low population was normalized to a relative intensity of 1. Red lines indicate a Stella-ECFP relative intensity threshold of 2.5, used to distinguish germ cells (> 2.5) from somatic cells (≤ 2.5). (G) Quantitative analysis of cell circularity in E12.5 gonads expressing Stella-ECFP during an 8-day ex vivo culture. Germ cells (blue; Stella-ECFP relative intensity >2.5) and somatic cells (red; Stella-ECFP relative intensity ≤2.5) were distinguished based on Stella-ECFP expression intensity. A circularity threshold of 0.88 (dashed line) also classified germ cells (> 0.88) and somatic cells (≤ 0.88) between E12.5 + 4 d and E12.5 + 8 d. Bars represent mean values ± standard deviations. (H) Quantitative analysis of cell area in E12.5 gonads expressing Stella-ECFP during an 8-day ex vivo culture. Germ cells (blue; Stella-ECFP relative intensity >2.5) and somatic cells (red; Stella-ECFP relative intensity ≤2.5) were distinguished based on Stella-ECFP expression intensity. Bars represent mean values ± standard deviations. (F–H) Sample numbers for quantification (germ cells; somatic cells; gonads) were as follows: E12.5 + 1 d (40; 40; 4), E12.5 + 2 d (40; 40; 7), E12.5 + 3 d (40; 40; 7), E12.5 + 4 d (40; 40; 7), E12.5 + 5 d (40; 40; 10), E12.5 + 6 d (40; 40; 7), E12.5 + 7 d (40; 40; 8), and E12.5 + 8 d (36; 36; 8). G germ cell, S somatic cell. Source data are available online for this figure.

Likewise, a nuclear circularity threshold of 0.93 reliably distinguished germ cells (≤ 0.93) from somatic cells (> 0.93) with over 99.0% accuracy between E12.5 + 3 d and E12.5 + 7 d (Fig. EV2D). In contrast, consistent with cell area analysis (Fig. 2H), nuclear area did not serve as a reliable parameter for distinguishing these cell populations (Fig. EV2E).

These analyses identified a set of morphological features that distinguish germ cells from somatic cells during oocyte formation. In addition to Stella-ECFP intensity, cell circularity, nuclear circularity, and histone intensity serve as reliable markers to distinguish germ cells from somatic cells beyond 3-days culture of E12.5 gonads. These markers were subsequently utilized to identify germ cells without reliance on Stella-ECFP reporters in the following experiments.

## Quantitative validation of meiotic axis formation dynamics in ex vivo culture

Having established a morphometric strategy for marker-independent identification of germ cells (Figs. 2 and EV2), we next examined whether meiotic axis formation progressed with comparable timing under ex vivo conditions. Germ cells were identified based on mCherry-SYCP3 signals up to E17.5 in vivo or E12.5 + 5 d ex vivo, and by a nuclear circularity exceeding 0.93 at later stages (Fig. EV3A,B). Using this approach, E12.5 gonads cultured ex vivo for 3-9 days were aligned with ovaries spanning the corresponding developmental stages from E15.5 to PD2. Distinct chromosome axis formation was observed in E12.5 + 6 d gonads, comparable to that observed in E18.5 ovaries. Quantification of the proportion of germ cells exhibiting axis formation demonstrated no significant differences between in vivo and ex vivo samples at corresponding stages (Fig. EV3C). Together, these data indicate

that meiotic chromosome axis formation initiates and progresses with comparable timing in the ex vivo culture system, independent of Stella reporter-based identification.

## Autonomous expansion and synchronous degeneration of germ cells

To comprehensively capture oocyte formation dynamics, we performed long-term live imaging of E12.5 + 4 d gonads cultured ex vivo. At this stage, germ cells were identified by their spherical morphology and low H2B-mCherry fluorescence intensity (Figs. 2G and EV2C). Continuous imaging of gonads expressing H2B-mCherry and stained with PlasMem Bright Green enabled tracking of individual germ and somatic cells over a 5-day period (Fig. 3A; Movie EV1).

Germ cells exhibited a marked increase in size over time, with cell area expanding more than threefold during the imaging period (Fig. 3A,B). Importantly, expansion accelerated after cells separated from the cluster, which likely represents a germ cell cyst, with the rate of area increase approximately threefold higher than before separation. These observations indicate that germ cells undergo autonomous growth following cyst breakdown.

To assess whether 2D area measurements reliably reflect physiological oocyte growth, we compared germ cell areas measured in vivo and ex vivo across corresponding developmental stages (Fig. 3C). Although ex vivo germ cells tended to be slightly larger at some stages, no significant differences were detected at any time point (all $P > 0.05$). Moreover, the oocytes tracked in long-term imaging (Fig. 3B) fell within the interquartile range (25th–75th percentile) of the corresponding ex vivo populations (Fig. 3C). The oocyte areas measured at E12.5 + 5 d (130.9 μm²), E12.5 + 7 d (187.8 μm²), and E12.5 + 9 d (328.4 μm²) each lay within the corresponding population distribution, indicating that

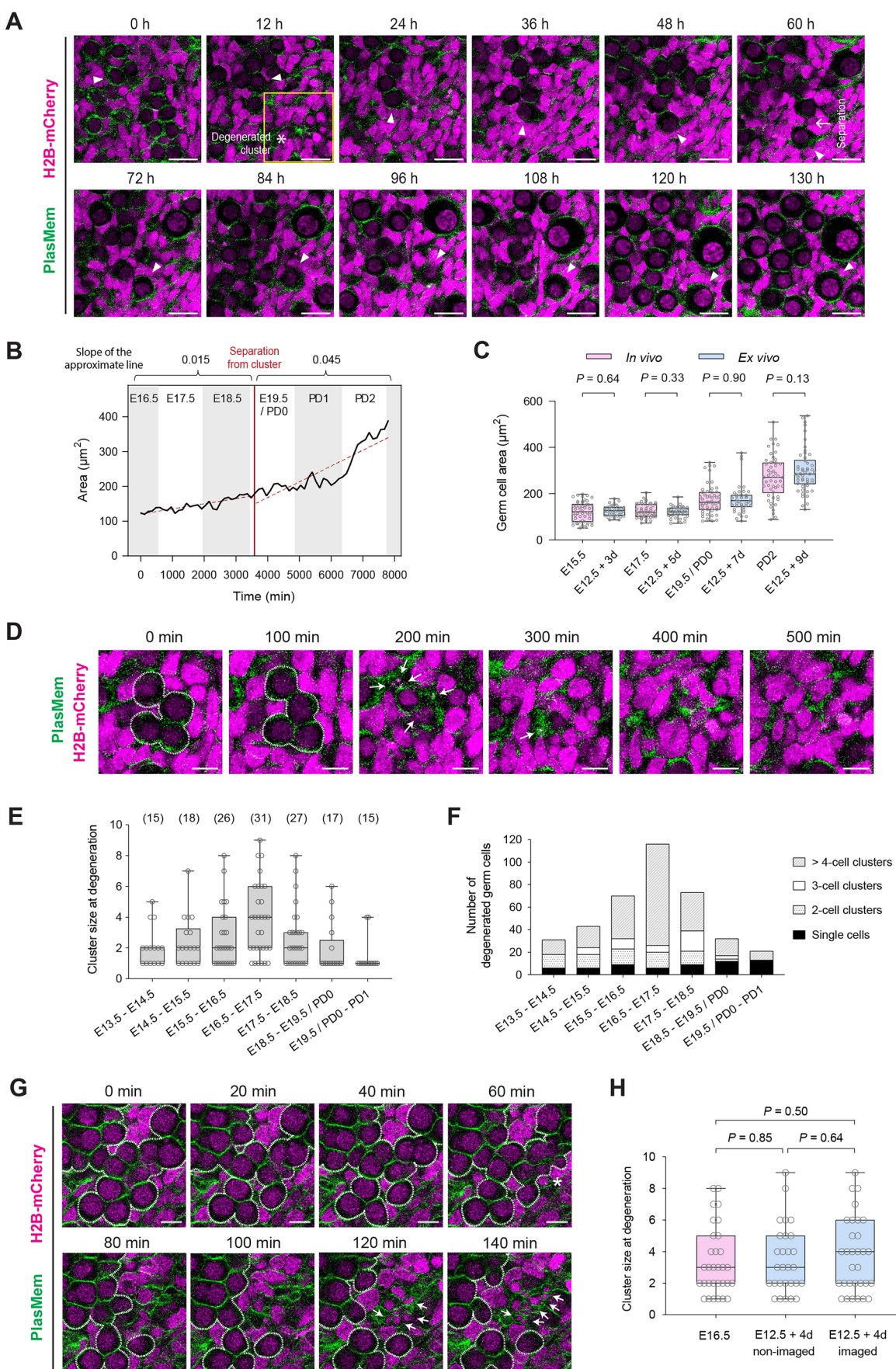

**Figure 3.   Live imaging analysis of germ cell development and cluster degeneration.**

(A) Representative live imaging of oocyte formation. An E12.5 + 4 d gonad expressing H2B-mCherry was stained with PlasMem Bright Green, and imaged every 10 min continuously for 130 h. PlasMem (green) and H2B-mCherry (magenta) signals are shown as merged images. Arrowheads mark tracked germ cells/oocytes. Asterisks indicate a degenerated cluster of germ cells, and an arrow marks the separation of the germ cell from a cluster. The region enclosed by the yellow square is shown in Fig. 3D at higher magnification and finer temporal resolution. Scale bar, 20 μm. See also Movie EV1. (B) Time-lapse quantification of the area of the germ cell tracked in (C). Cell area was measured every 120 min over a 130-h period. The red line marks the time point at which the germ cell separated from the cluster. Dotted lines indicate approximate trends in cell area before and after separation. The slope of each trend line is shown above the plot. Ex vivo time points are converted to corresponding in vivo developmental times, as shown at the top of the plot. (C) Comparison of germ cell area between in vivo (pink) and ex vivo (blue) gonads/ovaries from E15.5 to PD2. Three-dimensional confocal z-stacks were acquired, and the single z-plane containing the maximal cross-sectional area of each targeted germ cell was selected. Cell boundaries were manually delineated by tracing the PlasMem signal. Data points (circles) represent areas of individual germ cells. Box-and-whisker plots show the 25th, 50th (median), and 75th percentiles (box), with whiskers indicating the minimum and maximum values. Identical datasets for E12.5 + 3 d, E12.5 + 5 d, and E12.5 + 7 d are shown in Fig. 2H. Statistical comparisons were performed using Welch's *t* test. Sample sizes (germ cells; gonads/ovaries) were: E15.5 (50; 3), E12.5 + 3 d (40; 7), E17.5 (50; 3), E12.5 + 5 d (40; 10), PD0 (50; 4), E12.5 + 7 d (40; 8), PD2 (50; 4), and E12.5 + 9 d (50; 3). (D) Time-lapse view of the region marked in (A), showing synchronized degeneration of a germ cell cluster at higher magnification and finer temporal resolution. Images were extracted from the same time-lapse dataset as in (A) and are shown at 100-min intervals. Dotted lines outline the germ cell cluster; arrows indicate particles remaining after degeneration. See also Movie EV1. (E) Quantification of germ cell cluster size at the time of degeneration in E12.5 gonads cultured ex vivo for 1 to 8 days. Degenerating germ cells were counted for each event within 12 randomly selected 100 μm × 100 μm regions per developmental window. The data are plotted according to the corresponding in vivo developmental ages (E13.5 to PD1), converted from ex vivo time points (E12.5 + 1 d to E12.5 + 8 d). Data points (circles) represent the cluster sizes observed at individual degeneration events. Box and whisker plots show the 25th, 50th (median), and 75th percentiles (box), with whiskers indicating the minimum and maximum values. Numbers in brackets indicate the number of degeneration events observed at each developmental window. For each developmental window, six E12.5 gonads cultured ex vivo were analyzed for quantification. (F) Cumulative number of degenerated germ cells in E12.5 gonads cultured ex vivo for 1 to 8 days, categorized by cluster size at the time of degeneration. Degeneration events were grouped into four categories based on the number of germ cells per cluster: single cells, 2-cell clusters, 3-cell clusters, and clusters of more than 4 cells. Developmental ages correspond to the in vivo timeline (E13.5 to PD1), converted from ex vivo time points (E12.5 + 1 d to E12.5 + 8 d). Quantification was based on the same dataset used in (E). For each developmental window, six E12.5 gonads cultured ex vivo were analyzed. (G) Representative time-lapse images of an E16.5 ovary illustrating synchronized degeneration of a germ cell cluster. The ovary, expressing H2B-mCherry, was stained with PlasMem Bright Green and imaged ex vivo every 10 min for 24 h. Merged images show PlasMem (green) and H2B-mCherry (magenta). Dotted lines outline the germ cell cluster; the asterisk marks germ cells exhibiting abnormal morphology; arrows indicate residual particles following degeneration. Scale bar, 10 μm. See also Movie EV2. (H) Quantification of germ cell cluster size at the time of degeneration in freshly isolated E16.5 ovaries (referred to as "E16.5"), E12.5 gonads cultured for 4 days without prior imaging and subjected to live imaging only after the 4-day culture period ("E12.5 + 4 d non-imaged"), and E12.5 gonads cultured for 4 days under continuous live imaging conditions ("E12.5 + 4 d imaged"). For each degeneration event, the number of degenerating germ cells was recorded within twelve randomly selected 100 μm × 100 μm regions per developmental window. Data points (circles) represent cluster sizes for individual degeneration events. Box-and-whisker plots depict the 25th, 50th (median), and 75th percentiles (box), with whiskers indicating minimum and maximum values. Seven E16.5 ovaries were analyzed for "E16.5," six gonads for "E12.5 + 4 d non-imaged," and six gonads for "E12.5 + 4 d imaged." Identical data from "E12.5 + 4 d imaged" are shown in Fig. 3E. Statistical comparison was performed using Welch's corrected *t* test. Source data are available online for this figure.

the tracked cells represent typical germ cell growth behavior rather than outliers.

Taken together, these results support the interpretation that oocyte enlargement during this window occurs predominantly through cell-autonomous expansion. This contrasts with the prevailing model proposing that oocyte growth primarily results from cytoplasmic transfer within germ cell cysts (Lei and Spradling, 2016). Instead, our findings are consistent with a recent report suggesting cytoplasmic enrichment through engulfment of debris from degenerated oocytes (Zhang et al, 2026).

In addition to cell growth, we frequently observed synchronous degeneration of germ cells within individual clusters (Fig. 3A,D), often accompanied by residual histone signals indicative of apoptosis (Lei and Spradling, 2013). To quantify this process, degenerating germ cells were counted in 12 randomly selected 100 μm × 100 μm regions from E12.5 gonads cultured ex vivo, corresponding to in vivo stages E13.5 to PD1 (Fig. 3E). Degeneration of multicellular clusters was common throughout this period, with cluster size peaking between E16.5 and E17.5 (median = 4 cells per degenerating cluster). Degeneration events were further categorized by cluster size, and the cumulative number of degenerated germ cells was quantified (Fig. 3F). Across the observed window, single-cell degeneration accounted for only 15.8% of events, whereas multicellular degeneration (≥ 3 cells) predominated, particularly during E15.5–E16.5 (67.1%), E16.5–E17.5 (82.8%), and E17.5–E18.5 (71.2%).

To determine whether synchronous cluster degeneration also occurs under near-physiological conditions, we performed live imaging of freshly isolated E16.5 ovaries. Because synchronous events cannot be reliably inferred from fixed samples, ovaries were cultured briefly ex vivo and imaged for 24 h. Under these conditions, we observed synchronous degeneration of germ cells within clusters (Fig. 3G; Movie EV2), accompanied by residual histone signals consistent with apoptosis. Quantification of cluster size at the time of degeneration revealed a comparable range (1–8 cells) between E16.5 ovaries and E12.5 gonads cultured for 4 days under continuous imaging conditions (referred to as "E12.5 + 4 d imaged"), with no significant difference ($P = 0.50$; Fig. 3H). To further exclude imaging-related artifacts, E12.5 gonads were cultured for 4 days without prior imaging and subjected to live imaging only after the 4-day culture period (referred to as "E12.5 + 4 d non-imaged"; Fig. 3H). Cluster sizes did not significantly differ between imaged and non-imaged samples ($P = 0.64$), indicating that synchronous degeneration is not attributable to imaging.

Together, these results demonstrate that synchronous germ cell cluster degeneration occurs both in cultured gonads and in freshly isolated ovaries, supporting its relevance to physiological oocyte development. While previous studies have described gradual germ cell loss within cysts via selective nurse cell death (Ikami et al, 2023; Lei and Spradling, 2016; Niu and Spradling, 2022), our live imaging reveals that germ cells within a cluster often undergo synchronous degeneration. These findings indicate that synchronous cluster degeneration represents a distinct pattern of cyst breakdown during oocyte formation (see "Discussion").

## Quantitative analysis of germ cell nuclear rotation

To further characterize subcellular behaviors during oocyte formation, we analyzed the dynamics of nuclear movement in germ cells. Nuclear rotation has been demonstrated to be involved in chromosome movement in spermatocytes (Lee et al, 2015; Shibuya et al, 2014) and is proposed to play a role in the maintenance of oocyte dormancy in mice (Nagamatsu et al, 2019). Time-lapse imaging at 10-second intervals captured the rotational movements of germ cell nuclei in E12.5 + 4 d gonads stained with Hoechst 33342 (Fig. EV4A; Movie EV3). Quantification of nuclear rotation speed across multiple germ cells revealed considerable cell-to-cell variability, with some nuclei exhibiting rapid movements, while others moved more gradually (Fig. EV4B–D). Velocity plots further illustrated the directionality of nuclear rotation, with data distributed across all quadrants, ranging from 16.7% to 33.3% (Fig. EV4E–G). These results indicate that nuclear rotation occurs in multidirectional rather than restricted to a defined axis.

## Dynamic blebbing in germ cells during the mitosis-to-meiosis transition

The mitosis-to-meiosis transition represents a pivotal developmental shift in germ cell development; however, the extent to which this process involves specific changes in cellular behavior remains poorly understood. To investigate potential morphological features associated with this transition, we performed live imaging with high temporal resolution during the critical window encompassing the mitosis-to-meiosis shift. Unexpectedly, we frequently observed dynamic blebbing activity in germ cells from E12.5 + 2 d gonads, characterized by transient protrusions forming and retracting at various regions of the cell surface (Fig. 4A,B; Movie EV4).

To quantify this behavior, we measured the number of blebs with a height exceeding 2 μm from E11.5 to E12.5 + 5 d (Fig. 4C). Only female gonads were used for this analysis, with sex at E11.5 determined by genotyping (Fig. EV5A). The analysis revealed that blebbing frequency increased as development progressed, peaking at E12.5 + 2 d at a rate exceeding one bleb per cell per minute, followed by a marked decline by E12.5 + 3 d (Fig. 4D). This temporal pattern coincided with the mitosis-to-meiosis transition, which occurs around E13.5–E14.5 (Ishiguro, 2023), suggesting a potential link between blebbing activity and early meiotic events. Notably, blebbing was observed predominantly at interfaces between germ cells and adjacent somatic cells, with only rare events occurring at germ cell-germ cell contacts (Fig. 4E). This directional bias suggests differential regulation of membrane dynamics at germ cell–somatic cell interfaces compared with germ cell–germ cell contacts.

Because these blebbing events were observed under ex vivo culture conditions, we next asked whether blebbing also occurs in intact ovaries in vivo. Analysis of fixed ovaries at E14.5 and E16.5 revealed membrane protrusions consistent with blebs in germ cells at E14.5, whereas such structures were markedly reduced at E16.5 (Fig. 4F). Quantification confirmed that the proportion of germ cells exhibiting blebs significantly decreased from E14.5 to E16.5 (Fig. 4G). Importantly, this decline mirrors the reduction in blebbing frequency observed in ex vivo samples from E12.5 + 2 d to E12.5 + 4 d (Fig. 4D). Together, these observations indicate that

blebbing is not an artifact of ex vivo culture but represents a transient feature of germ cells that also occurs in vivo.

Since blebbing is often linked to fundamental cellular processes such as apoptosis, cell migration, and cytokinesis, its occurrence in germ cells may reflect an active developmental transition (Ikenouchi and Aoki, 2022). Given that a subset of germ cells remains mitotically active at this stage, we initially hypothesized that blebbing might be associated with cytokinetic activity, as previously reported (Burton and Taylor, 1997; Wang et al, 2021). However, contrary to this assumption, blebbing was absent in germ cells undergoing cytokinesis, whereas neighboring germ cells displayed active blebbing in the absence of cytokinetic activity (Fig. EV5B; Movie EV5).

To investigate the mechanisms underlying blebbing in germ cells, we analyzed cytoskeletal dynamics (Fig. EV5C; Movie EV6). Co-staining of actin filaments with SiR-Actin and a membrane marker revealed that actin accumulation was associated with bleb contraction. To quantify blebbing dynamics, we measured the bleb arc length, which increased during expansion and shortened during contraction (Fig. EV5D,E). Analysis of fluorescence intensity along the bleb arc showed that PlasMem Bright Red intensity remained stable throughout the blebbing process, whereas SiR-Actin intensity increased as blebbing transitioned into the contraction phase (Fig. EV5F). These observations indicate that actin dynamics contribute to bleb retraction in germ cells.

## Regulation of blebbing by meiotic initiation signals and cytoskeletal dynamics

To elucidate the molecular mechanisms underlying blebbing, we investigated the effects of specific signaling inhibitors. Given that blebbing activity peaked during the mitosis-to-meiosis transition (Fig. 4D), we first examined its potential correlation with meiotic initiation pathways, specifically BMP and RA signaling (Ishiguro, 2023; Spiller and Bowles, 2022). A previous study has demonstrated that ex vivo culture of E11.5 gonads with LDN193189, an ALK2/3 receptor inhibitor, or BMS493, an RA receptor antagonist, disrupts meiotic initiation (Miyauchi et al, 2017). Following a similar approach (Fig. 4H), we observed that treatment with these inhibitors significantly reduced blebbing frequency in both E11.5 + 2 d and E11.5 + 3 d gonads (Figs. 4I and EV6A; Movie EV7). These results indicate that blebbing is regulated by meiotic initiation pathways through BMP and RA signaling.

We next investigated the role of cytoskeletal dynamics, prompted by observations of actin involvement (Fig. EV5C–F). E12.5 gonads were cultured ex vivo and exposed to Jasplakinolide (an actin polymerization stabilizer; $IC_{50} \approx 15–170$ nM) (Bubb et al, 1994; Senderowicz et al, 1995), Cytochalasin D (an actin polymerization inhibitor; $IC_{50} \approx 25–150$ nM) (Luxenburg et al, 2012; Sayyad et al, 2015), or Ciliobrevin D (a dynein inhibitor; $IC_{50} \approx 15–50$ μM) (Firestone et al, 2012; Tati and Alisaraie, 2021) for 16 h prior to live imaging at E12.5 + 2 d. Quantification of blebbing frequency revealed no suppression at concentrations within the $IC_{50}$ range reported in other cell types (Fig. EV6B–D). Although higher doses of Jasplakinolide (1 μM) and Cytochalasin D (500 nM) reduced blebbing, these conditions also induced significant cell death, suggesting that the observed suppression was possibly due to cytotoxic effects rather than direct inhibition of blebbing (Fig. EV6E). To further dissect the underlying

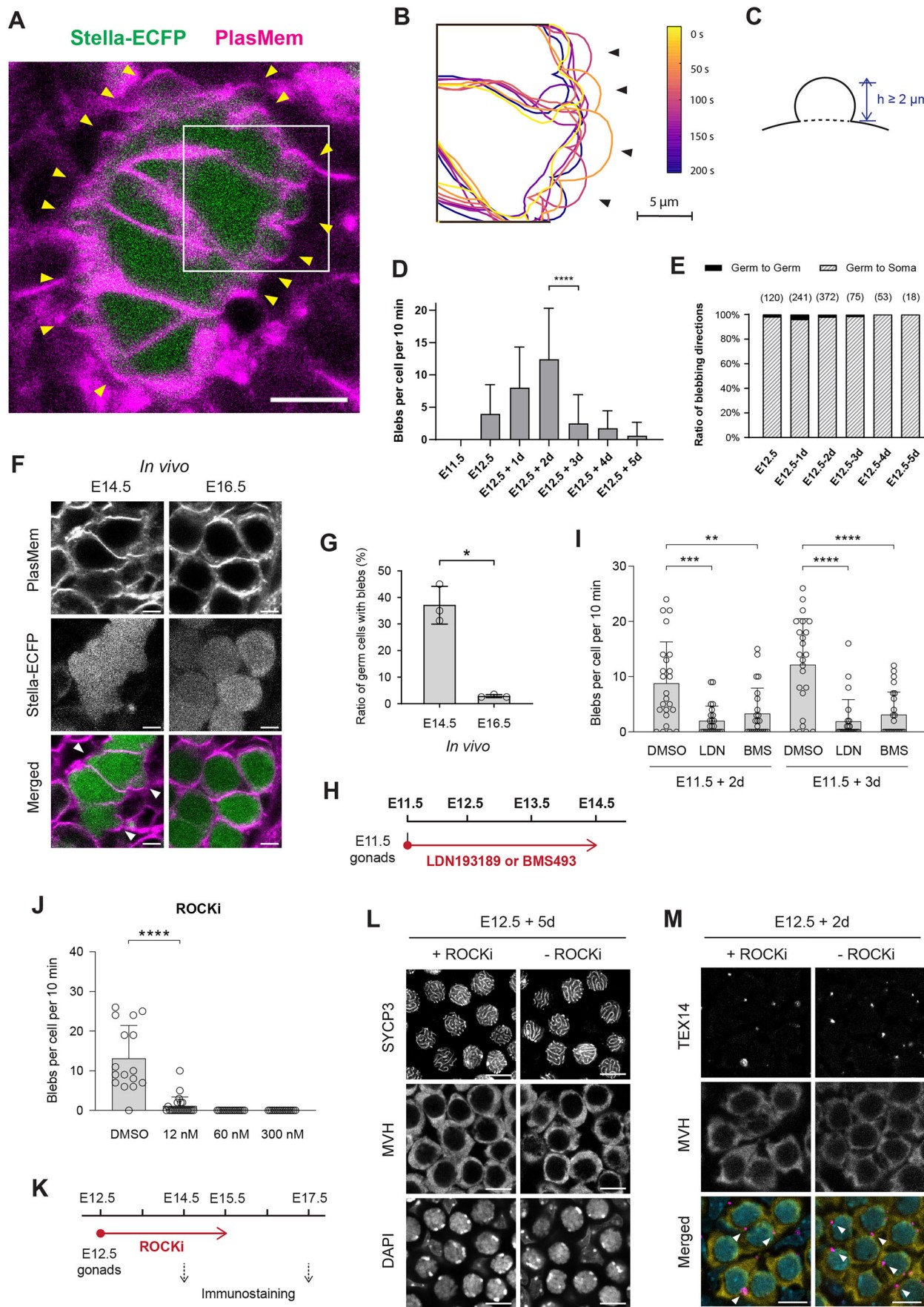

◀

**Figure 4. Dynamics of blebbing in germ cells observed during the mitosis-to-meiosis transition.**

(A) Representative image of cells in an E12.5 gonad expressing Stella-ECFP after 2 day of ex vivo culture. The gonad was stained with PlasMem Bright Red. Stella-ECFP (green) and PlasMem (magenta) signals are shown as a merged image. Arrowheads indicate blebs on germ cells. The time-course projection within the white square is shown in (B). Scale bar, 10 μm. See also Movie EV4. (B) Representative projection of germ cell membranes from the region outlined in (A). Germ cell membranes stained with PlasMem Bright Red were traced from live imaging captured every 2 s. The projection shows seven tracings over 200 s. Arrowheads indicate blebs. See also Movie EV4. (C) Schematic illustration of a bleb. A bleb is defined as an expanded region of the cell membrane with a height ($h$) exceeding 2 μm. (D) Blebbing frequency over the developmental timeline. Blebs with heights exceeding 2 μm were counted for each germ cell in female gonads expressing Stella-ECFP and stained with PlasMem Bright Red. The sex of E11.5 embryos was determined by PCR genotyping (Fig. EV5A), and only female gonads were analyzed. $N = 30$ germ cells per developmental time point. Bars represent mean values + standard deviations. Statistical analysis was performed using a $t$ test with Welch's correction. **** $P = 0.00000031$. (E) Direction of blebbing over the developmental timeline. Blebbing was categorized as either germ cell-to-germ cell or germ cell-to-somatic cell. No blebbing from somatic cells was observed. Numbers in brackets indicate the total number of blebbing observed at each developmental time point. $N = 30$ germ cells per developmental time point. (F) Representative images of germ cells in ovaries at E14.5 and E16.5 expressing Stella-ECFP following fixation. Ovaries were stained with PlasMem Bright Red. Merged images show PlasMem (magenta) and Stella-ECFP (green). Arrowheads indicate blebs on germ cells. Scale bar, 5 μm. (G) Quantification of germ cells exhibiting blebs in ovaries at E14.5 and E16.5. Bars represent mean values ± standard deviations. For each ovary, 80 germ cells were assessed for the presence or absence of blebs. $N = 3$ ovaries per stage. Statistical analysis was performed using Welch's corrected $t$ test. * $P = 0.0136$. (H) Schematic timeline of the ex vivo culture to evaluate the effects of LDN193189 or BMS493 on blebbing. Female E11.5 gonads were cultured with either inhibitor for 3 consecutive days. (I) Blebbing frequency following treatment with LDN193189 (500 nM) or BMS493 (10 μM). Bars represent mean values + standard deviations. $N = 24$ germ cells per condition. Statistical analysis was performed using a $t$ test with Welch's correction. ** $P = 0.0044$; *** $P = 0.00026$; **** $P = 0.0000053$ (E11.5 + 3 d, DMSO vs LDN); **** $P = 0.000037$ (E11.5 + 3 d, DMSO vs BMS). See also Fig. EV6A and Movie EV7. (J) Blebbing frequency following treatment with ROCKi for 16 h. The number of blebs was counted from live imaging of E12.5 + 2 d gonads expressing Stella-ECFP. Bars represent mean values + standard deviations. Statistical analysis was performed using a $t$ test with Welch's correction. Sample sizes: DMSO ($N = 16$), 12 nM ($N = 23$), 60 nM ($N = 16$), 300 nM ($N = 16$). **** $P = 0.000032$. (K) Schematic timeline of the ex vivo culture to evaluate the effect of ROCKi on meiosis initiation. E12.5 gonads were cultured in the presence of ROCKi (12 nM) for 2 or 3 consecutive days, followed by immunostaining analysis. (L) Representative immunostaining of SYCP3 in cultured gonads following ROCKi treatment. E12.5 + 5 d gonads were stained with antibodies against SYCP3 and MVH, followed by DAPI counterstaining. SYCP3-positive chromosome axes were detected in germ cells irrespective of ROCKi treatment. Scale bar, 10 μm. (M) Representative immunostaining of TEX14 in cultured gonads following ROCKi treatment. E12.5 + 2 d gonads were stained with antibodies against TEX14 and MVH, followed by DAPI counterstaining. Merged images show TEX14 (magenta), MVH (yellow), and DAPI (cyan). TEX14 signals (arrowheads) were detected between germ cells irrespective of ROCKi treatment. Scale bar, 10 μm. Source data are available online for this figure.

mechanisms, we evaluated the roles of actin polymerization and contractility. Inhibition of the Arp2/3 complex (CK666) and formin-mediated actin polymerization (SMIFH2) had minimal impact on blebbing (Fig. EV6F,G). In contrast, inhibition of myosin II using Blebbistatin markedly suppressed blebbing (Fig. EV6H), indicating that actomyosin contractility is essential for this process, consistent with findings in other cell types (Ikenouchi and Aoki, 2022). Given the role of myosin II, we further examined Rho-associated coiled-coil kinase (ROCK), which promotes actin polymerization via mDia/formin activation and enhances myosin II activity by inhibiting myosin light chain phosphatase (Amano et al, 2010). Treatment with the ROCK inhibitor H1152 significantly reduced blebbing frequency in E12.5 + 2 d gonads without affecting cell viability (Figs. 4J and EV6I). These findings underscore the importance of cytoskeletal regulation, particularly actomyosin contractility and ROCK signaling, in germ cell blebbing.

Finally, we assessed whether blebbing is functionally linked to meiotic progression. To this end, E12.5 gonads were cultured ex vivo in the presence of the ROCK inhibitor H1152 (12 nM) for 2 or 3 consecutive days, followed by immunostaining analysis (Fig. 4K). Despite continuous ROCK inhibition from E12.5 onward, meiotic markers STRA8 (E12.5 + 2 d) and SYCP3 (E12.5 + 5 d) remained expressed (Figs. 4L and EV6J), indicating that meiosis proceeds independently of blebbing. Similarly, TEX14 signals, associated with interconnected cyst formation, remained unaffected by ROCK inhibition (Fig. 4M). These findings suggest that while blebbing is regulated by RA and BMP signaling, it represents a process distinct from meiotic initiation. Given that blebbing is involved in various fundamental cellular processes, its occurrence in germ cells may reflect an active developmental transition. However, the precise functional significance of blebbing in oocyte formation remains to be elucidated (see "Discussion").

## Absence of detectable mitochondrial transfer revealed by live imaging

Organelle transfer between germ cells has been proposed as a key mechanism in oocyte formation, enabling the enrichment of cellular components essential for oocyte development. Lei and Spradling suggested that mitochondria, Golgi complexes, centrosomes, and other cytoplasmic materials are transferred from sister cyst germ cells to the developing oocyte, primarily between E16.5 and E19.5/PD0 (Lei and Spradling, 2016). This intercellular transfer has also been implicated in the assembly of the Balbiani body, an organelle-dense structure characteristic of early oocytes. However, direct evidence supporting such transfer remains limited. To better understand this mechanism, we utilized our live imaging system to investigate the dynamics of organelle transfer during oocyte formation.

We first performed live imaging of Golgi complexes using Golgi-EGFP, in which EGFP is fused to the N-terminal region of β-1,4-galactosyltransferase 1 (Fig. EV7A; Movie EV8) (Abe et al, 2011). This reporter effectively labeled Golgi complexes, and we observed the formation of a Golgi ring-like structure during oocyte development. Although Golgi ring formation has been associated with Balbiani body organization, this structure alone is not sufficient to define a Balbiani body. Importantly, the ubiquitous expression of the reporter across all cells prevented discrimination between intracellular redistribution and potential intercellular transfer events. We then imaged mitochondrial dynamics using Mito-EGFP, which labels mitochondria via fusion to the N-terminal region of cytochrome c oxidase subunit VIII A (Fig. EV7B; Movie EV9) (Abe et al, 2011). Although mitochondrial movements and germ cell disruption were clearly captured, evidence of intercellular transfer remained inconclusive due to

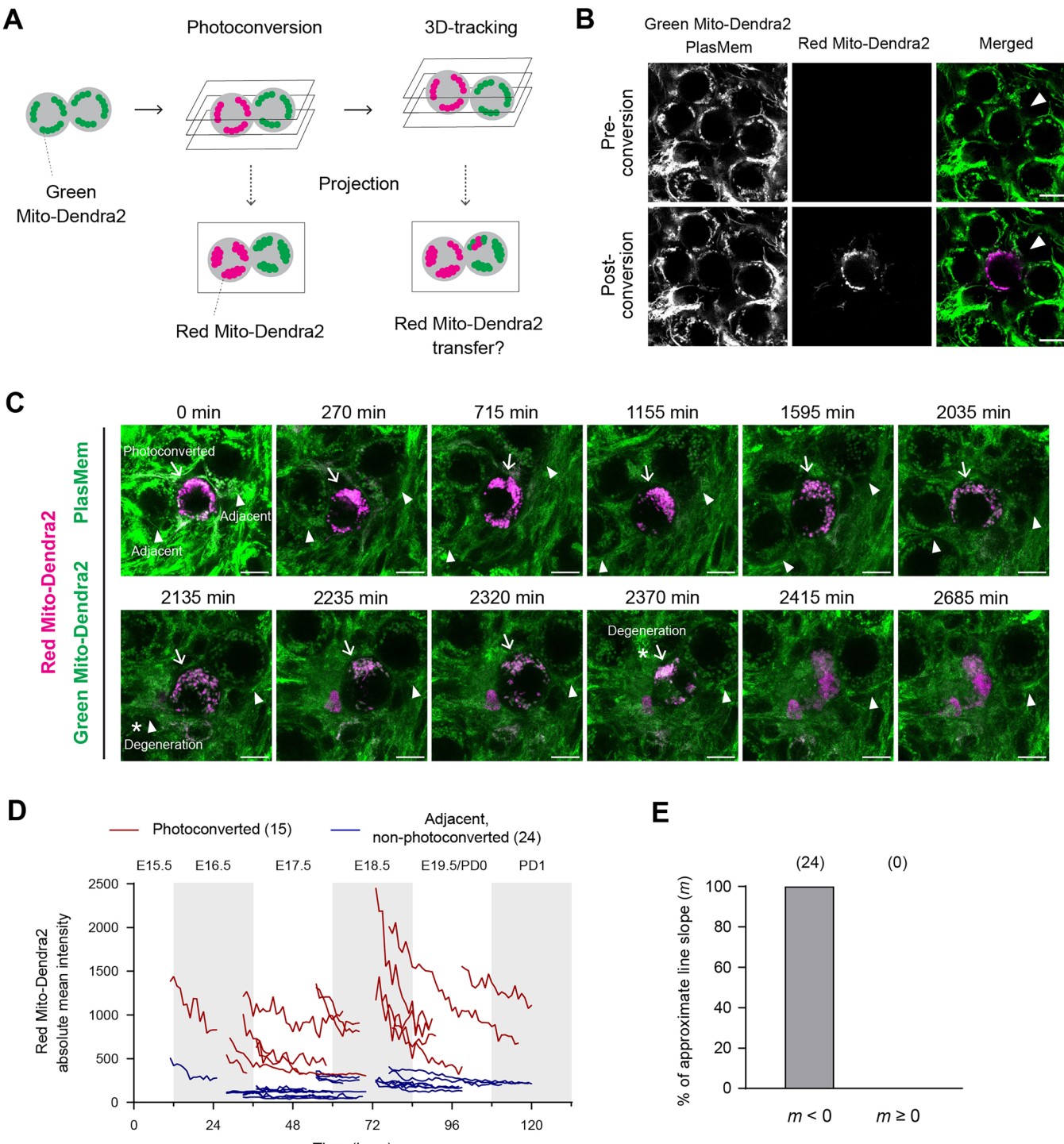

the ubiquitous expression of the reporter, similar to what was observed with Golgi-EGFP.

To enhance detection, we employed Mito-Dendra2 mice, which express Dendra2, a photoconvertible reporter (Pham et al, 2012). Mitochondria in a targeted germ cell were photoconverted from green to red fluorescence, followed by 3D time-lapse tracking and imaging to capture potential transfer to adjacent germ cells (Fig. 5A,B). As expected, the distinct red fluorescence facilitated precise mitochondrial tracking (Fig. 5C; Movie EV10). Despite the high resolution and improved visualization of our time-lapse imaging, we did not observe detectable mitochondrial transfer between germ cells across various time points, a total of 345 h, including observation of 15 photoconverted germ cells and 24 adjacent, non-photoconverted germ cells.

Quantification of photoconverted Red Mito-Dendra2 intensity further confirmed the absence of detectable mitochondrial transfer

**Figure 5.  Non-detectable mitochondrial transfer between germ cells during oocyte formation.**

(A) Schematic strategy for assessing mitochondrial transfer between germ cells. Green Mito-Dendra2 is photoconverted to Red Mito-Dendra2 in a targeted germ cell, followed by 3D time-lapse imaging of both the targeted and adjacent germ cells. The targeted cell is tracked during imaging based on the Red Mito-Dendra2 intensity. Captured 3D images are projected onto 2D planes to evaluate Red Mito-Dendra2 signals in adjacent germ cells. (B) Representative images of Mito-Dendra2 photoconversion. Green Mito-Dendra2 in a targeted germ cell (arrowheads) was photoconverted to Red Mito-Dendra2 in an E12.5 + 6 d gonad expressing Mito-Dendra2, following staining with PlasMem Bright Green. Scale bar, 10 µm. (C) Representative live imaging of an E12.5 + 6 d gonad expressing Mito-Dendra2. The gonad was stained with PlasMem Bright Green and imaged at 5-min intervals. Merged images show PlasMem (green), non-photoconverted Green Mito-Dendra2, and photoconverted Red Mito-Dendra2 signals. Arrows indicate a photoconverted germ cell, and arrowheads mark adjacent non-photoconverted germ cells. Asterisks denote the onset of germ cell degeneration. Scale bar, 10 µm. See also Movie EV10. (D) Comprehensive time-lapse quantification of photoconverted Red Mito-Dendra2 intensity during oocyte formation. Green Mito-Dendra2 in a germ cell within cultured gonads/ovaries was photoconverted to Red Mito-Dendra2, followed by 3D tracking and imaging at 5-min intervals. Absolute mean intensities of Red Mito-Dendra2 in photoconverted germ cells and adjacent non-photoconverted germ cells were measured and aligned by developmental time. Ex vivo time points were converted to corresponding in vivo developmental times, as shown at the top of the plot. $N = 15$ photoconverted germ cells and 24 adjacent non-photoconverted germ cells. See also Fig. EV8. (E) Summary of changes in Red Mito-Dendra2 intensity. Approximate lines on the plot of time versus Red Mito-Dendra2 intensity (D) were calculated for respective non-photoconverted germ cells adjacent to photoconverted germ cells. The slopes of these lines, denoted as $m$, were categorized into two groups: $m < 0$ or $m \geq 0$. $N = 24$ slopes. Numbers in brackets indicate the sample sizes for each category. Source data are available online for this figure.

(Figs. 5D,E and EV8). In our time-lapse analysis, we tracked the mean intensity of photoconverted Red Mito-Dendra2 in both the targeted germ cells and adjacent, non-targeted germ cells. In some adjacent germ cells, a modestly elevated Red Mito-Dendra2 intensity was already detectable at the initial time point, most likely reflecting unintended partial photoconversion of mitochondria outside the targeted region during UV illumination. Importantly, this signal declined over time rather than increasing. In addition, degenerating germ cells occasionally exhibited increased red-channel signal, consistent with degeneration-associated autofluorescence (Fig. 5C; Movie EV10). Throughout the imaging period, we did not observe a progressive increase in Red Mito-Dendra2 intensity in adjacent germ cells. Together, these analyses indicate that mitochondrial transfer is not a prominent event during this stage of oocyte formation.

## Characterization of centrosomes during oocyte formation

Given that mitochondrial transfer was not detected, we further examined centrosomes, which are visualized as discrete foci rather than clusters, allowing for more precise tracking of potential organelle transfer between germ cells. Centrosomes, marked by centrin 2 (CETN2), were analyzed using EGFP-CETN2 mice, which express a fusion protein of human CETN2 tagged with EGFP, driven by the chicken beta-actin promoter (Higginbotham et al, 2004). This approach enabled us to assess centrosome dynamics at a single-organelle resolution during oocyte formation.

To characterize centrosome distribution, we first quantified the number of EGFP-CETN2 foci at different developmental stages. Imaging of E12.5 gonads cultured ex vivo revealed considerable variation in centrosome numbers, which we categorized into single pair/cluster and two or more pairs/clusters (Fig. EV9A). Quantitative analysis showed a rapid increase in the proportion of germ cells exhibiting two or more pairs/clusters from E17.5 to E19.5/PD0 (Fig. 6A), consistent with reports of pericentrin and γ-tubulin volume expansion during this period (Lei and Spradling, 2016). Centrosome distribution relative to germ cell diameter did not exhibit rapid changes, suggesting that developmental timing plays a more significant role in centrosome number (Fig. EV9B). Time-lapse 3D imaging allowed precise visualization of EGFP-CETN2 foci within individual germ cells (Fig. EV9C; Movie EV11). However, similar to Golgi complex and mitochondrial tracking,

the widespread presence of EGFP-CETN2 signals across cells precluded clear identification of centrosome transfer between adjacent cells in the absence of targeted labeling.

To enable selective tracking, we adopted a photoconversion-based strategy. As a prerequisite for this approach, we first assessed the stability of EGFP-CETN2 signals using fluorescence recovery after photobleaching (FRAP) (Fig. EV9D,E). Photobleached EGFP-CETN2 signals did not recover significantly within 2 h, indicating low turnover of CETN2 fluorescence and supporting the feasibility of a photoconversion-based approach. We therefore employed a photoconvertible centrosome marker to directly monitor potential intercellular transfer (Fig. 6B).

## Absence of detectable centrosome transfer revealed by photoconvertible reporter imaging

For this purpose, we generated mKikGR-CETN2 mice, in which CETN2 is tagged with the photoconvertible monomeric Kikume Green-Red (mKikGR) reporter (Fig. EV10A) (Habuchi et al, 2008). Immunostaining confirmed strong colocalization of mKikGR-CETN2 with endogenous CETN2 in germ cells (94.0%), validating its utility for centrosome recognition (Fig. 6C,D). Exposure to 405 nm light successfully converted mKikGR fluorescence from green to red, enabling specific labeling of centrins within targeted germ cells (Fig. 6E).

Live imaging of photoconverted Red mKikGR-CETN2 in single germ cells allowed continuous tracking of centrin movement in 3D over 30 h. While centrins exhibited dynamic movement within individual germ cells, no intercellular transfer to adjacent, non-photoconverted germ cells was observed (Figs. 6F and EV10B; Movie EV12). Additionally, the number of photoconverted Red mKikGR-CETN2 foci occasionally fluctuated within single cells, with some foci splitting into two to four or merging into clusters (Fig. EV10C). While the biological significance of these events remains unclear, they likely contribute to the changes in centrosome numbers observed across stages. Comprehensive quantification further confirmed the absence of centrosome transfer (Figs. 6G,H and EV11). Across multiple time points, totaling 199 h of observation, including 13 photoconverted germ cells and 13 adjacent non-photoconverted germ cells, photoconverted Red mKikGR-CETN2 signals remained restricted to the initially targeted germ cells. No sustained increase in red

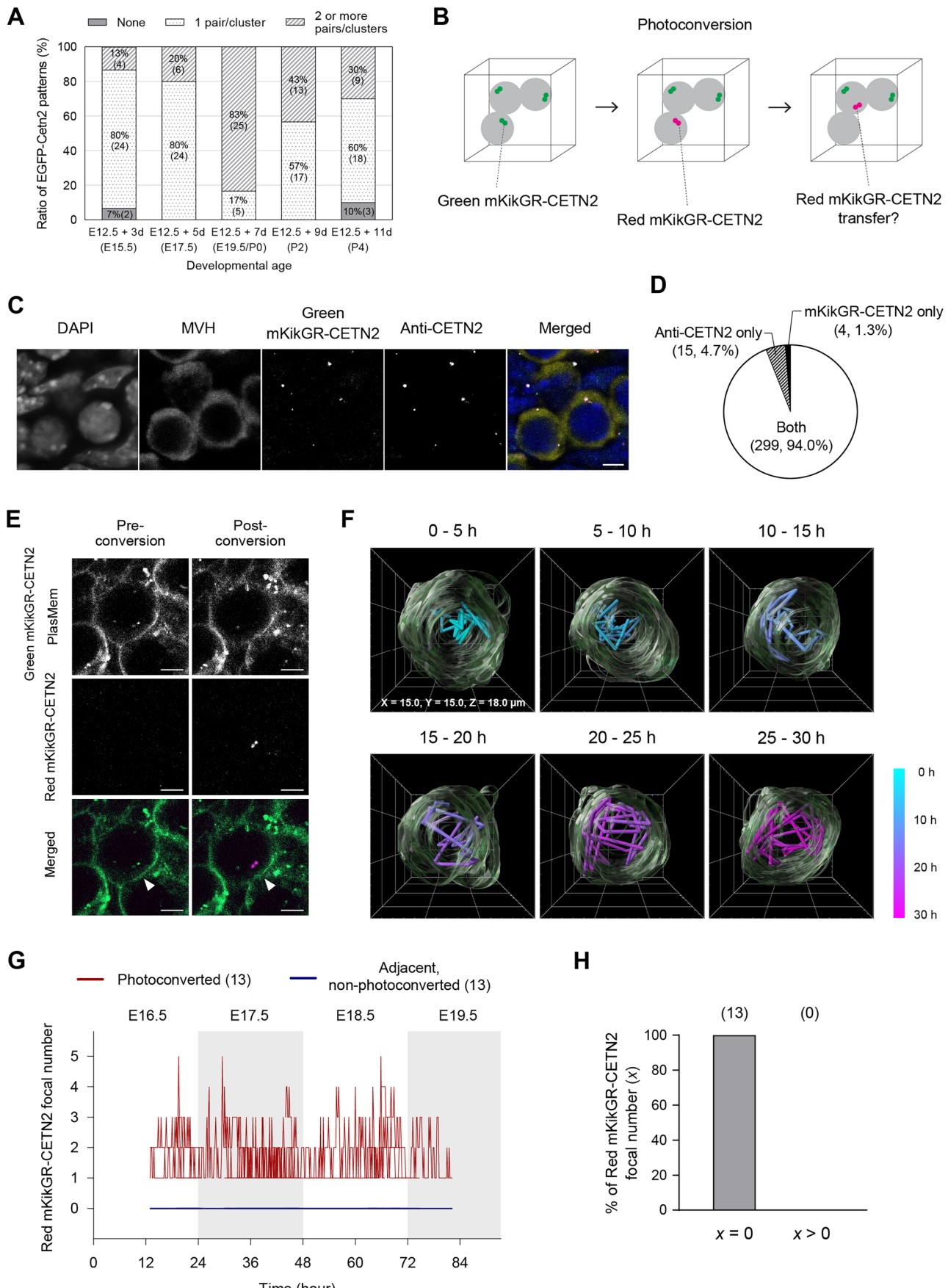

**A** Ratio of EGFP-Cetn2 patterns (%)

None | 1 pair/cluster | 2 or more pairs/clusters

**B** Photoconversion

Green mKikGR-CETN2 — Red mKikGR-CETN2 — Red mKikGR-CETN2 transfer?

**C** DAPI | MVH | Green mKikGR-CETN2 | Anti-CETN2 | Merged

**D** Anti-CETN2 only (15, 4.7%) — mKikGR-CETN2 only (4, 1.3%) — Both (299, 94.0%)

**E** Green mKikGR-CETN2

Pre-conversion | Post-conversion

PlasMem / Red mKikGR-CETN2 / Merged

**F** 0 - 5 h | 5 - 10 h | 10 - 15 h | 15 - 20 h | 20 - 25 h | 25 - 30 h

X = 15.0, Y = 15.0, Z = 18.0 µm

0 h / 10 h / 20 h / 30 h

**G** Photoconverted (13) | Adjacent, non-photoconverted (13)

E16.5 | E17.5 | E18.5 | E19.5

Red mKikGR-CETN2 focal number — Time (hour)

**H** % of Red mKikGR-CETN2 focal number (x)

(13) | (0)

x = 0 | x > 0

**Figure 6.   Generation of mKikGR-CETN2 mice and non-detectable centrin transfer between germ cells.**

(A) Distribution of EGFP-CETN2 patterns in germ cells across developmental age. EGFP-CETN2 patterns in each germ cell of E12.5 gonads cultured for 3 to 11 days were classified into three categories: None, 1 pair/cluster, and 2 or more pairs/clusters. The x axis shows the corresponding in vivo developmental ages in brackets, inferred from ex vivo culture durations. Numbers in brackets within the bars indicate germ cell counts. $N = 30$ germ cells per developmental age. See also Fig. EV9A,B. (B) Schematic of the strategy for assessing centrin transfer between germ cells. Green mKikGR-CETN2 is photoconverted to Red mKikGR-CETN2 in a targeted germ cell, followed by 3D time-lapse imaging of both the targeted and adjacent germ cells. Centrin transfer is indicated by the appearance of Red mKikGR-CETN2 in an adjacent germ cell. (C) Representative immunostaining of a mKikGR-CETN2 ovary. E17.5 ovaries expressing mKikGR-CETN2 were stained with MVH and CETN2 antibodies and counterstained with DAPI. The merged image shows DAPI (blue), MVH (yellow), mKikGR-CETN2 (green) and anti-CETN2 (magenta). Scale bar, 5 μm. (D) Summary of centrin immunostaining in mKikGR-CETN2 mouse ovaries. Green mKikGR-CETN2 and anti-CETN2 foci were assessed in E17.5 ovaries expressing mKikGR-CETN2. Focal patterns were classified into three categories: Both (colocalization of mKikGR-CETN2 and anti-CETN2), mKikGR-CETN2 only, and anti-CETN2 only. Numbers in brackets indicate foci counts and their proportions among all foci. Centrin signals were analyzed in a total of 226 germ cells from 3 ovaries. (E) Representative images of mKikGR-CETN2 photoconversion. An E12.5 + 5 d gonad expressing mKikGR-CETN2 was stained with PlasMem Bright Green, followed by 405 nm exposure targeting a centrin signal (arrowheads). In the merged image, PlasMem and unconverted Green mKikGR-CETN2 are shown in green, while photoconverted Red mKikGR-CETN2 is shown in magenta. Scale bar, 5 μm. (F) Three-dimensional reconstructed images of a germ cell with photoconverted Red mKikGR-CETN2 signals. An E12.5 + 5 d gonad expressing Green mKikGR-CETN2 was stained with PlasMem Bright Green, followed by photoconversion of Green mKikGR-CETN2 to Red mKikGR-CETN2. 3D time-lapse images of the photoconverted germ cell were acquired every 10 min with z-sections at 1.5 μm intervals. 3D reconstructions were generated at 5-h intervals over a total of 30 h and are displayed in perspective. Cell surface 3D images are shown in green with overlays of reconstructed images at 10-80 min intervals. The trajectory of photoconverted Red mKikGR-CETN2 signals is represented using a color bar indicating time-lapse. See also Fig. EV10B and Movie EV12. (G) Comprehensive time-lapse counts of photoconverted mKikGR-CETN2 foci during oocyte formation. Green mKikGR-CETN2 in a germ cell within cultured gonads/ovaries was photoconverted to Red mKikGR-CETN2, followed by 3D time-lapse imaging at 10-min intervals. The number of Red mKikGR-CETN2 foci in photoconverted germ cells and adjacent non-photoconverted germ cells was counted and aligned by developmental age. Ex vivo time points were converted to corresponding in vivo developmental ages, as shown at the top of the plot. $N = 13$ photoconverted germ cells and 13 adjacent non-photoconverted germ cells. See also Fig. EV11. (H) Summary of changes in photoconverted mKikGR-CETN2 foci in adjacent non-photoconverted germ cells. The number of Red mKikGR-CETN2 foci, denoted as x, in each non-photoconverted germ cell adjacent to 13 photoconverted germ cells (G) was classified into two categories: $x = 0$ (no Red mKikGR-CETN2 foci detected throughout the observation) and $x > 0$ (at least one Red mKikGR-CETN2 focus detected at any time point during the observation). $N = 13$ adjacent non-photoconverted germ cells. Numbers in brackets indicate the sample sizes for each category. See also Fig. EV11. Source data are available online for this figure.

fluorescence was detected in adjacent non-photoconverted germ cells, indicating that centrosomes, like mitochondria, do not undergo detectable intercellular transfer during oocyte formation. Our live imaging analyses using mitochondrial and centrosome markers provide no evidence supporting organelle transfer between germ cells, contradicting the model proposed by Lei and Spradling (Lei and Spradling, 2016). Instead, our findings align with recent data reported by Zhang and colleagues, which demonstrated no detectable cytoplasmic transfer between a total of 100 cell membrane borders using an independent imaging approach (Zhang et al, 2026).

In summary, our findings highlight several key aspects of germ cell development in the ex vivo culture system (Fig. 7). First, our cytological analyses demonstrate that the ex vivo system faithfully recapitulates meiotic prophase I progression and early folliculogenesis, including chromosome axis formation, synapsis, crossover designation, and the generation of GV oocytes. In addition, transcriptional analyses revealed stage-matched expression of oocyte-related genes under ex vivo conditions compared with in vivo development. Our live imaging analysis provided valuable insights into germ cell behavior, including spherical transformation, autonomous size expansion of individual germ cells, and synchronized cyst degeneration. Notably, we observed dynamic blebbing activity, particularly during the mitosis-to-meiosis transition, and demonstrated its regulation by meiotic initiation signals and ROCK activity. Furthermore, our investigation into organelle transfer during oocyte formation revealed an absence of detectable mitochondrial or centrosome transfer between germ cells, challenging the previously proposed model of extensive cytoplasmic transfer during cyst breakdown (see "Discussion"). Together, these results underscore the utility of our ex vivo culture system in studying these processes in real time and offer a comprehensive understanding of female germ cell development.

# Discussion

## Dynamic blebbing as a potential modulator of germ cell–somatic cell interactions

We identified pronounced but transient membrane blebbing in female germ cells during the mitosis-to-meiosis transition. Membrane blebbing has classically been described in contexts such as cytokinesis, apoptosis, and migration, where it arises from localized actomyosin-dependent cortical remodeling (Ikenouchi and Aoki, 2022). Increasing evidence, however, indicates that blebbing is not restricted to stress or motility-related processes, but can also emerge in developmental settings as a regulated form of cell surface dynamics. In mouse blastocysts, for example, cell surface fluctuations predominantly appear as membrane blebbing and represent a non-equilibrium behavior that influences cell positioning and physical cell–cell interactions (Yanagida et al, 2022).

In our study, blebs formed preferentially at interfaces between germ cells and surrounding somatic cells, rather than at germ cell–germ cell contacts. This spatial bias suggests that blebbing is selectively associated with interactions between germ cells and somatic cells and may locally modulate these interfaces. Dynamic blebbing has been proposed to act as a signaling platform by assembling membrane-proximal complexes that couple morphological fluctuations to biochemical signaling (Weems et al, 2023), raising the possibility that germ cell blebbing influences signal exchange with the somatic niche.

In the developing ovary, germ cell–somatic cell interactions play central roles in coordinating meiotic entry, promoting germ cell survival, and orchestrating subsequent follicle formation (Ishiguro, 2023; Spiller and Bowles, 2022). Notably, blebbing peaks around E14.5, coinciding with the mitosis-to-meiosis transition and integration of BMP and retinoic acid signaling. Although

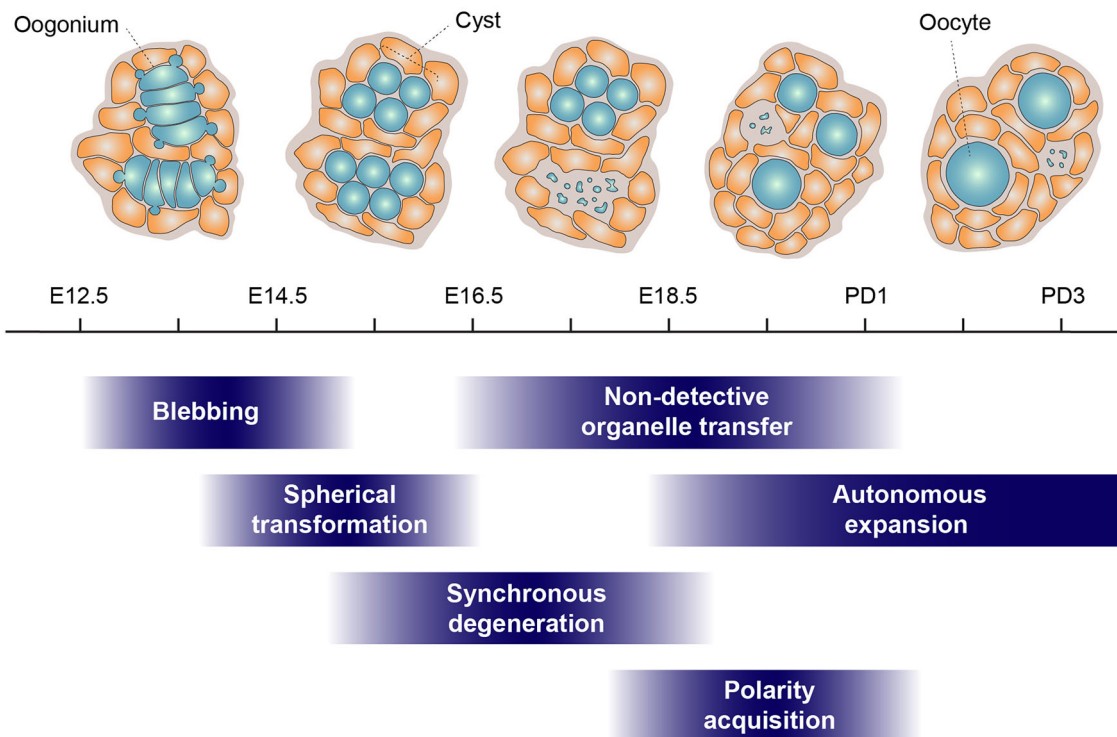

**Figure 7. Overview of dynamic cellular events in oocyte formation.**

During the mitosis-to-meiosis transition, flattened oogonia exhibit prominent membrane blebbing, with the highest frequency observed at E14.5. As development progresses, germ cells undergo spherical transformation and form interconnected cysts. Synchronous germ cell degeneration was commonly observed within cysts, and separation of germ cells was also observed in other cysts. Between E16.5 and PD1, no detectable transfer of mitochondria or centrosomes was observed. During this period, germ cells acquire polarity and undergo substantial autonomous growth. Cyan, germ cells; Orange, somatic cells.

pharmacological inhibition of blebbing did not disrupt meiotic marker expression, indicating that it is not required for meiotic initiation per se, its temporal and spatial restriction suggests that it forms part of a broader program of cortical and membrane reorganization. Blebbing may therefore facilitate transient engagement between germ cells and surrounding somatic cells by modulating local membrane architecture or signaling domains, potentially priming cells for subsequent events such as cyst breakdown or follicle assembly. Further studies will be required to determine the functional consequences of this behavior.

## Synchronous germ cell degeneration and the role of intercellular bridges

We observed a strikingly synchronous pattern of germ cell degeneration during cyst breakdown in the fetal mouse ovary. This observation contrasts with the prevailing model of progressive, asynchronous attrition of nurse cells described in previous studies (Ikami et al, 2023; Lei and Spradling, 2016; Niu and Spradling, 2022). The collective degeneration, observed prominently between E15.5 and E18.5 in our live imaging analyses, suggests the presence of a coordinated mechanism that regulates cyst disassembly as a unit rather than through individual cell fate decisions.

One possible mechanism underlying this synchronization involves the intercellular bridges that physically connect germ cells within each cyst (Ruby et al, 1969). Intercellular bridges have been

shown to mediate communication and coordination among germ cells in both male and female germlines. In the male germline, intercellular bridges are essential for coordinating meiotic progression and repressing transposable elements, as their absence leads to defects in DNA replication, synapsis, and increased genomic instability (Sorkin et al, 2025). In females, intercellular bridges have been shown to coordinate the transition from mitosis to meiosis through dilution of regulatory factors across the cyst (Soygur et al, 2021), implying that they contribute to the synchronization of developmental transitions. More recently, Vasilev and colleagues demonstrated that long-lived cytokinetic bridges in early mouse embryos mediate the simultaneous elimination of sister cells (Vasilev et al, 2025). Their study showed that pro-apoptotic proteins such as Bax and caspase-3 were able to diffuse into the directly connected sister cell, but not into non-sister cells. Based on these observations, intercellular bridges in germline cysts may permit coordinated propagation of apoptotic signals or structural collapse across connected germ cells, thereby contributing to synchronous degeneration. Importantly, this remains a speculative model that will require direct experimental validation.

These observations prompt a broader comparative question: Why do *Drosophila* germline cysts, which are also connected by intercellular bridges, show asynchronous cyst cell degeneration? One possible explanation lies in fundamental differences in cyst architecture and cytoskeletal organization between species. In *Drosophila*, intercellular bridges are transformed into large, actin-

rich ring canals that connect to the fusome, a specialized organelle that establishes polarity, regulates cell cycles, and directs material transfer toward the oocyte (Hinnant et al, 2020). This organization enables selective cytoplasmic transport and may constrain the diffusion of apoptotic signals, thereby supporting the stepwise removal of nurse cells. In contrast, mouse germ cell cysts lack a canonical fusome and appear to rely more heavily on direct cytoplasmic coupling and shared signaling through bridges. This structural difference may permit broader and more uniform propagation of death signals, leading to the synchronous degeneration we observed.

Importantly, our data do not exclude the previously reported asynchronous mode of germ cell degeneration (Ikami et al, 2023; Lei and Spradling, 2016; Niu and Spradling, 2022). In our live imaging analyses, we observed both multicellular, synchronous degeneration events and single-cell or smaller cluster degeneration events (Fig. 3E,F). While synchronous elimination accounted for the majority of events, particularly between E15.5 and E18.5, asynchronous single-cell degeneration was also present. We therefore propose that synchronous elimination represents an additional and possibly underrecognized mode of cyst breakdown during mouse oocyte formation. The coexistence of these two modes may reflect differences between developmental stages, variability among cysts, or temporal changes in how germ cell survival and elimination are regulated. Further investigation will be necessary to determine how these distinct modes are integrated and regulated during ovarian development.

## Rethinking organelle transfer and cytoplasmic enrichment

A prevailing model in mouse germ cell development posits that organelles are transferred from nurse cells to the prospective oocyte during cyst breakdown (Spradling et al, 2022). These organelles include Golgi complexes, mitochondria, and centrosomes, and the transfer is thought to occur between cyst cells connected by intercellular bridges. This model, largely extrapolated from studies in *Drosophila* and supported primarily by static observations in mouse ovaries, suggests that cytoplasmic and organelle transfer underlies the enrichment of cellular components critical for oocyte growth (Lei and Spradling, 2016; Niu and Spradling, 2022). However, our live imaging-based analysis challenges this view.

Using photoconvertible fluorescent reporters for mitochondria and centrosomes, we examined organelle dynamics within germ cell clusters during cyst breakdown. Although we observed active intracellular organelle movement, we did not detect directional transfer of either mitochondria or centrin-labeled centrosomes between neighboring germ cells. These findings suggest that, within the developmental window examined (E16.5 to E19.5/PD0) and under the spatial and temporal resolution of our imaging system, intercellular organelle transfer is unlikely to play a major role in cytoplasmic enrichment. While we cannot exclude the possibility of rare or subtle transfer events below the detection threshold, the robust detection of photoconverted signals within targeted cells argues against extensive transfer. Also, it remains possible that other classes of molecular cargo, such as RNAs or proteins, are selectively exchanged among germ cells without accompanying bulk organelle movement.

In *Drosophila*, intercellular bridges expand to form ring canals, which facilitate cytoplasmic transfer toward the oocyte (Robinson and Cooley, 1996). In contrast, in mice, the diameter of intercellular bridges reportedly decreases between E14.5 and E19.5/PD0 (Lei and Spradling, 2016). Lei and Spradling proposed the existence of plasma membrane gaps—structurally distinct from intercellular bridges—that might mediate organelle exchange between germ cells. However, the mechanism underlying the formation of these membrane gaps, as well as their presence and dynamics during cyst breakdown, remains largely uncharacterized. These insights highlight fundamental structural differences between *Drosophila* and mouse germ cell cysts, as well as potentially divergent mechanisms of organelle transfer during oocyte development.

A recent study reported a distinct mechanism of cytoplasmic enrichment in which dominant oocytes engulf remnants of degenerated germ cells through an autophagy-related process (Zhang et al, 2026). While our study was not designed to directly monitor such engulfment events, this model aligns more closely with our observation that organelles remain confined to individual cells during cyst breakdown. It also leaves open the possibility that organelle transfer may occur via engulfment following cyst dissociation.

In summary, our findings suggest that active organelle transfer between germ cells is unlikely to be the primary mechanism for cytoplasmic enrichment. Instead, the absence of detectable transfer highlights the need to revisit the models of oocyte formation and consider alternative pathways for cytoplasmic enrichment in oocytes. By enabling real-time observation of germ cell behavior, the imaging system established in this study uncovers previously inaccessible aspects and opens new directions for investigating the mechanisms underlying mammalian oocyte development.

## Methods

**Reagents and tools table**

| Reagent/resource | Reference or source | Identifier or catalog number |
|---|---|---|
| **Experimental models** | | |
| C57BL/6NCrSlc (*M. musculus*) | Japan SLC | N/A |
| EGFP-CETN2 (*M. musculus*) | Higginbotham et al, 2004; JAX | 8234 |
| mCherry-SYCP3 (*M. musculus*) | Enguita-Marruedo et al, 2018 | N/A |
| PhAM^floxed (*M. musculus*) | Pham et al, 2012; JAX | 18385 |
| R26-Golgi-EGFP (*M. musculus*) | Abe et al, 2011; RIKEN | CDB0246K |
| R26-H2B-mCherry (*M. musculus*) | Abe et al, 2011; RIKEN | CDB0239K |
| R26-Mito-EGFP (*M. musculus*) | Abe et al, 2011; RIKEN | CDB0251K |
| R26-mKikGR-CETN2 (*M. musculus*) | This study; RIKEN | CDB0470E |
| R26R-mKikGR-CETN2 (*M. musculus*) | This study; RIKEN | CDB0395E |

| Reagent/resource | Reference or source | Identifier or catalog number |
|---|---|---|
| Spo11-Cre (*M. musculus*) | Lyndaker et al, 2013; JAX | 32646 |
| Stella-ECFP (*M. musculus*) | Ohinata et al, 2008; RIKEN | CDB0465T |
| **Recombinant DNA** | | |
| hCent2-pEGFP-C1 | Addgene | #29559 |
| mKikGR-C1 | Addgene | #54656 |
| pCMV-mKikGR-CETN2 | This study | N/A |
| ROSA26 targeting vector | Srinivas et al, 2001 | N/A |
| **Antibodies** | | |
| Alexa Fluor 488 goat anti-mouse IgG | Thermo Fisher Scientific | A11029 |
| Alexa Fluor 555 goat anti-mouse IgG | Thermo Fisher Scientific | A21424 |
| Alexa Fluor 568 goat anti-rabbit IgG | Thermo Fisher Scientific | A11036 |
| Alexa Fluor 647 donkey anti-mouse IgG | Thermo Fisher Scientific | A31571 |
| Alexa Fluor 647 donkey anti-rabbit IgG | Thermo Fisher Scientific | A31573 |
| Anti-CETN2 antibody | Proteintech | 15877-1-AP |
| Anti-MLH1 antibody | BD bioscience | 51-1327GR |
| Anti-MVH antibody | Abcam | ab27591 |
| Anti-SCP1 antibody | Abcam | ab15090 |
| Anti-SCP3 antibody (mouse) | Abcam | ab97672 |
| Anti-SCP3 antibody (rabbit) | Abcam | ab15093 |
| Anti-STRA8 antibody | Shimada et al, 2023 | N/A |
| Anti-TEX14 antibody | Proteintech | 18351-1-AP |
| **Oligonucleotides and other sequence-based reagents** | | |
| crRNA | FASMAC | See Methods |
| Genotyping primers | FASMAC | See Appendix Table S1 |
| gRNA | FASMAC | See "Methods" |
| RT-qPCR primers | FASMAC | See Appendix Table S2 |
| tracrRNA | FASMAC | See "Methods" |
| **Chemicals, enzymes and other reagents** | | |
| (−)-Blebbistatin | FUJIFILM | 021-17041 |
| 4-well cover glass chamber | IWAKI | 5222-004 |
| Accutase | Nacalai Tesque | 12679-54 |
| AgeI | New England Biolabs | R3552S |
| AscleStem Dissociation Solution | Nacalai Tesque | 21777-84 |
| Beta-mercaptoethanol | Thermo Fisher Scientific | 21985023 |
| Blocking Reagent | PerkinElmer | FP1012 |

| Reagent/resource | Reference or source | Identifier or catalog number |
|---|---|---|
| BMS493 | Merck | B6688 |
| Bovine Albumin Fraction V (7.5% solution) | Thermo Fisher Scientific | 15260037 |
| Bovine Serum Albumin | Sigma-Aldrich | A3294 |
| BspEI | New England Biolabs | R0540S |
| Cell-Tak | Corning | 354240 |
| Ciliobrevin D | Merck | 250401 |
| CK 666 | Tocris Bioscience | 3950 |
| Collagenase type IV | MPB | 195110 |
| Cytochalasin D | Merck | C8273 |
| DAPI | Nacalai Tesque | 11034-56 |
| Di-Sodium Dihydrogen Ethylenediaminetetraacetate Dihydrate (EDTA) | Nacalai Tesque | 15111-45 |
| FBS | Thermo Fisher Scientific | A3161002 |
| H1152 | SYNkinase | 1221 |
| HEPES | Thermo Fisher Scientific | 15630080 |
| Hoechst 33342 | Thermo Fisher Scientific | H3570 |
| Jasplakinolide | AdipoGen | AG-CN2-0037 |
| Lab-Tek chambered cover glass | Thermo Fisher Scientific | 155383 |
| LDN193189 | MedchemExpress | HY-12071A |
| Omnipaque 350 | GE HealthCare | N/A |
| Paraformaldehyde 16% | Electron Microscopy Sciences | 15710 |
| Paraformaldehyde, powder | Nacalai Tesque | 26126-54 |
| Penicillin-streptomycin | Thermo Fisher Scientific | 15070063 |
| PicoPure RNA Isolation Kit | Thermo Fisher Scientific | KIT0204 |
| PlasMem Bright Green | Dojindo Laboratories | P504 |
| PlasMem Bright Red | Dojindo Laboratories | P505 |
| PowerUp SYBR Green Master Mix | Thermo Fisher Scientific | A25742 |
| QuantiTect Reverse Transcription Kit | QIAGEN | 205311 |
| Saponin | Nacalai Tesque | 30502-42 |
| SiR-Actin | Cytoskeleton Inc | CY-SC001 |
| SMIFH2 | Tocris Bioscience | 4401 |
| Sucrose | Nacalai Tesque | 30404-45 |
| Tris Ultrapure | Apollo Scientific | BI2888 |
| Tri-Sodium Citrate Dihydrate | Nacalai Tesque | 31404-15 |
| Triton X-100 | FUJIFILM | 169-21105 |

| Reagent/resource | Reference or source | Identifier or catalog number |
|---|---|---|
| TrueCut Cas9 Protein v2 | Thermo Fisher Scientific | A36497 |
| VECTASHIELD Vibrance® Antifade Mounting Medium | Vector Laboratories | H-1700 |
| αMEM | Nacalai Tesque | 21444-05 |
| **Software** | | |
| Adobe Illustrator | Adobe Inc | N/A |
| Fiji | Schindelin et al, 2012 | N/A |
| GraphPad Prism 9 | GraphPad Software | N/A |
| Imaris | Oxford Instruments | N/A |
| softWoRx 7.2.0 | Cytiva | N/A |
| ZEN (black edition) | ZEISS | N/A |
| ZEN (blue edition) | ZEISS | N/A |

## Mice

Mouse lines used in this study were obtained from the following sources: C57BL/6NCrSlc (Japan SLC, Inc.); Stella-ECFP (accession no. CDB0465T, RIKEN) (Ohinata et al, 2008); H2B-mCherry (also referred to as R26-H2B-mCherry; accession no. CDB0239K, RIKEN) (Abe et al, 2011); mCherry-SYCP3 (Erasmus MC) (Enguita-Marruedo et al, 2018); Golgi-EGFP (also referred to as R26-Golgi-EGFP; accession no. CDB0246K, RIKEN) (Abe et al, 2011); Mito-EGFP (also referred to as R26-Mito-EGFP; accession no. CDB0251K, RIKEN) (Abe et al, 2011); PhAM^floxed (strain no. 018385, JAX) (Pham et al, 2012); EGFP-CETN2 (strain no. 008234, JAX) (Higginbotham et al, 2004); Spo11-Cre (strain no. 032646, JAX) (Lyndaker et al, 2013). Mito-Dendra2 mice were generated by crossing PhAM^floxed mice with Spo11-Cre mice to excise the stop sequences flanked by loxP sites. mCherry-SYCP3 mice, originally on an FVB background, were backcrossed to C57BL/6NCrSlc mice.

The mKikGR-CETN2 mouse line (also referred to as R26-mKikGR-CETN2; accession no. CDB0470E, RIKEN) was generated in this study. The conditional mKikGR-CETN2 mouse line (also referred to as R26R-mKikGR-CETN2; accession no. CDB0395E, RIKEN; https://large.riken.jp/distribution/reporter-mouse.html) was generated via CRISPR/Cas9-mediated knock-in in zygotes, following the method described by Abe and colleagues (see Fig. EV10A) (Abe et al, 2020). To construct the pCMV-mKikGR-CETN2 vector, the mKikGR cDNA sequence was excised from the mKikGR-C1 vector (Addgene #54656) and ligated into the hCent2-pEGFP-C1 vector (Addgene #29559) after digestion with AgeI and BspEI restriction enzymes. The donor vector contains 5' (900 bp) and 3' (600 bp) homology arms targeting intron 1 of the ROSA26 locus, an adenovirus splicing acceptor (SA), a stop cassette flanked by loxP sites, and a bovine growth hormone polyadenylation (bpA) sequence (Srinivas et al, 2001). The mKikGR-CETN2 cDNA fragment was excised from pCMV-mKikGR-CETN2 and inserted between the stop cassette and bpA sequences to complete the donor

vector. The guide RNA (gRNA) was designed to target a site downstream of exon 1 in the ROSA26 locus (5'-CGC CCA TCT TCT AGA AAG AC-3'). A mixture of Cas9 protein (100 ng/μl) (Thermo Fisher Scientific), crRNA (5'-CGC CCA UCU UCU AGA AAG AC guu uua gag cua ugc ugu uuu g-3'; 50 ng/μl), tracrRNA (5'-AAA CAG CAU AGC AAG UUA AAA UAA GGC UAG UCC GUU AUC AAC UUG AAA AAG UGG CAC CGA GUC GGU GCU-3'; 100 ng/μl) (FASMAC), and the donor vector (10 ng/μl) was microinjected into the pronuclei of C57BL/6 N one-cell zygotes. After microinjection, zygotes were transferred into the oviduct of 0.5 dpc pseudopregnant females for implantation. A total of 28 F0 founder mice were obtained, of which 21 (8 females and 13 males) were identified as correctly targeted based on PCR genotyping. Genotyping was performed using primer pairs as follows: wild-type allele (217 bp), P1 and P2; R26R allele (270 bp), P1 and P3; and mKikGR insertion (543 bp), P4 and P5. Primer sequences are listed in Appendix Table S1.

To generate the constitutive mKikGR-CETN2 mouse line, two males carrying the targeted sequence were selected and crossed with Spo11-Cre female mice. F1 offspring were genotyped by PCR using primers P6 and P7 to amplify the R26 allele (606 bp; see Appendix Table S1). A total of 10 F1 mice (6 females and 4 males) were confirmed to have the R26 allele based on PCR genotyping and sequencing. Among these, two males were selected for breeding to establish the mKikGR-CETN2 mouse colony.

Timed-pregnant females in this study were obtained by natural mating of transgenic males aged 8 to 24 weeks with either wild-type C57BL/6NCrSlc females or transgenic females aged 7 to 16 weeks. The presence of a copulatory plug was designated as 0.5 days post coitum. All animal experiments were conducted in accordance with the ethical approvals of the Institutional Animal Care and Use Committee at RIKEN Kobe Branch.

## Ex vivo culture

Gonads or ovaries were harvested from female fetuses between E11.5 and E16.5, with the mesonephros separated using 30-gauge needles. Fetal sex was determined by examining the gonad or ovary under a stereo microscope or by PCR genotyping of the fetal mesonephros, targeting the *Sry* and *Xist* genes, as previously described (Aizawa et al, 2020). For ex vivo culture, a 4-well cover glass chamber (5222-004, IWAKI) or a Lab-Tek chambered cover glass (155383, Thermo Fisher Scientific) was coated with Cell-Tak (354240, Corning) according to the manufacturer's protocol. Each well of the chamber was overlaid with 190 μl of culture medium for the IWAKI chamber or 170 μl for the Lab-Tek chamber. The culture medium consisted of αMEM (21444-05, Nacalai Tesque) supplemented with 10% FBS (A3161002, Thermo Fisher Scientific), 10 mM HEPES (15630080, Thermo Fisher Scientific), 55 μM beta-mercaptoethanol (21985023, Thermo Fisher Scientific), and 50 U/ml penicillin-streptomycin (15070063, Thermo Fisher Scientific). Each gonad or ovary was placed at the center of a well and carefully transferred to an incubator maintained at 37 °C with 5% $CO_2$ in air. After 12–18 h, the medium was replaced with 200–240 μl of fresh culture medium to ensure the gonad or ovary remained slightly submerged. The culture was sustained by refreshing the medium every other day until further experimentation.

As appropriate for each experiment, the cell membrane, nucleus, or filamentous actin was stained using PlasMem Bright Green/Red

(P504/P505, Dojindo Laboratories), Hoechst 33342 (H3570, Thermo Fisher Scientific), or SiR-Actin (CY-SC001, Cytoskeleton Inc), respectively. The staining reagents were added to the culture medium approximately 8 h before imaging at a 2000-fold dilution for PlasMem Bright Green/Red, at a final concentration of 1 μg/ml for Hoechst 33342 or at a final concentration of 0.5 μM for SiR-Actin. Before imaging, the medium volume in each well was adjusted to over 300 μl as needed. For live imaging sessions exceeding 4 h, the surface of the medium was covered with mineral oil to minimize evaporation.

## Microscopy and imaging

All live imaging and three-dimensional imaging of gonads or ovaries were performed using Zeiss LSM780, LSM880, and LSM900 confocal microscopes equipped with GaAsP detectors, a 40× C-Apochromat 1.2 NA water immersion objective lens (Zeiss), and an environmental chamber for $CO_2$ and temperature control. The LSM780 and LSM880 were operated with ZEN (Black Edition), and the LSM900 was operated with ZEN (Blue Edition). The LSM900 was used for Mito-Dendra2 imaging, while the LSM780 and LSM880 were used for all other confocal imaging applications.

For long-term imaging of oocyte formation, a single z-plane of $512 \times 512$ pixel xy images, covering a field of $141.70 \times 141.70$ μm, was acquired at 10-min intervals. For 3D characterization of germ and somatic cells, 31 confocal z-sections were acquired at 1-μm intervals, generating $512 \times 512$ pixel xy images that covered a total volume of $70.85 \times 70.85 \times 30.00$ μm. For nuclear rotation imaging, a single z-plane of $512 \times 512$ pixel xy images, covering $26.57 \times 26.57$ μm, was acquired at 10-s intervals for at least 20 min. For blebbing imaging, a single z-plane of $512 \times 512$ pixel xy images, covering $53.14 \times 53.14$ μm, was acquired at 2- or 3-s intervals for at least 10 min. For Golgi-EGFP imaging, a single z-plane of $512 \times 512$ pixel xy images, covering $70.85 \times 70.85$ μm, was acquired at 4-min intervals over a period of 72 h. For Mito-EGFP imaging, a single z-plane of $512 \times 512$ pixel xy images, covering $53.14 \times 53.14$ μm, was acquired at 1-min intervals for 14 h. For Mito-Dendra2 imaging, photoconversion of the Dendra2 protein was performed using the Bleaching option in ZEN software. The cytoplasmic region of a single germ cell was selected, and photoconversion was induced by applying 50 iterations of irradiation with a 405 nm laser at 5% output. Following photoconversion, Mito-Dendra2 signals were tracked for 4D spatial and temporal imaging using an adapted version of a macro initially developed by the research group of Dr. Jan Ellenberg (Politi et al, 2018). Three confocal z-sections were acquired at 2-μm intervals, generating $512 \times 512$ pixel xy images that covered a total volume of $53.24 \times 53.24 \times 4.00$ μm at 5-min intervals. For EGFP-CETN2 imaging, 12 confocal z-sections were acquired at 1.5-μm intervals, generating $512 \times 512$ pixel xy images covering a total volume of $106.27 \times 106.27 \times 16.50$ μm at 10-min intervals over a 24-h period. For mKikGR-CETN2 imaging, photoconversion of the mKikGR protein was performed using the Bleaching option in ZEN software. A centrin focus region within a single germ cell was selected, and photoconversion was induced by irradiation with a 405 nm laser at 1% output with 20 iterations. Following photoconversion, 13 confocal z-sections were acquired at 1.5-μm spacing, generating $512 \times 512$ pixel xy images covering a total volume of $106.27 \times 106.27 \times 18.00$ μm at 10-min intervals.

## Chromosome spreads and immunostaining

Chromosome spreads were prepared using oocytes isolated from E17.5 or E18.5 fetal ovaries (in vivo) and from E12.5 gonads cultured ex vivo for 5 or 6 days, respectively. Ovaries or cultured gonads were incubated in 1 mg/ml collagenase type IV at 37 °C for 20 min to facilitate tissue dissociation, followed by centrifugation. The cell pellet was resuspended in hypotonic buffer (30 mM Tris-HCl, pH 7.5; 17 mM trisodium citrate; 5 mM EDTA; 50 mM sucrose) and incubated for 20 min at room temperature to induce nuclear swelling. After centrifugation, cells were resuspended in 100 mM sucrose and gently dissociated by repeated pipetting to obtain a single-cell suspension. An equal volume of freshly prepared fixation solution (PBS containing 1% paraformaldehyde and 0.1% Triton X-100) was added, and the suspension was mixed gently. Approximately 20–30 μl of the fixed cell suspension was dropped onto clean glass slides and allowed to spread and air dry under ambient conditions. Slides were stored at −80 °C until immunostaining.

For immunostaining, slides were equilibrated to room temperature and washed twice in PBS containing 0.1% Triton X-100 and once in PBS. Blocking was performed in PBS containing 5% BSA for 20 min at room temperature. Primary antibodies diluted in blocking buffer were applied and incubated overnight at 4 °C in a humidified chamber. Slides were then washed twice in PBS containing 0.1% Triton X-100 and once in PBS before incubation with secondary antibodies for 1 h at room temperature. After washing (two times in PBS containing 0.1% Triton X-100 and once in PBS), slides were mounted in VECTASHIELD Vibrance mounting medium (Vector Laboratories). Primary antibodies used were rabbit anti-SYCP1 (Abcam, ab15090; 1:400), mouse anti-SYCP3 (Abcam, ab97672; 1:400), rabbit anti-SYCP3 (Abcam, ab15093; 1:400), and mouse anti-MLH1 (BD Biosciences, 51-1327GR; 1:50). Secondary antibodies were Alexa Fluor 488 goat anti-mouse IgG (Thermo Fisher Scientific, A11029; 1:500) and Alexa Fluor 568 goat anti-rabbit IgG (Thermo Fisher Scientific, A11036; 1:500). Images of chromosome spreads were acquired on a DeltaVision widefield system (Cytiva) mounted on an IX71 microscope (Olympus) and equipped with a CoolSNAP HQ2 camera (Photometrics), using softWoRx 7.2.0 software (Cytiva). For MLH1/SYCP3 co-immunostaining, Z-stacks were captured and maximum-intensity projections were generated using Fiji software as described in the corresponding figure legend.

## Fluorescence-activated cell sorting

Fluorescence-activated cell sorting (FACS) was performed to isolate Stella-ECFP-positive germ cells from ovaries or gonads harvested from Stella-ECFP transgenic fetuses. In vivo samples were collected at E14.5, E17.5, and PD1, and ex vivo samples were obtained from E12.5 gonads cultured for 2, 5, or 8 days. The dissociation and sorting strategy was based on a previously described protocol (Shimamoto et al, 2019), with minor modifications. Briefly, freshly isolated ovaries or ex vivo-cultured gonads were enzymatically dissociated by incubation in AscleStem Dissociation Solution (Nacalai Tesque) at 37 °C for 30 min, followed by treatment with Accutase (Nacalai Tesque) at 37 °C for 5 min to facilitate single-cell dissociation. Tissues were gently triturated by pipetting to obtain a single-cell suspension. Cells were centrifuged and resuspended in PBS containing 0.1% BSA, and the suspension was passed through

a cell strainer to remove debris and aggregates. Germ cells were identified and isolated based on Stella-ECFP fluorescence following the previously described gating strategy (Shimamoto et al, 2019). Cell sorting was performed using a MA900 cell sorter (SONY). Stella-ECFP-positive cells were collected, pelleted by centrifugation, snap-frozen in liquid nitrogen, and stored at $-80\,°C$ until RNA extraction.

## Quantitative reverse transcription PCR

Total RNA was extracted from FACS-isolated Stella-ECFP-positive germ cells using the PicoPure RNA Isolation Kit (Thermo Fisher Scientific) according to the manufacturer's instructions. Complementary DNA (cDNA) was synthesized using the QuantiTect Reverse Transcription Kit (QIAGEN), which includes a genomic DNA elimination step to remove contaminating DNA. Quantitative PCR was performed using PowerUp SYBR Green Master Mix (Thermo Fisher Scientific) on a QuantStudio 5 Real-Time PCR System (Thermo Fisher Scientific). Each biological replicate was analyzed with two technical replicates per gene. The following genes were examined: transcription factors (*Nobox, Figla, Sohlh1*), meiotic progression genes (*Spo11, Hormad1*), cohesin complex components (*Rec8, Smc1b*), zona pellucida gene (*Zp3*), and DNA methylation regulators (*Dnmt3a, Dnmt3b*). Primer sequences are listed in Appendix Table S2. Cycle threshold (Ct) values were normalized to the housekeeping gene *Rplp0* to calculate ΔCt values for each sample. For comparisons across developmental stages, ΔCt values were referenced to the mean ΔCt value of E14.5 in vivo samples to calculate ΔΔCt. Data are presented as −ΔΔCt values (log2 fold change relative to E14.5).

## Characterization of germ and somatic cell morphology

For three-dimensional analysis, confocal z-stack images were reconstructed using Imaris software (Oxford Instruments). Cell and nuclear surfaces were segmented by manually delineating boundaries in each z-section based on PlasMem Bright Red and Hoechst signals, respectively. Parameters including cell volume, cell sphericity, and nuclear sphericity were measured from these segmentations. The distance between the nuclear center and the cell surface center was calculated based on their spatial coordinates derived from the 3D reconstructions.

For 2D analysis, imaging data were processed using Fiji software (Schindelin et al, 2012). Cell boundaries were manually delineated by tracing the PlasMem Bright Red signal. Cellular parameters including cell circularity, cell area, and Stella-ECFP intensity were measured based on these boundaries. Nuclear boundaries were defined using the H2B-mCherry signal, and nuclear circularity, nuclear area, and H2B-mCherry intensity were quantified accordingly. Measurements obtained from Fiji software were exported and analyzed using GraphPad Prism (GraphPad Software). Quantitative data were plotted at each time point to assess changes in cellular parameters over the course of the culture period.

## Long-term analysis of germ cell growth and degeneration

Cell boundaries of tracked germ cells were manually delineated by tracing the PlasMem Bright Green signal using Fiji software, and cell area was quantified. To evaluate area expansion trends, a linear approximation was fitted to each time series using GraphPad Prism, and the slope of the fitted line was used as an indicator of the expansion rate. Developmental time was defined as the number of hours elapsed since E0.5, which corresponds to noon on the day a copulatory plug was detected in the female and was treated as a continuous variable based on the actual imaging time.

To analyze germ cell degeneration, we identified degeneration events in germ cells expressing H2B-mCherry based on chromatin condensation or fragmentation. For each developmental window between E13.5 and PD1, twelve randomly selected $100\,\mu m \times 100\,\mu m$ regions were analyzed per sample across E12.5 gonads cultured under continuous imaging ("E12.5 + 4 d imaged"), E12.5 gonads cultured without prior imaging ("E12.5 + 4 d non-imaged"), and freshly isolated E16.5 ovaries. All degeneration events within these areas were manually counted. Germ cells undergoing degeneration in immediate proximity were considered part of the same cluster, and the number of germ cells per cluster was defined as the cluster size. Cluster sizes were plotted as individual data points and visualized using box and whisker plots generated in GraphPad Prism. Also, degenerated germ cells were categorized based on cluster size into four groups (single cells, 2-cell clusters, 3-cell clusters, and >4-cell clusters), and the cumulative number of germ cells in each category was calculated for each developmental window.

## Nuclear rotation assay

To assess nuclear rotation, wild-type E12.5 gonads were cultured ex vivo for four days and subsequently stained with Hoechst 33342 to visualize chromatin. Time-lapse imaging was performed at 10-s intervals using the Zeiss LSM880 confocal microscope, as described above. In each germ cell nucleus, specific foci within Hoechst-dense regions, presumed to correspond to heterochromatin, were manually selected as reference points. The movement of these foci was tracked across all time points using Imaris software, and their displacement and velocity were quantified at each time point. The resulting values were plotted over time.

## Blebbing assay

To assess membrane blebbing dynamics in germ cells, female gonads from Stella-ECFP embryos at E11.5–E12.5 were cultured ex vivo and incubated with PlasMem Bright Red (Dojindo Laboratories) at a 1:2000 dilution for ~8 h prior to imaging. Where indicated, gonads were also stained with SiR-Actin (Cytoskeleton Inc.) at a final concentration of $0.5\,\mu M$ to visualize filamentous actin. Live imaging was performed at 2- or 3-second intervals using LSM780 or LSM880 confocal microscopes, as described above. Blebs were defined as membrane protrusions exceeding $2\,\mu m$ in height, and the number of blebs was manually counted from the live imaging data.

For temporal projection analysis, membrane outlines were traced across seven consecutive time points (200 s total) using Adobe Illustrator (Adobe Inc.). To investigate the role of actin in blebbing, the bleb arc length was measured during both the expansion and contraction phases. This measurement was performed using Fiji software by manually tracing the arc at 3-s intervals. The distance between the arc ends was measured, along with the mean intensity of PlasMem Bright Red and SiR-Actin, which were plotted over time.

Pharmacological reagents were added to the culture medium either at the onset of culture or 16 h prior to imaging, depending on the experimental design. The reagents used included LDN193189 (MedchemExpress, HY-12071A; 500 nM), BMS493 (Merck, B6688; 10 μM), Jasplakinolide (AdipoGen, AG-CN2-0037; 10 nM-1 μM), Cytochalasin D (Merck, C8273; 31.3-500 nM), Ciliobrevin D (Merck, 250401; 6.3-100 μM), CK666 (Tocris Bioscience, 3950; 10-250 μM), SMIFH2 (Tocris Bioscience, 4401; 10-250 μM), Blebbistatin (FUJIFILM, 021-17041; 1-25 μM), and H1152 (SYN-kinase, 1221; 12-300 nM). Following treatment, blebbing frequency was quantified as described above.

## Whole-mount immunostaining and tissue clearing

Whole-mount immunostaining and optical clearing of fetal gonads or ovaries were conducted based on the SeeDB2 protocol, with minor modifications (Ke et al, 2016). Tissues were fixed in 4% paraformaldehyde (PFA) in PBS for 2 h at 4 °C, except for SYCP3 and MVH co-immunostaining, for which fixation was performed for 30 min at 4 °C. Samples were then washed three times with PBS. Permeabilization was performed overnight at 4 °C in PBS containing 2% saponin. Samples were then incubated in blocking buffer (FP1012, PerkinElmer) for 3 h at 4 °C. Primary antibodies were diluted in the blocking buffer and incubated with the tissues overnight at 4 °C. After three washes with washing buffer (PBS containing 2% saponin and 0.5% Triton X-100), secondary antibodies diluted in the blocking buffer were applied and incubated overnight at 4 °C.

For tissue clearing, samples were sequentially incubated in a 1:2 mixture of Omnipaque 350 (GE HealthCare) and Milli-Q water containing 2% saponin (solution 1) for 4–6 h, followed by a 1:1 mixture (solution 2) for 6-12 h. Clearing was completed by incubation in 100% Omnipaque 350 with 2% saponin and DAPI (0.1 μg/ml) for counterstaining (solution 3), for at least 12 h. Cleared tissues were mounted in solution 3 on glass-bottom dishes. Images were acquired using a Zeiss LSM880 confocal microscope and processed with ZEN lite software (Zeiss).

The primary antibodies and their dilutions were anti-CETN2 (15877-1-AP, Proteintech; 1:400), anti-MVH (ab27591, Abcam; 1:400), anti-STRA8 (provided by Dr. Kei-ichiro Ishiguro; 1:1000) (Shimada et al, 2023), anti-SCP3 (ab15093, Abcam; 1:400), and anti-TEX14 (18351-1-AP, Proteintech; 1:500). The secondary antibodies and their dilutions were Alexa Fluor 488 goat anti-mouse IgG (A11029, Thermo Fisher Scientific; 1:500), Alexa Fluor 555 goat anti-mouse IgG (A21424, Thermo Fisher Scientific; 1:500), Alexa Fluor 568 goat anti-rabbit IgG (A11036, Thermo Fisher Scientific; 1:500), Alexa Fluor 647 donkey anti-mouse IgG (A31571, Thermo Fisher Scientific; 1:500), and Alexa Fluor 647 donkey anti-rabbit IgG (A31573, Thermo Fisher Scientific; 1:500).

## Mitochondria transfer assessment

To evaluate mitochondrial transfer between germ cells, we used Mito-Dendra2 mice expressing a photoconvertible mitochondrial reporter. Fetal gonads or ovaries were cultured ex vivo and stained with PlasMem Bright Green to label membranes. A single germ cell was photoconverted using a 405-nm laser to shift Mito-Dendra2 fluorescence from green to red, followed by 3D time-lapse imaging at 5-min intervals using a Zeiss LSM900 confocal microscope, as described above. Photoconverted cells were tracked using the red Mito-Dendra2 signal. For each time point, a single Z-plane showing the maximum area of the targeted cell was selected, and cell boundaries were manually delineated using Fiji software. Mean red fluorescence intensity was quantified within both the photoconverted cell and neighboring, non-photoconverted germ cells. These intensity values were plotted over developmental time. Developmental time was determined based on the time elapsed from E0.5, defined as noon on the day a copulatory plug was detected in the mother, and was treated as a continuous variable based on the actual time of observation. To assess trends in red signal intensity in adjacent cells, an approximate line was fitted to each time series using Graphpad Prism, and its slope ($m$) was evaluated as an indicator of potential mitochondrial transfer. Slopes were categorized into two groups: a positive slope ($m \geq 0$), indicating a mitochondrial transfer, and a negative slope ($m < 0$), indicating no transfer.

## Fluorescence recovery after photobleaching (FRAP)

Gonads were collected from E12.5 fetuses generated by crossing C57BL/6NCrSlc mice with EGFP-CETN2 transgenic mice. After seven days of ex vivo culture (E12.5 + 7 d), germ cells exhibiting EGFP-CETN2 signals were selected for FRAP analysis. Photobleaching was performed by exposing a defined region of EGFP-CETN2 signal to a 488 nm laser at 4% output for 15 s. A circular region of interest (ROI) with a diameter of 2 μm was placed over the bleached area to measure fluorescence recovery. The mean EGFP-CETN2 fluorescence intensity within the ROI was quantified every 5 min for a total of 120 min using Fiji software. Fluorescence intensity values were normalized to a baseline value of 1, corresponding to the mean pre-bleach intensity.

## Centrosome transfer assessment

To assess centrosome transfer between germ cells, we utilized mKikGR-CETN2 mice expressing a photoconvertible centrin reporter. Fetal gonads or ovaries were cultured ex vivo and stained with PlasMem Bright Green to visualize cell membranes. A centrin signal within a single germ cell was photoconverted from green to red using a 405-nm laser on Zeiss LSM780 or LSM880 confocal microscopes, as described above. Three-dimensional time-lapse imaging was performed at 10-min intervals with 1.5 μm z-sections for up to 30 h. For 3D reconstruction, time-lapse image stacks of the photoconverted germ cell were processed using Imaris software. The cell border was manually delineated in each z-section based on the PlasMem Bright Green signal. Three-dimensional reconstructions of the cell surface were generated at selected time intervals between 10 and 80 min. Red mKikGR-CETN2 signals were detected at every time point and visualized using the Dragon Tail function in Imaris to represent the temporal trajectory of photoconverted centrosomes. To evaluate potential centrosome transfer, Red mKikGR-CETN2 foci were manually counted in both the photoconverted germ cell and a neighboring, non-photoconverted germ cell at each time point using Imaris software and aligned to developmental time. Developmental time was determined based on the time elapsed from E0.5, defined as noon on the day a copulatory plug was detected in the mother, and was treated as a continuous variable based on the actual time of observation.

## Statistical analysis

Statistical analyses were performed using GraphPad Prism 9. A two-tailed, unpaired Welch's *t* test was used to assess differences between groups. Sample sizes (*N*) and *P* values are provided in the corresponding figures and figure legends. Error bars in figures indicate mean ± standard deviation (SD). No statistical methods were used to predetermine sample sizes. The threshold for statistical significance was set at $P < 0.05$.

## Data availability

The source image datasets underlying the figures and movies are available at ssbd-repos-000464, https://doi.org/10.24631/ssbd.repos.2025.08.464 in SSBD:repository.

The source data of this paper are collected in the following database record: biostudies:S-SCDT-10_1038-S44318-026-00780-6.

## Peer review information

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

## Acknowledgements

We are grateful to Dr. Willy Baarends (Erasmus MC) for providing the mCherry-SYCP3 mouse line and to Dr. Mitinori Saitou (Kyoto University) for the Stella-ECFP mouse line. We thank the Kobe Imaging and Information Analysis Facility (KbiIF) at RIKEN BDR for technical support with imaging, and Ms. Shiori Ori (RIKEN BDR) for assistance with image analysis. We are grateful to Dr. Ryuki Shimada and Dr. Kei-ichiro Ishiguro (Kumamoto University) for providing the STRA8 antibody and for their insightful comments and suggestions. We also acknowledge Dr. Katsuhiko Hayashi (Osaka University) and Dr. Genshiro Sunagawa (RIKEN BDR) for valuable discussions and suggestions. This work was supported by RIKEN Pioneering Project "Long-

Timescale Molecular Chronobiology"; JSPS Overseas Research Fellowship (202460115); JSPS KAKENHI Grant (23H04948, 21H02407, 25H00981); and the Naito Foundation Research Grant.

## Author contributions

**Eishi Aizawa**: Conceptualization; Resources; Formal analysis; Funding acquisition; Validation; Investigation; Visualization; Methodology; Writing—original draft; Writing—review and editing. **So Shimamoto**: Formal analysis; Validation; Investigation; Methodology; Writing—review and editing. **Eriko Kajikawa**: Investigation; Methodology. **Junko Hara**: Resources; Methodology. **Takaya Abe**: Resources; Methodology. **Hiroki Shibuya**: Resources; Methodology. **Tomoya S Kitajima**: Conceptualization; Resources; Supervision; Funding acquisition; Writing—original draft; Project administration; Writing—review and editing.

Source data underlying figure panels in this paper may have individual authorship assigned. Where available, figure panel/source data authorship is listed in the following database record: biostudies:S-SCDT-10_1038-S44318-026-00780-6.

## Disclosure and competing interests statement

The authors declare no competing interests.

# Expanded View Figures

**Figure EV1. Three-dimensional reconstruction and germ cell identification during ex vivo culture.**

(**A**) Three-dimensional reconstructed images of cells in E12.5 gonads expressing Stella-ECFP during an 8-day ex vivo culture. Gonads were stained with Hoechst 33342 and PlasMem Bright Red. Germ cell membranes (green), germ cell nuclei (magenta), somatic cell membranes (cyan), and somatic cell nuclei (yellow) are shown. Confocal z-sections were acquired at 1 μm intervals, and 3D images were reconstructed orthogonally. Identical images of E12.5 + 2 d, E12.5 + 5 d, and E12.5 + 8 d gonads are shown in Fig. 2B. Scale bar, 10 μm. (**B**) Development of germ cells in E12.5 gonads expressing Stella-ECFP during an 8-day ex vivo culture. Gonads were stained with PlasMem Bright Red. The merged image shows Stella-ECFP expression (green) alongside the PlasMem signal (magenta). Stella-ECFP expression weakened around day 5 of culture but strengthened again by days 7 and 8. Asterisks indicate cells with weakened Stella-ECFP signals. Scale bar, 10 μm.

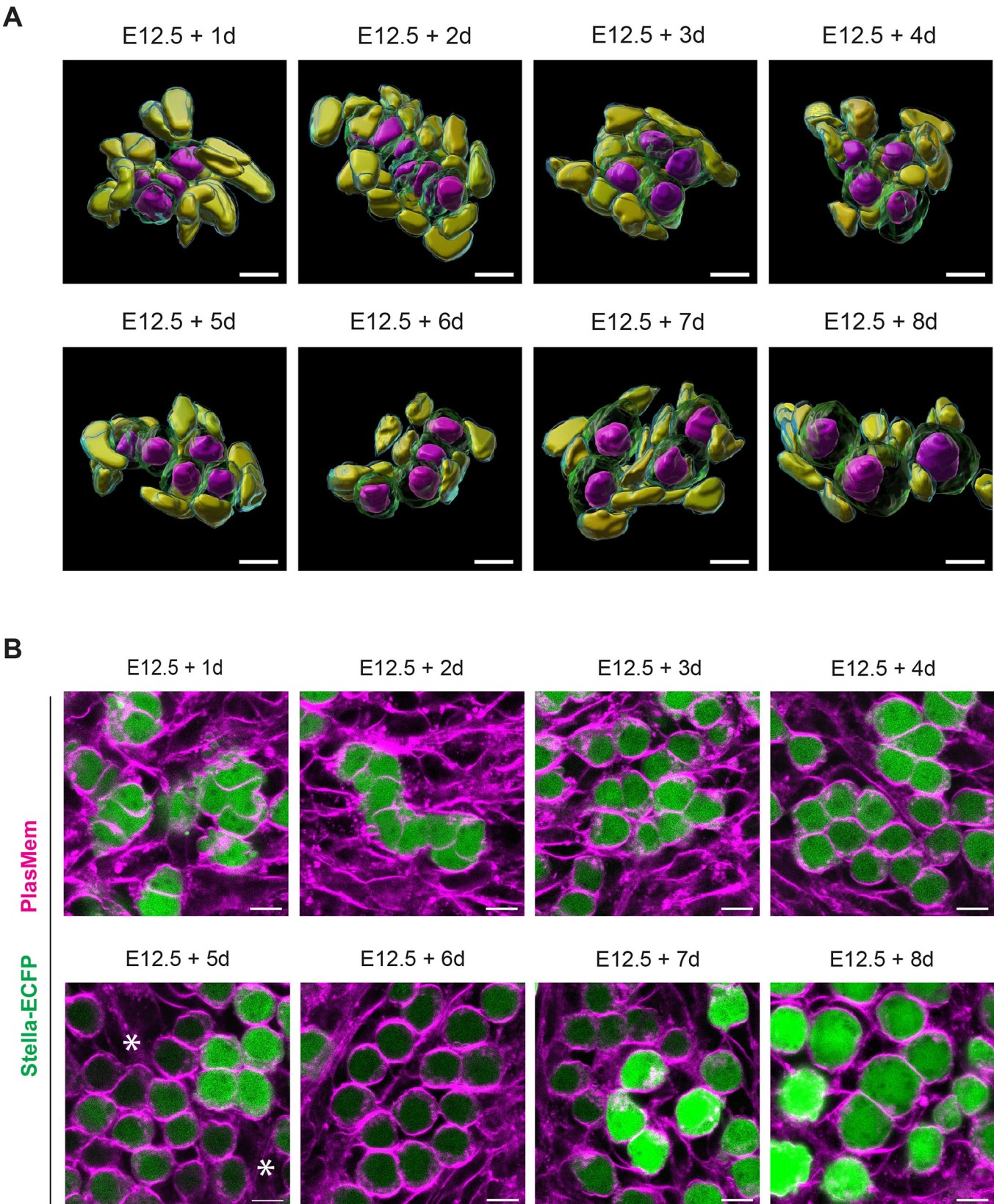

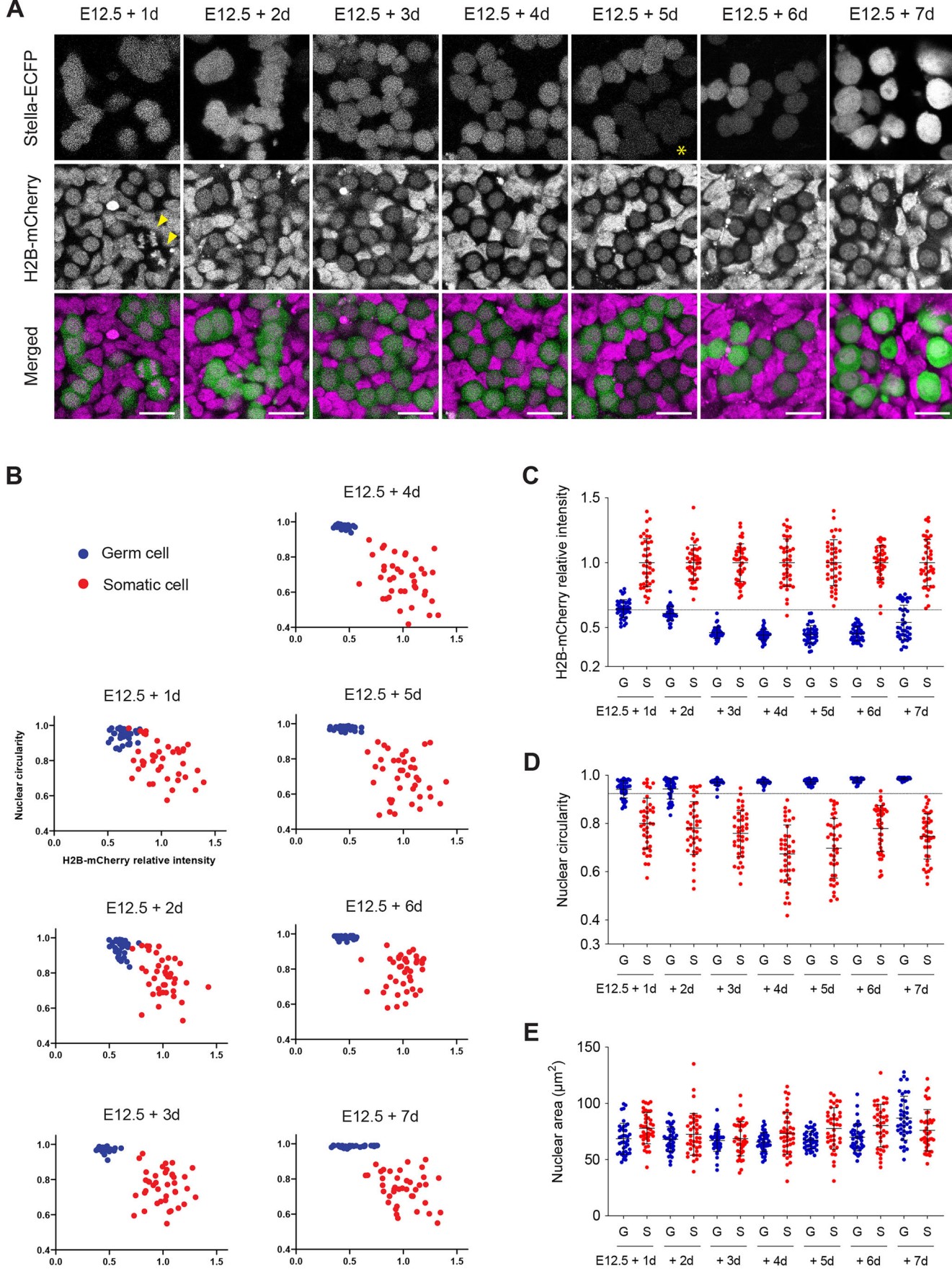

◄ **Figure EV2. Validation of germ and somatic cell classification using nuclear features.**

(**A**) Development of germ cells in E12.5 gonads co-expressing Stella-ECFP and H2B-mCherry during a 7-day ex vivo culture. The merged image (bottom) shows Stella-ECFP (green) and H2B-mCherry (magenta) expression. Stella-ECFP signals weakened around day 5 of culture but strengthened by day 7. An asterisk indicates cells with weakened Stella-ECFP signals. Arrowheads mark Stella-ECFP-positive germ cells undergoing presumed anaphase chromosome segregation. Scale bar, 20 μm. (**B**) Quantitative analysis of histone intensity and nuclear circularity in cells from E12.5 gonads co-expressing Stella-ECFP and H2B-mCherry during the 7-day ex vivo culture. From E12.5 + 3 d to E12.5 + 7 d, two distinct populations were identified: germ cells with low H2B-mCherry intensity and high nuclear circularity, and somatic cells with high H2B-mCherry intensity and low nuclear circularity. The mean H2B-mCherry intensity of somatic cells was normalized to a relative intensity of 1. (**C–E**) Quantitative analysis of histone intensity (**C**), nuclear circularity (**D**) and nuclear area (**E**) in cells from E12.5 gonads co-expressing Stella-ECFP and H2B-mCherry during the 7-day ex vivo culture. Bars represent mean values ± standard deviations. (**C**) The mean H2B-mCherry intensity of somatic cells was normalized to a relative intensity of 1. A dashed line indicates a H2B-mCherry relative intensity threshold of 0.63, distinguishing somatic cells (> 0.63; 98.8% (158/160)) and germ cells (≤ 0.63; 100% (160/160)) between E12.5 + 3 d and E12.5 + 6 d. (**D**) A dashed line indicates a nuclear circularity threshold of 0.93, distinguishing somatic cells (> 0.93; 99.0% (198/200)) and germ cells (≤ 0.93; 99.5% (199/200)) between E12.5 + 3 d and E12.5 + 7 d. (**B–E**) Germ cells (blue) and somatic cells (red) were distinguished based on Stella-ECFP expression intensity. Quantification included 40 germ cells and 40 somatic cells at each developmental time point. G germ cell, S somatic cell. Source data are available online for this figure.

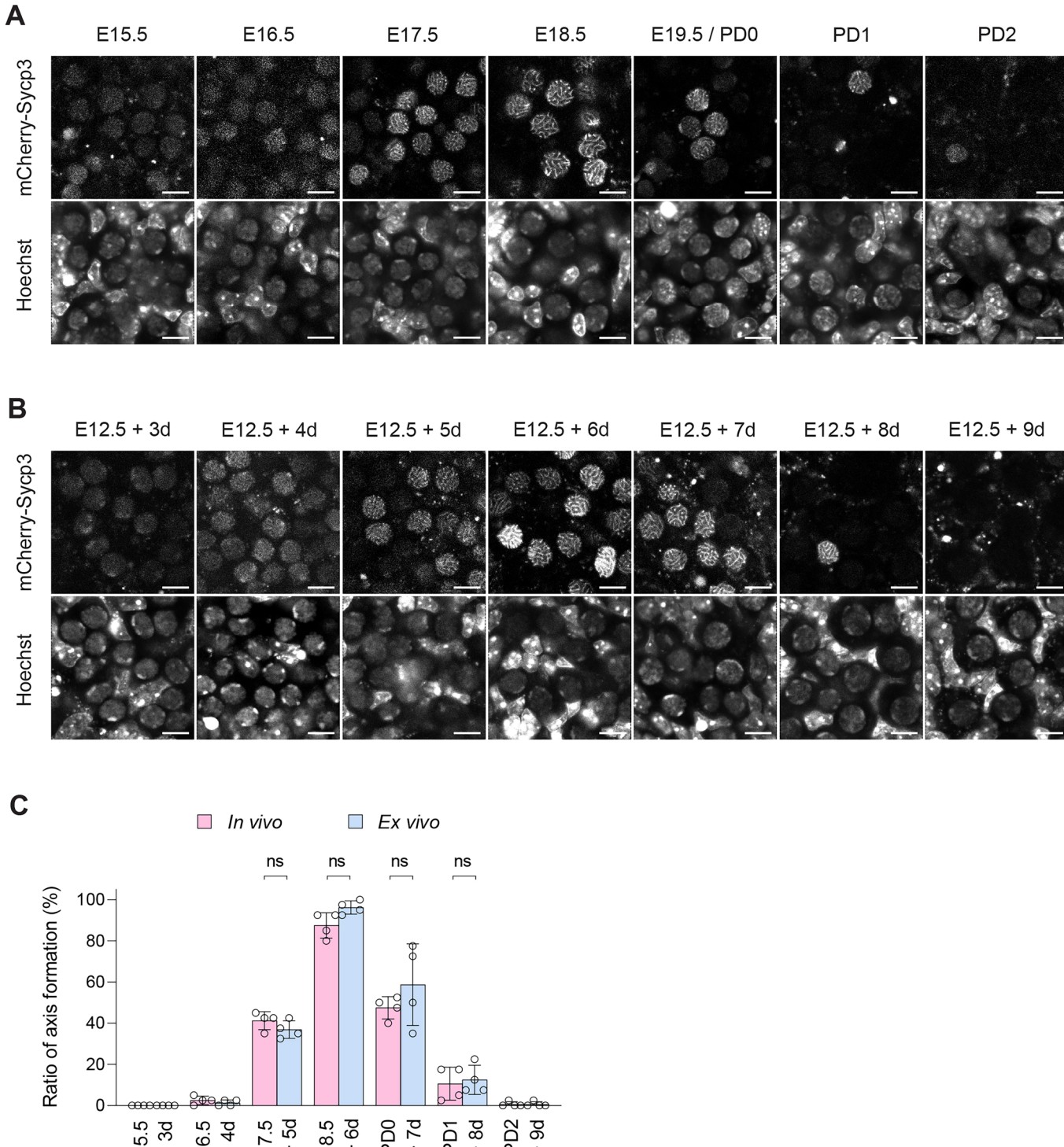

◀ **Figure EV3. Comparison of chromosome axis formation in germ cells during in vivo and ex vivo development.**

(A) Representative images of germ cells in ovaries collected from E15.5 fetuses to PD2 neonates. Ovaries expressing mCherry-SYCP3 were stained with Hoechst 33342 prior to imaging. Chromosome axis formation was prominently observed between E17.5 and E19.5/PD0. Scale bar, 10 μm. (B) Representative images of germ cells in E12.5 gonads during a 9-day ex vivo culture. E12.5 gonads expressing mCherry-SYCP3 were stained with Hoechst 33342 prior to imaging. Chromosome axis formation was prominently observed between E12.5 + 5 d and E12.5 + 7 d. Scale bar, 10 μm. (C) Quantitative analysis of chromosome axis formation ratios in germ cells from ovaries and E12.5 gonads cultured ex vivo, both expressing mCherry-SYCP3. Germ cells with or without distinct axis formation were evaluated based on mCherry-SYCP3 signals from confocal z-section images. Bars represent mean values ± standard deviations. $N = 4$ gonads/ovaries; 40 germ cells per gonad/ovary. Statistical analysis was performed using a $t$ test with Welch's correction. ns non-significant. Source data are available online for this figure.

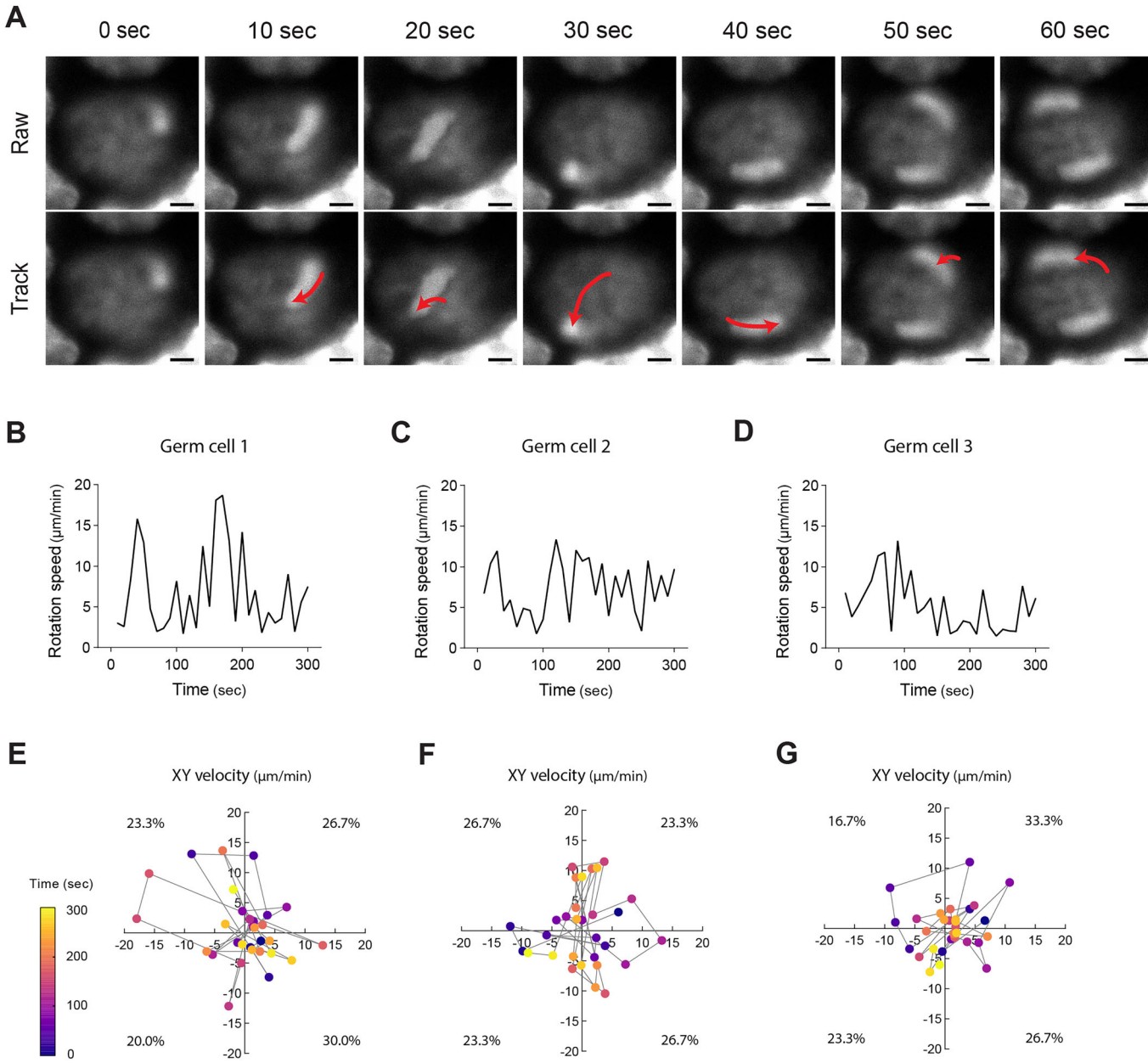

**Figure EV4. Characterization of nuclear rotation dynamics in germ cells.**

(A) Representative live imaging of a germ cell nucleus in an E12.5 + 4 d gonad cultured ex vivo. The gonad was stained with Hoechst 33342 and imaged at 10-second intervals. Raw and tracking images are shown, with red lines indicating trajectories of nuclear rotation. Scale bar, 2 μm. See also Movie EV3. (B–D) Time-lapse quantification of nuclear rotation speed for three representative germ cells. Rotation speed was measured every 10 s over a 300-second period. (E–G) Plots of X velocity (x axis) versus Y velocity (y axis) recorded at 10-s intervals for a 300-second sequence. Each plot corresponds to the germ cells analyzed in (B–D), respectively. The number within each quadrant indicates the proportion of XY velocity plots among the 30 measurements. Source data are available online for this figure.

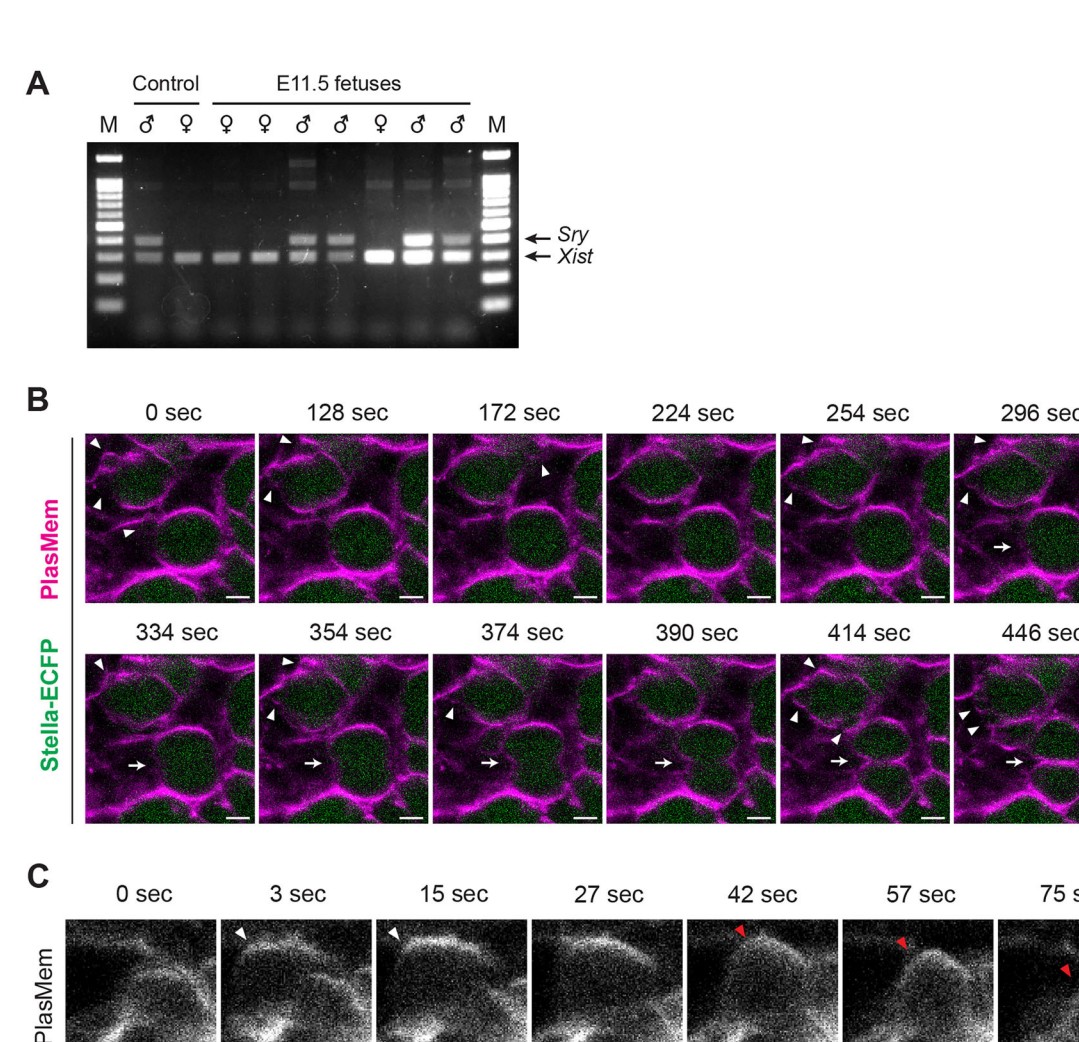

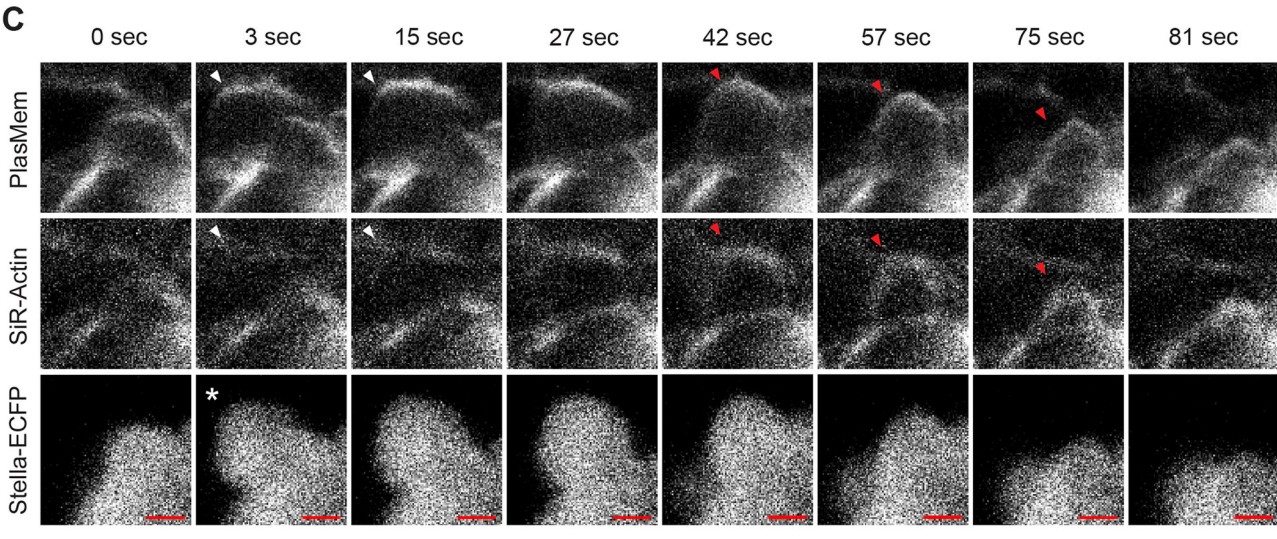

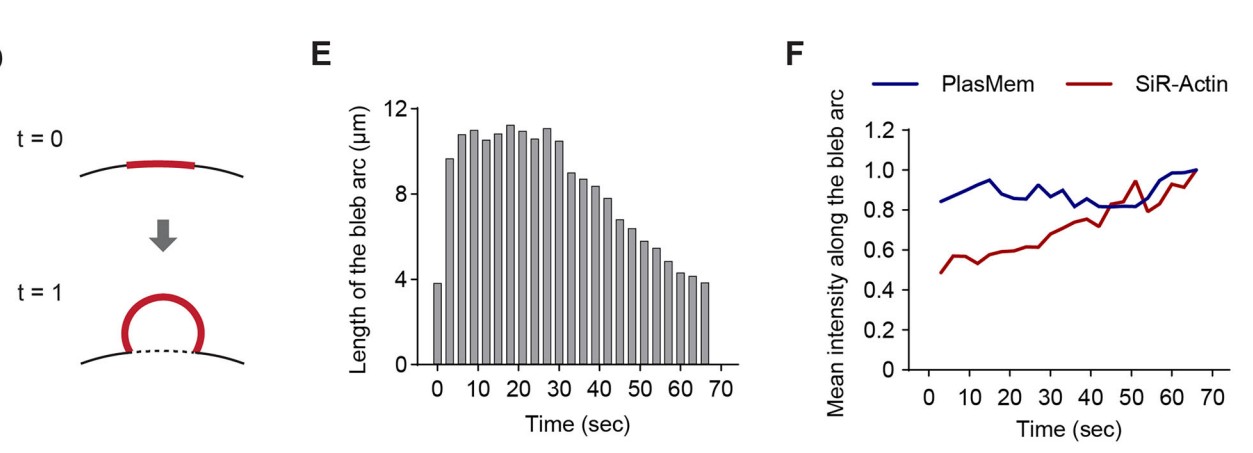

◀

**Figure EV5. Validation of germ cell blebbing and analysis of actin dynamics.**

(A) PCR analysis of E11.5 fetuses using *Sry* and *Xist* primers to determine fetal sex. M marker. (B) Live imaging of germ cells in an E12.5 gonad expressing Stella-ECFP after 2 days of ex vivo culture. The gonad was stained with PlasMem Bright Red and imaged every 2 s. Stella-ECFP (green) and PlasMem (red) signals are shown as merged images. Arrowheads mark blebs, and arrows indicate the position of cytokinesis. Scale bar, 5 μm. See also Movie EV5. (C) Representative live-imaging of blebbing in an E12.5 gonad expressing Stella-ECFP after 1 day of ex vivo culture. The gonad was stained with PlasMem Bright Red and SiR-Actin. Images were captured every 3 s. An asterisk marks an emerging bleb. White arrowheads indicate cell membranes with weak SiR-Actin signals, while red arrowheads denote membranes with distinct SiR-Actin signals. Scale bar, 2 μm. See also Movie EV6. (D–F) Quantitative analysis of the blebbing shown in (C). (D) Schematic illustration of the measurement. The red line indicates the bleb arc, representing the measured linear region of interest. (E) Quantification of the bleb arc length over time. The arc length decreases after approximately 30 s. (F) Mean intensities of PlasMem Bright Red (blue) and SiR-Actin (red) along the bleb arc over time. PlasMem intensity remained stable, while SiR-Actin intensity increased as the bleb shrank. Source data are available online for this figure.

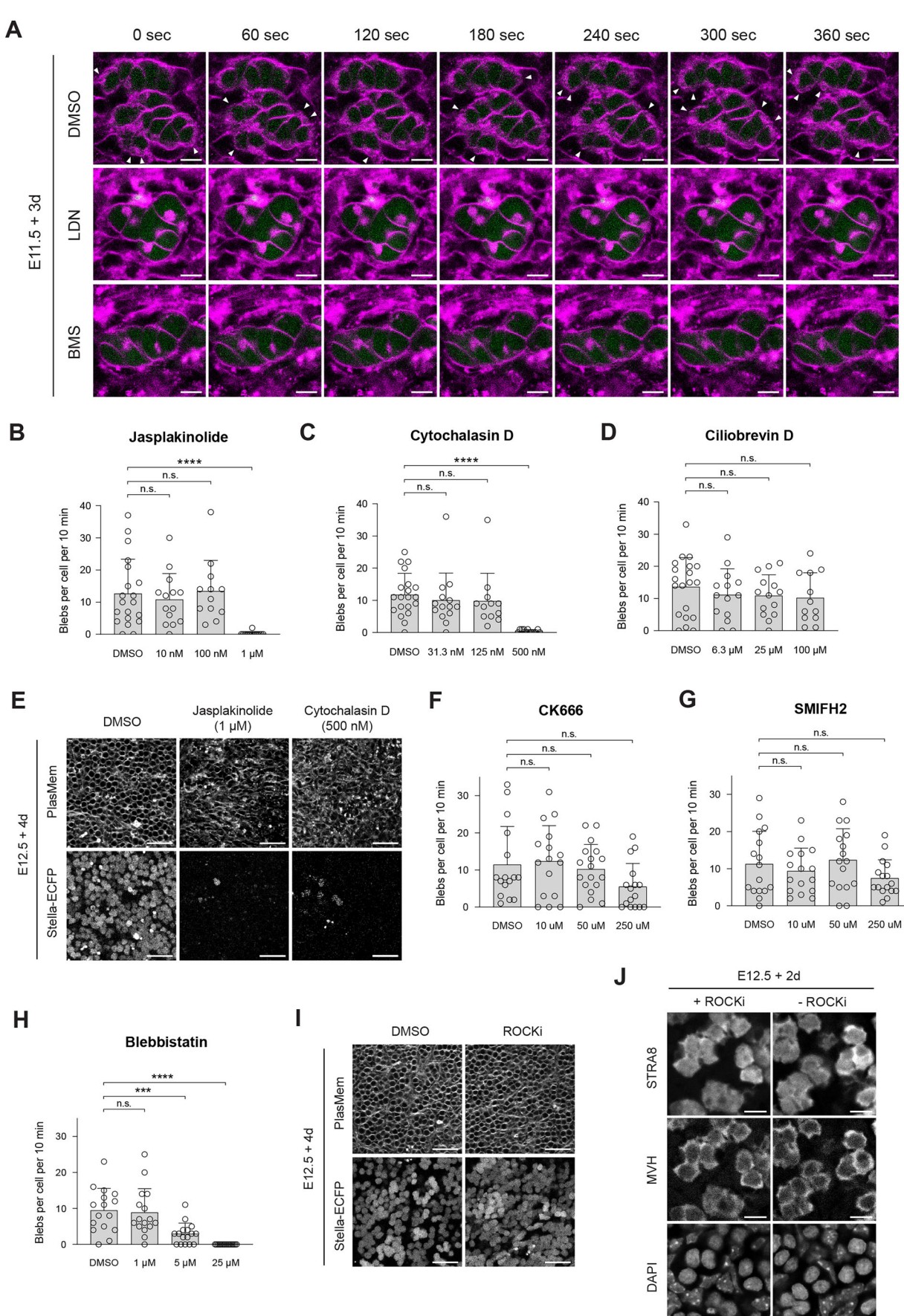

**Figure EV6.   Analysis of signaling pathways regulating blebbing in germ cells.**

(A) Representative live imaging of E11.5 + 3 d gonads under inhibition of meiotic initiation signals. Female E11.5 gonads were cultured for 3 consecutive days with DMSO, LDN193189 (500 nM) or BMS493 (10 μM) treatments, followed by staining with PlasMem Bright Red. Arrowheads indicate blebs. Scale bar, 10 μm. See also Movie EV7. (B–D) Blebbing frequency following treatment with Jasplakinolide (B), Cytochalasin D (C), or Ciliobrevin D (D). The number of blebs was counted from live imaging of E12.5 + 2 d gonads expressing Stella-ECFP. Each inhibitor was supplemented 16 h prior to imaging. Bars represent mean values + standard deviations. Statistical analysis was performed using a $t$ test with Welch's correction. Sample sizes: (B) DMSO ($n = 20$), 10 nM ($n = 14$), 100 nM ($n = 13$), 1 μM ($n = 14$); (C) DMSO ($n = 20$), 31.3 nM ($n = 14$), 125 nM ($n = 12$), 500 nM ($n = 12$); (D) DMSO ($n = 20$), 6.3 μM ($n = 14$), 25 μM ($n = 14$), 100 μM ($n = 12$). ns, non-significant; ****$P = 0.000048$ (Jasplakinolide); ****$P = 0.00000029$ (Cytochalasin D). (E) Representative images of cells in E12.5 + 4 d gonads expressing Stella-ECFP. Gonads were cultured with DMSO, Jasplakinolide (1 μM), or Cytochalasin D (500 nM) for 16 h between E12.5 + 1 d and E12.5 + 2 d, followed by staining with PlasMem Bright Red. Stella-ECFP-positive cells were abundant in samples treated with DMSO but were scarcely observed following treatment with Jasplakinolide or Cytochalasin D. Scale bar, 50 μm. (F–H) Blebbing frequency following treatment CK666 (F), SMIFH2 (G), or Blebbistatin (H). The number of blebs was counted from live imaging of E12.5 + 2 d gonads expressing Stella-ECFP. Each inhibitor was supplemented 16 h prior to imaging. Bars represent mean values + standard deviations. Statistical analysis was performed using a $t$ test with Welch's correction. Sample sizes: (F) DMSO ($n = 16$), 10 μM ($n = 16$), 50 μM ($n = 18$), 250 μM ($n = 16$); (G) DMSO ($n = 16$), 10 μM ($n = 16$), 50 μM ($n = 16$), 250 μM ($n = 16$); (H) DMSO ($n = 16$), 1 μM ($n = 16$), 5 μM ($n = 16$), 25 μM ($n = 16$). ns, non-significant; ***$P = 0.00092$; ****$P = 0.000017$. (I) Representative images of cells in E12.5 + 4 d gonads expressing Stella-ECFP. Gonads were cultured with either DMSO or ROCKi (12 nM) for 16 h between E12.5 + 1 d and E12.5 + 2 d, followed by staining with PlasMem Bright Red. Scale bar, 50 μm. (J) Representative immunostaining of STRA8 in cultured gonads following ROCKi treatment. E12.5 + 2 d gonads were stained with antibodies against STRA8 and MVH, followed by DAPI counterstaining. STRA8 signals were detected in germ cells irrespective of ROCKi treatment. Scale bar, 10 μm. Source data are available online for this figure.

   

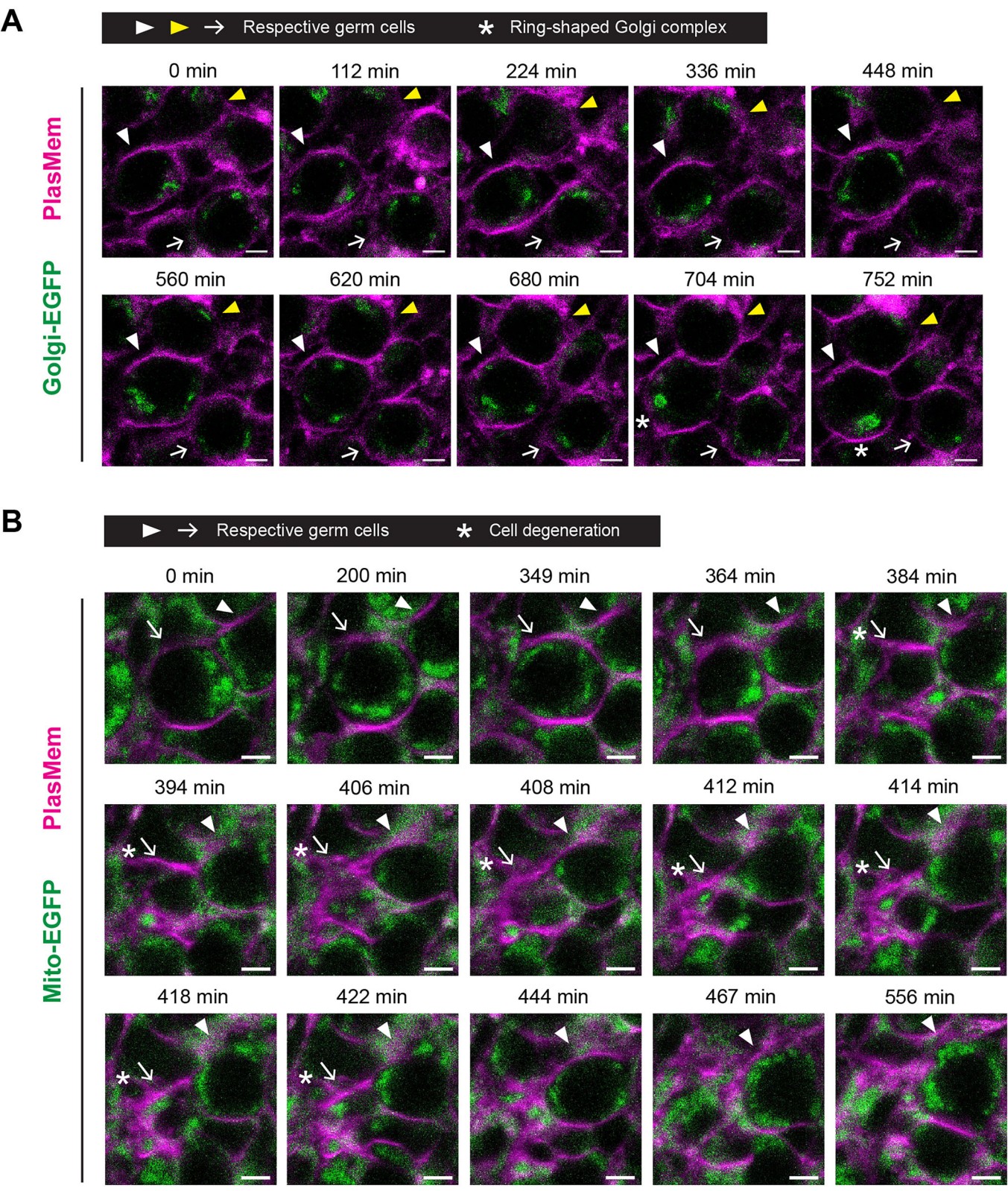

**Figure EV7. Live imaging of Golgi and mitochondrial dynamics during oocyte formation.**

(**A**) Representative live imaging of an E15.5 + 2 d ovary expressing Golgi-EGFP. The ovary was stained with PlasMem Bright Red and imaged at 4-min intervals. Golgi-EGFP (green) and PlasMem (magenta) signals are shown as merged images. Tracked germ cells are indicated by arrows, white arrowheads, and yellow arrowheads, respectively. Asterisks mark ring-shaped Golgi complexes. Scale bar, 5 μm. See also Movie EV8. (**B**) Representative live imaging of an E14.5 + 6 d ovary expressing Mito-EGFP. The ovary was stained with PlasMem Bright Red and imaged at 1-min intervals. Merged images show Mito-EGFP (green) and PlasMem (magenta) signals. Arrows and arrowheads indicate a tracked germ cell and its adjacent germ cell, respectively. Asterisks denote the degeneration of the tracked germ cell. Scale bar, 5 μm. See also Movie EV9.

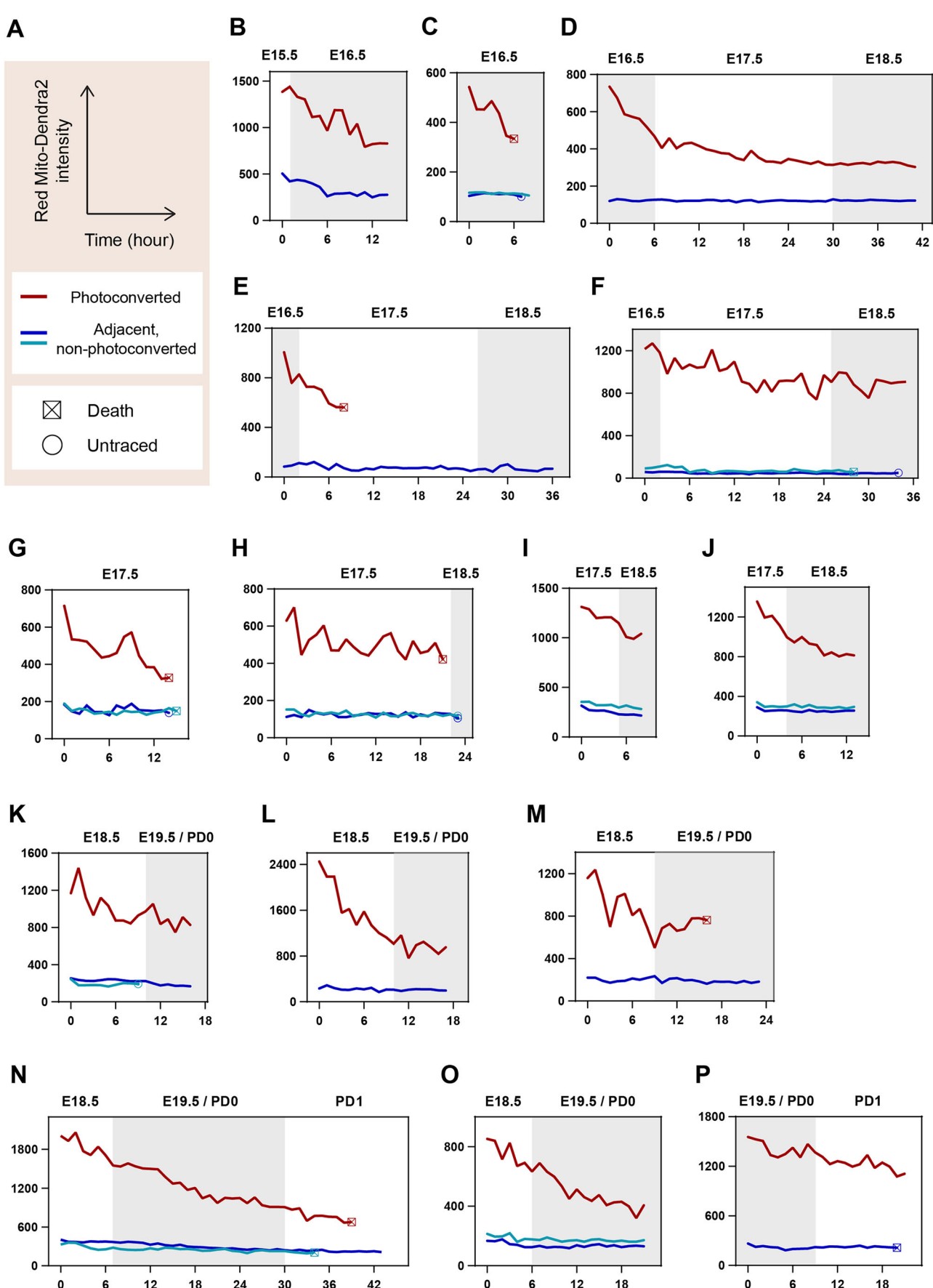

◀ **Figure EV8. Individual time-lapse quantification of photoconverted Mito-Dendra2 intensity.**

Green Mito-Dendra2 in a germ cell within cultured gonads/ovaries was photoconverted to Red Mito-Dendra2, followed by 3D tracking and imaging at 5-min intervals. Absolute mean intensities of Red Mito-Dendra2 in photoconverted germ cells, along with one or two adjacent non-photoconverted germ cells, were measured and aligned by developmental time. Ex vivo time points were converted to their corresponding in vivo developmental times, as shown at the top of each plot. (A) Axes titles and legends. (B–P) Plots of quantified Red Mito-Dendra2 intensity. The samples used for imaging include: (B) E12.5 + 3 d gonad, (C) E12.5 + 4 d gonad, (D) E12.5 + 4 d gonad, (E) E14.5 + 2 d ovary, (F) E14.5 + 2 d ovary, (G) E12.5 + 5 d gonad, (H) E12.5 + 5 d gonad, (I) E12.5 + 5 d gonad, (J) E12.5 + 5 d gonad, (K) E12.5 + 6 d gonad, (L) E12.5 + 6 d gonad, (M) E12.5 + 6 d gonad, (N) E12.5 + 6 d gonad, (O) E12.5 + 6 d gonad, (P) E12.5 + 7 d gonad. See also Fig. 5D,E. Source data are available online for this figure.

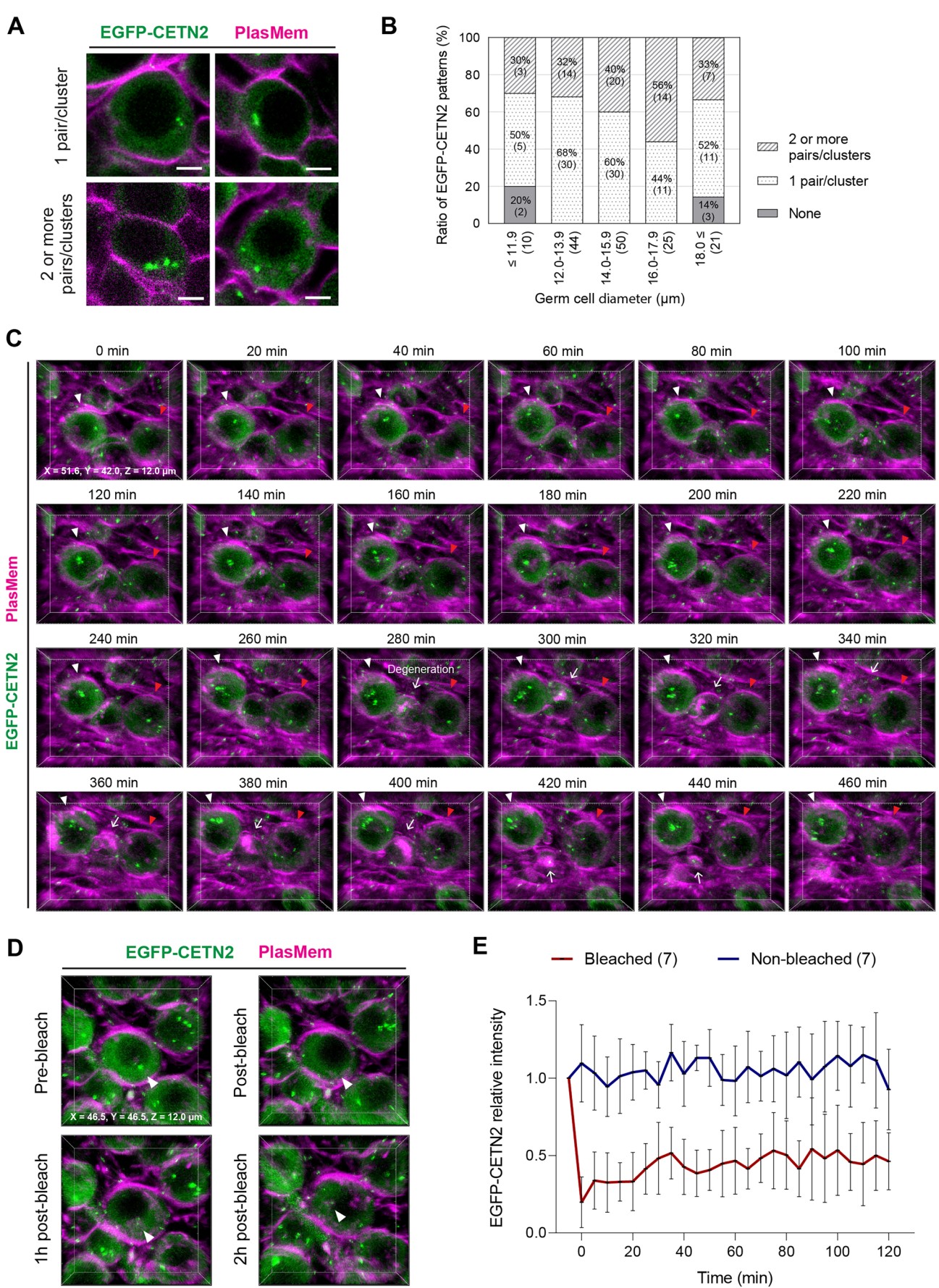

**Figure EV9. Characterization of EGFP-CETN2 dynamics during oocyte formation.**

(A) Patterns of EGFP-CETN2 signals in germ cells. Images were captured from E12.5 + 7 d (bottom left) and E12.5 + 9 d (top left, top right, and bottom right) gonads stained with PlasMem Bright Red. Merged images show EGFP-CETN2 (green) and anti-CETN2 (magenta). EGFP-CETN2 signal patterns include 1 pair (top left), 1 cluster (top right), and 2 or more pairs/clusters (bottom left and bottom right). Scale bar, 5 μm. (B) Distribution of EGFP-CETN2 patterns in germ cells by diameter. EGFP-CETN2 patterns in germ cells from E12.5 gonads cultured for 3 to 11 days were classified into three categories: None, 1 pair/cluster, and 2 or more pairs/clusters. Germ cell diameters were measured and grouped into five categories: ≤11.9 μm, 12.0– 13.9 μm, 14.0–15.9 μm, 16.0–17.9 μm, ≥18.0 μm. Numbers in brackets indicate germ cell counts. A total of 180 germ cells were analyzed, with 30 cells each from E12.5 + 3 d, + 5 d, + 7 d, + 9 d, and + 11 d gonads. See also Fig. 6A. (C) Representative live imaging of an E12.5 + 7 d gonad expressing EGFP-CETN2. The gonad was stained with PlasMem Bright Red and subjected to 3D time-lapse imaging every 10 min with z-sections at 1.5 μm intervals. Merged 3D images show EGFP-CETN2 (green) and PlasMem (magenta) signals displayed every 20 min in perspective. White and red arrowheads indicate tracked germ cells, respectively, while arrows denote germ cell degeneration. See also Movie EV11. (D, E) FRAP analysis of EGFP-CETN2 in germ cells. E12.5 + 7 d gonads expressing EGFP-CETN2 were stained with PlasMem Bright Red, followed by targeted photobleaching of EGFP-CETN2 signals in germ cells. (D) Representative 3D images of EGFP-CETN2 photobleaching. Merged images show EGFP-CETN2 (green) and PlasMem (magenta) signals, with arrowheads indicating the photobleached EGFP-CETN2 focus. (E) Time-lapse analysis of EGFP-CETN2 intensity in response to photobleaching. Mean EGFP-CETN2 intensity before photobleaching was normalized to a relative intensity of 1. The measurement area was manually defined using a circle with a diameter of 2 μm. The time point directly after photobleaching was set as 0 on the x axis. Bars represent mean values ± standard deviations. $N = 7$ photobleached germ cells and 7 non-photobleached germ cells. Source data are available online for this figure.

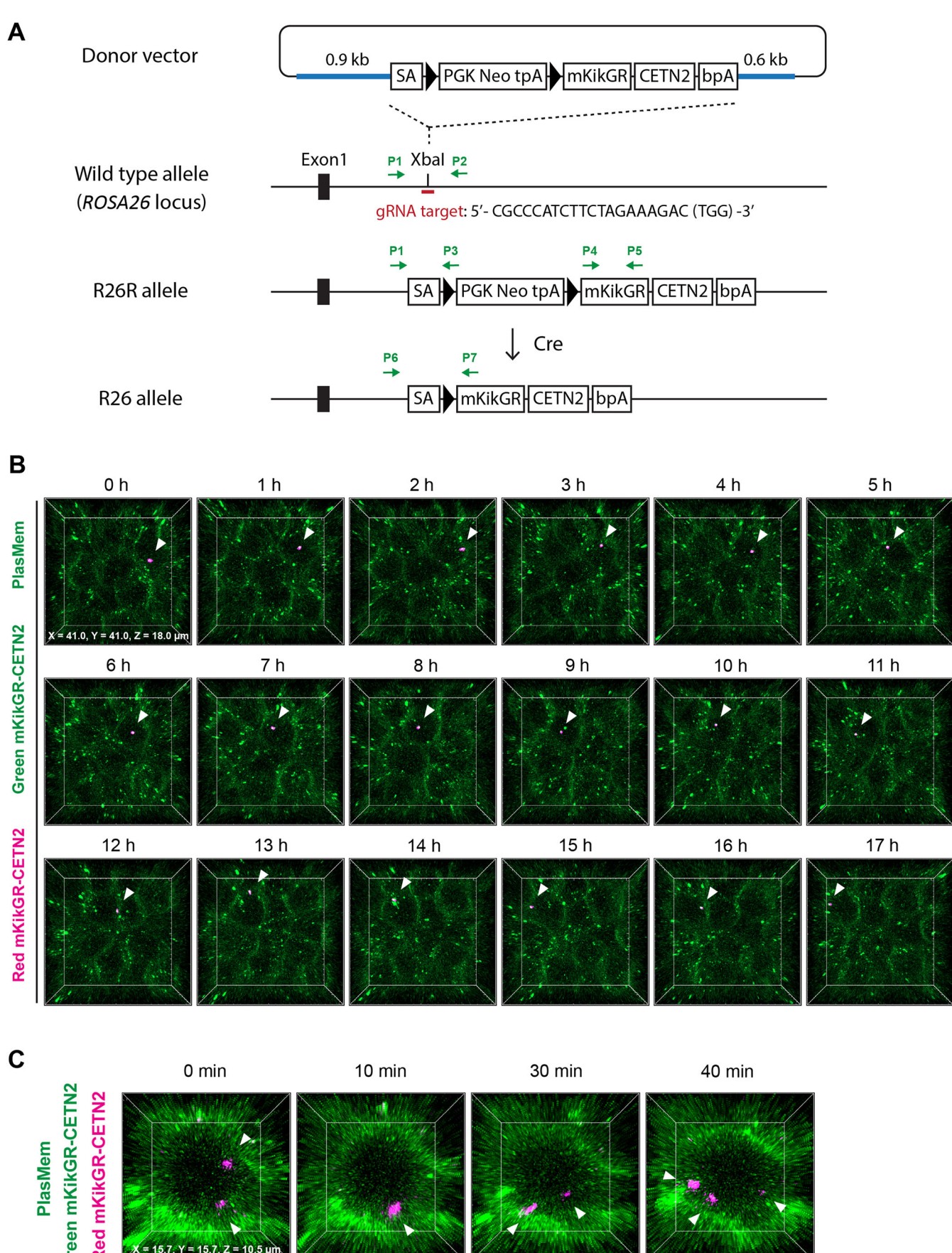

**Figure EV10. Generation of mKikGR-CETN2 mice and characterization of mKikGR-CETN2 signals.**

(A) Generation of mKikGR-CETN2 mice. The donor vector contains 5' (0.9 kb) and 3' (0.6 kb) homology arms (blue) targeting intron 1 of the ROSA26 locus, flanking an expression cassette. The vector was inserted into the genome of C57BL/6N zygotes using the CRISPR-Cas9 system with a guide RNA targeting the ROSA26 genomic locus. The resulting mice were crossed with Spo11-Cre mice to excise the stop sequences flanked by loxP sites. The black box represents exon 1 of the ROSA26 locus, and black triangles indicate loxP sequences. Green arrows indicate PCR primers. SA adenovirus splice acceptor sequence, PGK Neo neomycin resistance gene driven by the PGK1 promoter, tpA triple repeats of the SV40 polyadenylation sequence, mKikGR monomeric Kikume Green-Red, bpA bovine growth hormone polyadenylation sequence. (B) Representative time-lapse 3D Images of photoconverted Red mKikGR-CETN2 signals. An E12.5 + 5 d gonad expressing Green mKikGR-CETN2 was stained with PlasMem Bright Green, followed by photoconversion of Green mKikGR-CETN2 to Red mKikGR-CETN2. 3D time-lapse images of the photoconverted and neighboring germ cells were captured every 10 min with z-sections at 1.5 µm intervals. In the merged images, PlasMem and non-photoconverted Green mKikGR-CETN2 are shown in green, while photoconverted Red mKikGR-CETN2 is shown in magenta. Arrowheads indicate photoconverted Red mKikGR-CETN2 signals. See also Fig. 6F and Movie EV12. (C) Representative images of changes in mKikGR-CETN2 focal count. An E12.5 + 5 d gonad expressing mKikGR-CETN2 was stained with PlasMem Bright Green, followed by photoconversion of Green mKikGR-CETN2 to Red mKikGR-CETN2. 3D time-lapse images were captured every 10 min with z-sections at 1.5 µm intervals. Arrowheads indicate photoconverted Red mKikGR-CETN2, with focal counts varying over time. In merged images, PlasMem and non-photoconverted Green mKikGR-CETN2 are shown in green, while photoconverted Red mKikGR-CETN2 is shown in magenta.

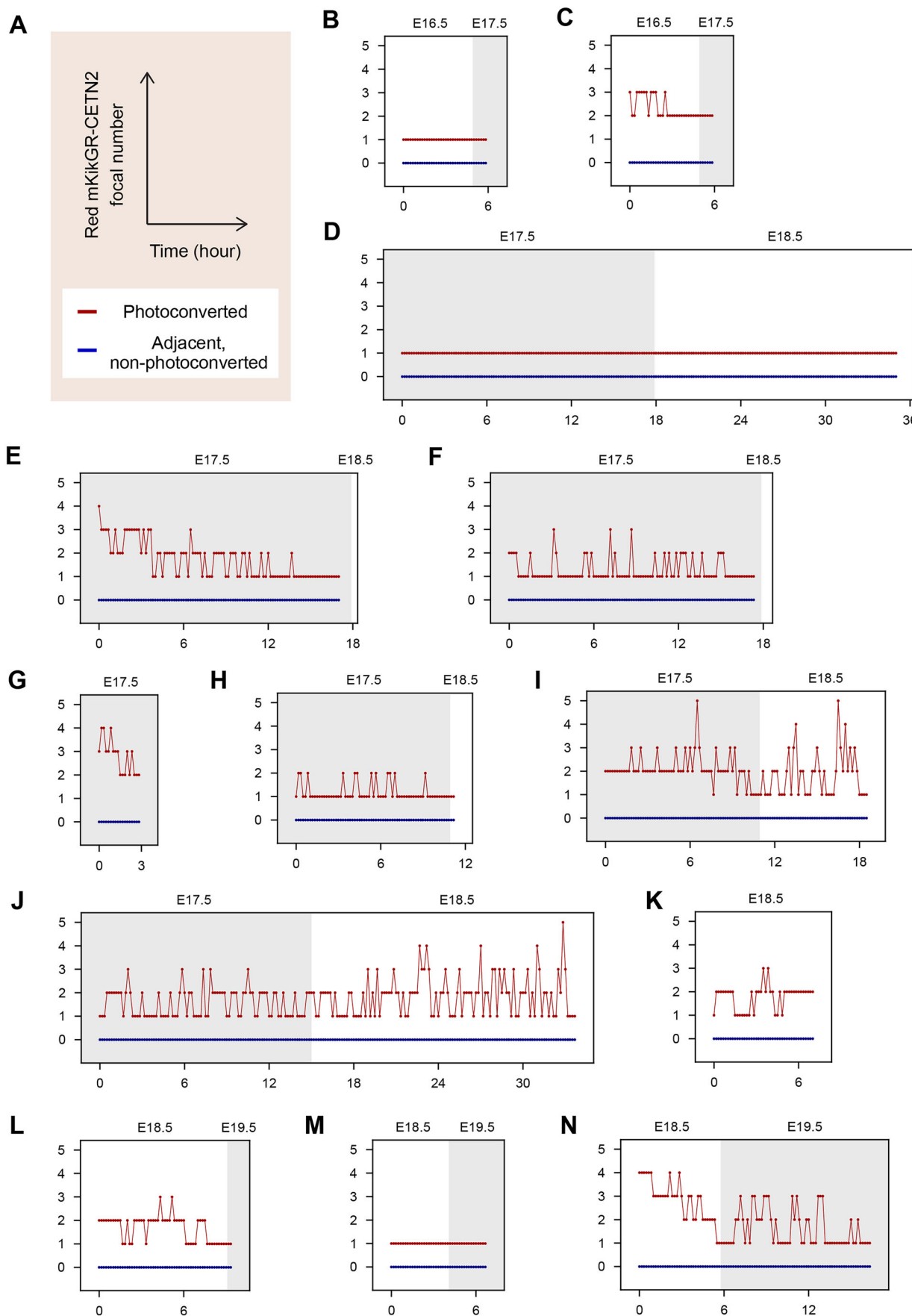

**Figure EV11.  Individual time-lapse counts of photoconverted mKikGR-CETN2 foci.**

Green mKikGR-CETN2 in a germ cell within cultured gonads was photoconverted to Red mKikGR-CETN2, followed by 3D imaging at 10-min intervals. The number of Red mKikGR-CETN2 foci in photoconverted germ cells and adjacent non-photoconverted germ cells was manually counted at each time point and aligned by developmental time. Ex vivo time points were converted to corresponding in vivo developmental times, as shown at the top of each plot. (A) Axes titles and legends. (B–N) Plots showing the number of Red mKikGR-CETN2 foci. The samples used for imaging include: (B) E12.5 + 4 d gonad, (C) E12.5 + 4 d gonad, (D) E12.5 + 5 d gonad, (E) E12.5 + 5 d gonad, (F) E12.5 + 5 d gonad, (G) E12.5 + 5 d gonad, (H) E12.5 + 5 d gonad, (I) E12.5 + 5 d gonad, (J) E12.5 + 5 d gonad, (K) E12.5 + 6 d gonad, (L) E12.5 + 6 d gonad, (M) E12.5 + 6 d gonad, and (N) E12.5 + 6 d gonad. See also Fig. 6G,H. Source data are available online for this figure.

