## [Peer Review File · The EMBO Journal]

Dynamic blebbing and absence of organelle transfer during mouse oocyte formation

Eishi Aizawa, So Shimamoto, Eriko Kajikawa, Junko Hara, Takaya Abe, Hiroki Shibuya, and Tomoya Kitajima

Corresponding author(s): Tomoya Kitajima (tomoya.kitajima@riken.jp) , Eishi Aizawa (eaizawa@fas.harvard.edu)

Review Timeline:

Submission Date:	30th Jul 25
Editorial Decision:	3rd Sep 25
Revision Received:	5th Mar 26
Editorial Decision:	25th Mar 26
Revision Received:	30th Mar 26
Accepted:	30th Mar 26

Editor: Hartmut Vodermaier

Transaction Report:

Dr. Tomoya S Kitajima
RIKEN Center for Biosystems Dynamics Research
Laboratory for Chromosome Segregation
2-2-3 Minatojima-minamimachi, Chuo-ku
Kobe
650-0047
Japan

3rd Sep 2025

Re: EMBOJ-2025-122016
Ex vivo live imaging unveils the dynamics of oocyte formation in mice

Dear Tomoya,

Thank you for submitting your manuscript on ex vivo imaging of mammalian oocyte formation for our consideration. I sent it to three expert referees, who have now returned the reports that are copied below. As you will see, the referees appreciate your technical advances and express interest in your findings. However, they also raise important concerns, especially regarding the need for additional in vivo support and more decisive evidence for faithful recapitulation of in vivo physiology.

Should you be able to clarify these main criticisms and to address the various specific points raised in the reports, we would be interested in pursuing a revised manuscript further for EMBO Journal publication. However, I should remind you that our single-major-revision-round policy makes it important to comprehensively respond to each referee point at the time of resubmission; and especially in light of the substantive concerns regarding physiological relevance of the settings, I would encourage you to send me a preliminary point-by-point response and revision plan already during the early stages of your revision work, in order to clarify if and how the key issues raised in the reports may be solved. We would also be open to extension of the regular three-months revision period if needed; our 'scooping protection' (meaning that competing work appearing elsewhere in the meantime will not affect our considerations of your study) would of course remain valid also throughout such an extension.

Further information on preparing, formatting and uploading a revised manuscript can be found below and in our Guide to Authors. Thank you again for the opportunity to consider this work for The EMBO Journal, and I look forward to hearing from you soon.

With kind regards,

Hartmut

- 3) Revised manuscript text (including main tables, and figure legends for main and EV figures) has to be submitted as editable text file (e.g., .docx format). We encourage highlighting of changes (e.g., via text color) for the referees' reference.
- 4) Each main and each Expanded View (EV) figure should be uploaded as individual production-quality files (preferably in .eps, .tif, .jpg formats). For suggestions on figure preparation/layout, please refer to our Figure Preparation Guidelines: <http://bit.ly/EMBOPressFigurePreparationGuideline>
- 5) Point-by-point response letters should include the original referee comments in full together with your detailed responses to them (and to specific editor requests if applicable), and also be uploaded as editable (e.g., .docx) text files.
- 6) Please complete our Author Checklist, and make sure that information entered into the checklist is also reflected in the manuscript; the checklist will be available to readers as part of the Review Process File. A download link is found at the top of our Guide to Authors: embopress.org/page/journal/14602075/authorguide
- 7) All authors listed as (co-)corresponding need to deposit, in their respective author profiles in our submission system, a unique ORCID identifier linked to their name. Please see our Guide to Authors for detailed instructions.
- 8) Please note that supplementary information at EMBO Press has been superseded by the 'Expanded View' for inclusion of additional figures, tables, movies or datasets; with up to five EV Figures being typeset and directly accessible in the HTML version of the article. For details and guidance, please refer to: embopress.org/page/journal/14602075/authorguide#expandedview
- 9) To facilitate reproducibility and cross-laboratory adoption of methodologies, please structure the Materials & Methods section as outlined in our guide to authors, including a completed Reagents and Tools Table that can be downloaded from our author guidelines as well (<https://www.embopress.org/page/journal/14602075/authorguide#structuredmethods>).
- 10) Digital image enhancement is acceptable practice, as long as it accurately represents the original data and conforms to community standards. If a figure has been subjected to significant electronic manipulation, this must be clearly noted in the figure legend and/or the 'Materials and Methods' section. The editors reserve the right to request original versions of figures and the original images that were used to assemble the figure. Finally, we generally encourage uploading of numerical as well as gel/blot image source data; for details see: embopress.org/page/journal/14602075/authorguide#sourcedata

Revision to The EMBO Journal should be submitted online within 90 days, unless an extension has been requested and approved by the editor; please click on the link below to submit the revision online before 2nd Dec 2025:
Link Not Available

If you choose to alternatively have this study further considered by another EMBO Press publication, please use the following hyperlink to directly transfer the manuscript, optionally with inclusion of referee reports and identities:
Link Not Available

Referee #1:

The authors established an ex vivo culture system and performed long-term live imaging of mouse fetal ovaries. Using high-resolution imaging, they reported blebbing activity during the mitosis-to-meiosis transition of oogonia. They observed no detectable transfer of mitochondria or centrosomes, as organelles remained confined to individual cells. The authors concluded that alternative mechanisms for cytoplasmic enrichment in oocytes may exist, though the exact nature of these mechanisms remains unclear.

While I appreciate the authors' efforts in developing a powerful platform for real-time analysis of germ cell dynamics, I have serious reservations about whether culturing embryonic E12.5 ovaries for extended periods (7 to 36 days) in ordinary α MEM can accurately represent physiological conditions during oogonia and oocyte cyst breakdown. For sure necrosis would occur. Despite the small size of embryonic ovaries, I find it difficult to believe that such prolonged ex vivo culture maintains the organ in a truly physiological state. Consequently, many of the observed phenomena-though documented with impressive technical skill-may not reflect biological reality.

Given these concerns, I suggest that this technically advanced study may be better suited for a methods-focused journal.

Referee #2:

Aizawa et al. establishes an ex vivo culture and live-imaging system that enables continuous visualization of mouse oocyte formation from the mitosis-to-meiosis transition through early folliculogenesis. Using this approach, they uncover dynamic germ cell behaviors, including pronounced blebbing concurring with meiotic initiation and ROCK-dependent contractility, as well as the synchronous degeneration of oocytes in cysts. Contrary to the model of cytoplasmic enrichment via organelle transfer from nurse cells, photoconversion-based tracking revealed no detectable transfer of mitochondria or centrosomes between germ cells. These findings suggest the concept of nurse-to-oocyte cytoplasmic organelle donation might be specialized for species such as *Drosophila*, and instead point to alternative mechanisms in mammals, such as engulfment of degenerating cells, as contributors to oocyte formation.

Taken together, this work is timely, well-executed and a valuable contribution to the field. IF supported by additional in vivo evidence (pls see below) it could effectively resolve the debate about nurse cell-to-oocyte organelle transfer in mammals and consolidate concerns, raised repeatedly at recent germ cell meetings, regarding the reproducibility of the Lei & Spradling results in mouse oocytes.

Major points requiring attention/ addressing:

- 1- Claim of recapitulation of oocyte folliculogenesis: The authors should provide stronger evidence that the in vitro-grown oocytes are comparable to their in vivo counterparts. At present, axis formation is the only criterion used to support this claim, but additional markers or evidence are needed to show that the culture system truly recapitulates both oocyte formation and growth. Examining other proteins involved in meiotic processes and/or stage-specific transcriptional markers would provide a more robust assessment of oocyte formation and growth in the ex vivo system.
- 2- Germ cell growth dynamics were analyzed in 2D (Figure 2). These measurements are more appropriate in 3D. If this is not possible, the authors should provide evidence that their 2D measurements are not confounded by the flattening of cultured ovaries over time, which is a known issue.
- 3- The manuscript lacks a simple but important control: Does imaging itself alter the measured dynamics? Authors should check (i.e., by comparing non-imaged cultured ovaries with long-term imaged ones) if phototoxicity/ imaging conditions contribute to the observed phenomena.
- 4- The study needs validation of blebbing in vivo ovaries.
- 5- Similarly, synchronous degeneration should be validated in in vivo ovaries -e.g., is continuous degeneration prominent in freshly isolated E16.5 ovaries (not cultured in vitro)?
- 6- The Mito-EGFP and Dendra2 mitochondrial signals appear quite different; in the latter, mitochondria look more punctate. How do the authors explain this difference?
- 7- In Movie S9, there are clear "pink" traces in the oocyte adjacent to the photoconverted oocyte from an early timepoint onwards. How do the authors explain this signal? There also appears to be a large pink signal after oocyte degeneration-again, an explanation is required.

Minor

- 1- The statement "we successfully tracked the formation of the Golgi ring, a hallmark of the Balbiani body" is misleading. The Golgi ring is a Golgi apparatus structure (Dhandapani et al., JCS) and does not by itself imply the presence of a Balbiani body, as it is also observed in many other cell types that lack one.

Referee #3:

Oogenesis in mammals, a pivotal process for reproductive success, involves the critical step of oogonia entering meiosis and assembling into distinct follicles. While considerable progress has been made in understanding this process, the fundamental mechanisms underlying oocyte formation remain largely elusive. The fact that oocyte formation occurs in vivo within the fetal gonads during gestation adds an additional layer of complexity to mechanistic studies. Therefore, establishing a reliable ex vivo system that can recapitulate this in vivo process is crucial for advancing our understanding of the mechanisms governing mammalian oocyte formation. In this manuscript, Aizawa et al. established an ex vivo culture system for mouse fetal ovaries that not only recapitulates the early events of in vivo oogenesis, but also enables live recording of oocyte formation, leveraging state-of-the-art live cell imaging technologies and mouse genetic engineering technique.

Building on this ex vivo system, Aizawa et al. made several intriguing observations, including the pronounced blebbing activity in oogonia driven by the meiotic initiation signal during the mitosis-to-meiosis transition, as well as the synchronous degeneration of germ cells, with no detectable transfer of mitochondria or centrosomes from the degenerating germ cells to the future oocyte during germ cell nest breakdown. These findings are novel, with the latter challenging the prevailing view on mammalian oocyte

formation-that organelles are transferred from nursing germ cells to the dominant oocyte within a germ cell nest. This discovery could significantly reshape future studies in the field.

A major concern with this study is that all the observations are based on an ex vivo system. The question remains as to how well these findings correlate with in vivo physiology. Although the authors stained germ cells for SYCP3 to indicate their entry into meiosis at the appropriate timing, as observed in vivo, this marker alone is not sufficient to confirm normal entry and progression through meiotic prophase I. The authors may consider conducting a detailed chromatin spread experiment to compare the progression of meiotic prophase I in the germ cells cultured in vitro with those at the same developmental stage in vivo.

Otherwise, readers may view the statement that 'the ex vivo culture system faithfully recapitulates in vivo development' as an overstatement. 10x genomics single cell RNA-seq is another choice, but it may take a while.

Some minor concerns and suggestions:

1. The title could be revised to more accurately reflect the key findings of the paper.
2. Consider including only the fetal ovary data in the results, as the sex of the cultured gonads could eventually be determined either by genotyping or later morphological analysis.
3. On Page 25, the section titled "Generation of mKikGR-CETN2 Mice" could be moved to the "Mice" section, following the first paragraph on Page 19.
4. On Page 6, line 20, "we used Stella-ECFP reporter mice, in which the germline-specific gene Stella (also known as 21 Dppa3)" - please switch the order of the words "Stella" and "Dppa3," as Dppa3 is the standard nomenclature for this gene.
5. On Page 16, second paragraph, the phrasing seems ambiguous. Do the authors intend to convey that intercellular bridges are essential for meiotic entry and progression, or are they mediators of germ cell death in the female fetal ovaries?
6. On Page 17, lines 3-9, this section needs revision. Did the authors actually observe the coexistence of the two types of germ cell elimination in their system?
7. The blebbing phenomenon is an intriguing and unusual observation. The authors should discuss its relevance to oocyte formation in more detail. Does it relate to exocytosis and possibly play a role in germ cell-somatic cell interactions?
8. On Page 10, lines 9-11, this appears to be an incomplete sentence.
9. On Page 23, lines 8-10, the vendors of the reagents should be provided.
10. In Figure 1G, the Y-axis label should be "distance between the nucleus and the cell surface center." Additionally, it should clearly indicate the two groups that are statistically significantly different.
11. In Figure 3K, a higher-quality image of the SYCP3 staining is needed.

Point-by-point response

Referee #1:

The authors established an *ex vivo* culture system and performed long-term live imaging of mouse fetal ovaries. Using high-resolution imaging, they reported blebbing activity during the mitosis-to-meiosis transition of oogonia. They observed no detectable transfer of mitochondria or centrosomes, as organelles remained confined to individual cells. The authors concluded that alternative mechanisms for cytoplasmic enrichment in oocytes may exist, though the exact nature of these mechanisms remains unclear.

While I appreciate the authors' efforts in developing a powerful platform for real-time analysis of germ cell dynamics, I have serious reservations about whether culturing embryonic E12.5 ovaries for extended periods (7 to 36 days) in ordinary α MEM can accurately represent physiological conditions during oogonia and oocyte cyst breakdown. For sure necrosis would occur. Despite the small size of embryonic ovaries, I find it difficult to believe that such prolonged *ex vivo* culture maintains the organ in a truly physiological state. Consequently, many of the observed phenomena-though documented with impressive technical skill-may not reflect biological reality.

Given these concerns, I suggest that this technically advanced study may be better suited for a methods-focused journal.

Response:

We thank the reviewer for raising this important concern regarding the physiological relevance of prolonged *ex vivo* culture. We fully agree that rigorous validation is required to ensure that the dynamic behaviors observed reflect genuine developmental processes rather than culture-induced artifacts.

Specifically, we have expanded cytological and transcriptional assessments (Referee #2, comment 1), including analysis of synapsis (SYCP1/SYCP3 colocalization), meiotic crossover designation (MLH1 foci), and stage-matched RT-qPCR profiling of representative oocyte-related genes. We have also quantitatively compared germ cell growth dynamics between *in vivo* and *ex vivo* samples by measuring cell size distributions across developmental stages (Referee #2, comment 2). In addition, we confirmed the presence of germ cell membrane blebs in fixed *in vivo* ovary samples (Referee #2, comment 4) and confirmed synchronous germ cell degeneration using live imaging of freshly isolated fetal ovaries (Referee #2, comment 5).

Together, these expanded cytological and transcriptional analyses provide evidence that the *ex vivo* culture system faithfully recapitulates meiotic progression and stage-matched oocyte differentiation within the developmental window examined. Furthermore, the key dynamic and

developmental features described in this study are detectable under *in vivo* or minimally perturbed conditions and show close correspondence between *ex vivo* and *in vivo* samples. We have revised the manuscript accordingly and moderated statements where appropriate to avoid overinterpretation.

We respectfully believe that these additions reinforce the biological relevance of the *ex vivo* system and strengthen the conclusions of the study.

Referee #2:

Aizawa et al. establishes an *ex vivo* culture and live-imaging system that enables continuous visualization of mouse oocyte formation from the mitosis-to-meiosis transition through early folliculogenesis. Using this approach, they uncover dynamic germ cell behaviors, including pronounced blebbing concurring with meiotic initiation and ROCK-dependent contractility, as well as the synchronous degeneration of oocytes in cysts. Contrary to the model of cytoplasmic enrichment via organelle transfer from nurse cells, photoconversion-based tracking revealed no detectable transfer of mitochondria or centrosomes between germ cells. These findings suggest the concept of nurse-to-oocyte cytoplasmic organelle donation might be specialized for species such as *Drosophila*, and instead point to alternative mechanisms in mammals, such as engulfment of degenerating cells, as contributors to oocyte formation.

Taken together, this work is timely, well-executed and a valuable contribution to the field. IF supported by additional *in vivo* evidence (pls see below) it could effectively resolve the debate about nurse cell-to-oocyte organelle transfer in mammals and consolidate concerns, raised repeatedly at recent germ cell meetings, regarding the reproducibility of the Lei & Spradling results in mouse oocytes.

Major points requiring attention/ addressing:

1- Claim of recapitulation of oocyte folliculogenesis: The authors should provide stronger evidence that the *in vitro*-grown oocytes are comparable to their *in vivo* counterparts. At present, axis formation is the only criterion used to support this claim, but additional markers or evidence are needed to show that the culture system truly recapitulates both oocyte formation and growth. Examining other proteins involved in meiotic processes and/or stage-specific transcriptional markers would provide a more robust assessment of oocyte formation and growth in the *ex vivo* system.

Response:

We thank the reviewer for this important suggestion. We agree that demonstrating faithful recapitulation of oocyte development requires evaluation beyond chromosome axis formation alone. In response, we performed additional cytological and transcriptional analyses, which have now been incorporated into the revised manuscript as Figure 1E-G. For clarity, these figure panels and their legends are shown below.

First, we assessed synapsis and crossover formation by immunostaining for SYCP3 and SYCP1 and by examining MLH1 foci in *ex vivo*-cultured ovaries compared with stage-matched *in vivo* controls (new Fig. 1E, F). In both conditions, pachytene-stage oocytes exhibited robust colocalization of SYCP3 and SYCP1 along fully synapsed chromosome axes. Moreover, MLH1 foci were detected at comparable frequencies and distributions in *ex vivo* and *in vivo* oocytes, indicating normal crossover designation. These results demonstrate that key events of meiotic prophase I, including axis formation, synapsis, and recombination, are faithfully reproduced in the *ex vivo* system.

Second, to evaluate stage-specific transcriptional progression, we performed RT-qPCR analyses of representative oocyte-related genes encompassing transcription factors (e.g., *Nobox*, *Figla*, *Sohlh1*), meiotic regulators (*Spo11*, *Hormad1*), cohesin components (*Rec8*, *Smc1b*), zona pellucida genes (*Zp3*), and DNA methylation factors (*Dnmt3a*, *Dnmt3b*) (new Fig. 1G). Expression profiles in *ex vivo* samples closely mirrored those of stage-matched *in vivo* ovaries, demonstrating appropriate temporal activation of oocyte-specific transcriptional programs.

Together, these additional cytological and transcriptional analyses provide evidence that the *ex vivo* culture system faithfully recapitulates meiotic progression and stage-matched oocyte differentiation, thereby substantially strengthening our original conclusion. These observations are described in the revised manuscript (Lines 134-166).

G

- (E) Representative chromosome spreads of oocytes derived from E17.5 ovaries and E12.5 gonads cultured *ex vivo* for 5 days. Immunostaining was performed using antibodies against SYCP1 and SYCP3. In merged images, SYCP1 is displayed in magenta and SYCP3 in green. Scale bar, 5 μ m.
- (F) Representative chromosome spreads of oocytes derived from E18.5 ovaries and E12.5 gonads cultured *ex vivo* for 6 days. Immunostaining was performed using antibodies against MLH1 and SYCP3. Images are maximum-intensity Z-projections generated from three optical sections for each marker acquired at 200 nm Z-intervals. Scale bar, 5 μ m.
- (G) Quantitative RT-PCR analysis of stage-specific gene expression in developing female germ cells *in vivo* and *ex vivo*. Relative expression levels of transcription factors (*Nobox*, *Figla*, *Sohlh1*), meiotic progression genes (*Spo11*, *Hormad1*), cohesin complex components (*Rec8*, *Smc1b*), zona pellucida gene (*Zp3*), and DNA methylation regulators (*Dnmt3a*, *Dnmt3b*) were measured in Stella-ECFP-positive germ cells isolated by FACS from *in vivo* ovaries (E14.5, E17.5, PD1) and *ex vivo* cultured gonads (E12.5 + 2d, + 5d, + 8d). Ct values were normalized to the housekeeping gene *Rplp0*. Data are presented as $-\Delta\Delta\text{Ct}$ (log₂ fold change) relative to the mean ΔCt value of E14.5 samples. Bars shown mean \pm SD from three biological replicates, each analyzed with two technical replicates. Statistical analyses were performed on ΔCt values using a two-tailed Welch's t-test. ns, not significant; * $P < 0.05$, ** $P < 0.01$.

2- Germ cell growth dynamics were analyzed in 2D (Figure 2). These measurements are more appropriate in 3D. If this is not possible, the authors should provide evidence that their 2D

measurements are not confounded by the flattening of cultured ovaries over time, which is a known issue.

Response:

We appreciate the reviewer's comment regarding the limitations of 2D measurements and the potential confounding effects of ovary flattening during long-term culture. In principle, 3D volume measurements would indeed be ideal. However, in our hands, long-term 3D live imaging of oocyte formation over the full period analyzed in Fig. 2D (approximately 5 days) is technically challenging due to substantially increased phototoxicity compared with 2D imaging, which compromises tissue viability and developmental progression.

To directly address the concern that our 2D analyses might be confounded by flattening of cultured ovaries over time, we performed additional experiments comparing germ cell areas measured *in vivo* and *ex vivo* across the corresponding developmental stages (E15.5-PD2). These new data are presented in new Fig. 3C (shown below). Although the mean germ cell area in *ex vivo* samples tended to be slightly larger than *in vivo* at some stages, no statistically significant differences were detected at any time point (all $P > 0.05$; e.g., PD2 vs. E12.5 + 9d, mean = 271.0 vs. 299.7 μm^2). These results indicate that, despite potential tissue flattening, the 2D germ cell area measurements remain representative of physiological oocyte development *in vivo*.

Also, the oocyte areas tracked by long-term live imaging in new Fig. 3B (measurement of oocyte area during *ex vivo* culture) at E12.5 + 5d (130.9 μm^2), E12.5 + 7d (187.8 μm^2), and E12.5 + 9d (328.4 μm^2) fall within the interquartile range (25th-75th percentile) of the corresponding *ex vivo* populations measured in new Fig. 3C. This indicates that the cells followed in new Fig. 3B represent typical germ cell growth behavior rather than outliers.

Taken together, these results support the conclusion that flattening of cultured ovaries does not substantially confound our 2D measurements, and that the 2D germ cell area quantifications used in this study reliably reflect oocyte growth dynamics during this developmental window. These results have been incorporated into the revised manuscript (lines 235-242).

(C) Comparison of germ cell area between *in vivo* (pink) and *ex vivo* (blue) gonads/ovaries from E15.5 to PD2. Three-dimensional confocal z-stacks were acquired, and the single z-plane containing the maximal cross-sectional area of each targeted germ cell was selected. Cell boundaries were manually delineated by tracing the PlasMem signal. Data points (circles) represent areas of individual germ cells. Box-and-whisker plots show the 25th, 50th (median), and 75th percentiles (box), with whiskers indicating the minimum and maximum values. Identical datasets for E12.5 + 3d, E12.5 + 5d, and E12.5 + 7d are shown in Fig 2H. Statistical comparisons were performed using Welch's t-test. Sample sizes (germ cells; gonads/ovaries) were: E15.5 (50; 3), E12.5 + 3d (40; 7), E17.5 (50; 3), E12.5 + 5d (40; 10), PD0 (50; 4), E12.5 + 7d (40; 8), PD2 (50; 4), and E12.5 + 9d (50; 3).

3- The manuscript lacks a simple but important control: Does imaging itself alter the measured dynamics? Authors should check (i.e., by comparing non-imaged cultured ovaries with long-term imaged ones) if phototoxicity/ imaging conditions contribute to the observed phenomena.

Response:

We thank the reviewer for this important suggestion. To examine whether long-term imaging influences the observed degeneration dynamics, we cultured E12.5 gonads *ex vivo* without imaging pre-exposure and performed live imaging only at E12.5 + 4d. We then compared germ cell cluster degeneration between continuously imaged samples and non-imaged samples.

As shown in new Fig. 3H, the distribution of cluster sizes at the time of degeneration was comparable between “E12.5 + 4d non-imaged” and “E12.5 + 4d imaged” conditions, with no statistically significant differences detected. These results indicate that synchronous degeneration is not induced by phototoxicity or long-term imaging conditions.

Because the same experimental framework also addresses whether synchronous degeneration occurs under near-physiological conditions, a detailed description of these additional analyses is described in our response to Comment 5 below.

4- The study needs validation of blebbing in vivo ovaries.

Response:

We thank the reviewer for this important point. To validate that germ cell blebbing also *occurs in vivo*, we analyzed freshly isolated fetal ovaries after fixation and new data are presented in new Fig. 4F-G (shown below). Membrane protrusions consistent with blebs were observed in germ cells in E14.5 ovaries, with a mean frequency of 37.1%. In contrast, the proportion of germ cells with blebs was significantly reduced to 2.8% in E16.5 ovaries.

This developmental decline in germ cells with blebs closely mirrors our *ex vivo* observations, in which germ cells exhibit frequent blebbing at E12.5 + 2d (12.4 blebs per cell per 10 min) but show a rapid reduction by E12.5 + 4d (1.8 blebs per cell per 10 min), as shown in new Fig. 4D. Together, these results demonstrate that germ cell blebbing is not an artifact of *ex vivo* culture but rather represents a physiological cellular behavior that also occurs *in vivo* during the corresponding developmental window. These additional analyses have been incorporated into the revised manuscript (lines 308-315).

- (F) Representative images of germ cells in ovaries at E14.5 and E16.5 expressing Stella-ECFP following fixation. Ovaries were stained with PlasMem Bright Red. Merged images show PlasMem (magenta) and Stella-ECFP (green). Arrowheads indicate blebs on germ cells. Scale bar, 5 μ m.
- (G) Quantification of germ cells exhibiting blebs in ovaries at E14.5 and E16.5. Bars represent mean \pm standard deviation. For each ovary, 80 germ cells were assessed for the presence or absence of blebs. $N = 3$ ovaries per stage. Statistical analysis was performed using Welch's corrected t-test. * $P < 0.05$.

5- Similarly, synchronous degeneration should be validated in *in vivo* ovaries -e.g., is continuous degeneration prominent in freshly isolated E16.5 ovaries (not cultured *in vitro*)?

Response:

We thank the reviewer for raising this important point. To determine whether synchronous germ cell cluster degeneration also occurs under conditions closely reflecting the *in vivo* state, we performed live imaging of freshly isolated E16.5 ovaries (new Fig. 3G-H; shown below).

Because synchronous degeneration cannot be reliably assessed in fixed samples, dissected ovaries were subjected to short-term *ex vivo* imaging (24 h) after isolation. Time-lapse analysis revealed clear examples of synchronized degeneration of germ cell clusters in E16.5 ovaries (new Fig. 3G), accompanied by residual nuclear signals indicative of apoptosis.

Quantification of cluster size at the time of degeneration demonstrated a comparable range (1–8 cells) between E16.5 ovaries, E12.5 + 4d gonads cultured without imaging pre-exposure (“E12.5 + 4d non-imaged”), and E12.5 + 4d gonads subjected to long-term imaging (“E12.5 + 4d imaged”) (new Fig. 3H). No statistically significant differences were detected among these conditions.

Together, these results demonstrate that synchronous germ cell cluster degeneration:

1. Is not an artifact of long-term imaging.
2. Is not induced by phototoxicity.
3. Occurs in freshly isolated ovaries under near-physiological conditions.

These additional analyses have been incorporated into the revised manuscript (line 258-272).

(G) Representative time lapse images of an E16.5 ovary illustrating synchronized degeneration of a germ cell cluster. The ovary, expressing H2B-mCherry, was stained with PlasMem Bright Green and imaged *ex vivo* every 10 min for 24 h. Merged images show PlasMem (green) and H2B-mCherry (magenta). Dotted lines outline the germ cell cluster; the asterisk marks germ cells exhibiting abnormal morphology; arrows indicate residual particles following degeneration. Scale bar, 10 μ m. See also Movie EV2.

(H) Quantification of germ cell cluster size at the time of degeneration in freshly isolated E16.5 ovaries (referred to as “E16.5”), E12.5 gonads cultured for 4 days without prior imaging and subjected to live imaging only after the 4-day culture period (“E12.5 + 4d non-imaged”), and E12.5 gonads cultured for 4 days under continuous live imaging conditions (“E12.5 + 4d

imaged”). For each degeneration event, the number of degenerating germ cells was recorded within twelve randomly selected $100\ \mu\text{m} \times 100\ \mu\text{m}$ regions per developmental window. Data points (circles) represent cluster sizes for individual degeneration events. Box-and-whisker plots depict the 25th, 50th (median), and 75th percentiles (box), with whiskers indicating minimum and maximum values. Seven E16.5 ovaries were analyzed for “E16.5,” six gonads for “E12.5 + 4d non-imaged,” and six gonads for “E12.5 + 4d imaged.” Identical data from “E12.5 + 4d imaged” are shown in Fig. 3E. Statistical comparison was performed using Welch’s corrected t-test.

6- The Mito-EGFP and Dendra2 mitochondrial signals appear quite different; in the latter, mitochondria look more punctate. How do the authors explain this difference?

Response:

We thank the reviewer for this observation. In principle, Mito-EGFP also displays a punctate mitochondrial pattern, as shown in the original report (Fig. 1F in Abe et al., 2011; PMID: 21445964). The difference in mitochondrial morphology between new Fig. EV7B/Movie EV9 (Mito-EGFP) and new Fig. 5C/Movie EV10 (Mito-Dendra2) arises from differences in imaging acquisition parameters rather than intrinsic biological differences.

Because Mito-EGFP does not permit photoconversion-based tracking, potential transfer events had to be monitored by continuously observing mitochondrial behavior at high temporal resolution (every 1 minute). To minimize phototoxicity during prolonged imaging (~10 hours), exposure time and excitation intensity were reduced, resulting in lower signal intensity and a less distinctly punctate appearance.

In contrast, in the Mito-Dendra2 experiments, a defined mitochondrial population was photoconverted and subsequently tracked based on its converted fluorescence. Because transfer could be assessed by monitoring the spatial distribution of the converted signal, continuous high-frequency imaging was not required. This allowed imaging at lower temporal resolution (every 5 minutes), enabling longer exposure times and improved signal-to-noise ratio, thereby resolving mitochondria with a clearer punctate morphology.

Thus, the observed difference reflects technical imaging parameters required by the experimental design rather than differences in mitochondrial organization.

7- In Movie S9, there are clear "pink" traces in the oocyte adjacent to the photoconverted oocyte from an early timepoint onwards. How do the authors explain this signal? There also appears to be a large pink signal after oocyte degeneration-again, an explanation is required.

Response:

We thank the reviewer for this careful observation. As noted, faint "pink" signal is detectable in the oocyte adjacent to the photoconverted cell from early time points, and a more prominent "pink" signal appears following degeneration of the photoconverted oocyte.

The faint early signal in adjacent oocytes is most likely due to unintended partial photoconversion of mitochondria outside the targeted region during the initial UV illumination. Such off-target photoconversion would generate low-level Red Mito-Dendra2 fluorescence in neighboring cells. This interpretation is consistent with our quantitative analysis (new Figs. 5D, EV8), in which some adjacent, non-targeted oocytes displayed a modestly elevated Red Mito-Dendra2 intensity at the initial time point, followed by a gradual decline over time.

The larger "pink" signal observed after oocyte degeneration is most likely attributable to increased autofluorescence associated with cellular degeneration. We consistently detected elevated red-channel signal in degenerating germ cells, even in the absence of photoconversion, indicating that this signal reflects a degeneration-associated optical artifact rather than mitochondrial redistribution.

Taken together, these signals are best explained by technical artifacts (off-target photoconversion and degeneration-associated autofluorescence) rather than bona fide mitochondrial transfer events. We have clarified this point in the revised manuscript (lines 404-408).

Minor

1- The statement "we successfully tracked the formation of the Golgi ring, a hallmark of the Balbiani body" is misleading. The Golgi ring is a Golgi apparatus structure (Dhandapani et al., JCS) and does not by itself imply the presence of a Balbiani body, as it is also observed in many other cell types that lack one.

Response:

We thank the reviewer for this clarification. We agree that the Golgi ring is not necessarily a hallmark of the Balbiani body, as it can also be observed in other cell types lacking this structure. We have therefore revised the text to remove the wording "a hallmark of the Balbiani body" and now describe the observed structure more precisely as a Golgi ring-like morphology. In addition,

we explicitly state that this structure alone is not sufficient to define the presence of a Balbiani body in the revised manuscript (lines 383-386).

Referee #3:

Oogenesis in mammals, a pivotal process for reproductive success, involves the critical step of oogonia entering meiosis and assembling into distinct follicles. While considerable progress has been made in understanding this process, the fundamental mechanisms underlying oocyte formation remain largely elusive. The fact that oocyte formation occurs *in vivo* within the fetal gonads during gestation adds an additional layer of complexity to mechanistic studies. Therefore, establishing a reliable *ex vivo* system that can recapitulate this *in vivo* process is crucial for advancing our understanding of the mechanisms governing mammalian oocyte formation. In this manuscript, Aizawa et al. established an *ex vivo* culture system for mouse fetal ovaries that not only recapitulates the early events of *in vivo* oogenesis, but also enables live recording of oocyte formation, leveraging state-of-the-art live cell imaging technologies and mouse genetic engineering technique.

Building on this *ex vivo* system, Aizawa et al. made several intriguing observations, including the pronounced blebbing activity in oogonia driven by the meiotic initiation signal during the mitosis-to-meiosis transition, as well as the synchronous degeneration of germ cells, with no detectable transfer of mitochondria or centrosomes from the degenerating germ cells to the future oocyte during germ cell nest breakdown. These findings are novel, with the latter challenging the prevailing view on mammalian oocyte formation—that organelles are transferred from nursing germ cells to the dominant oocyte within a germ cell nest. This discovery could significantly reshape future studies in the field.

A major concern with this study is that all the observations are based on an *ex vivo* system. The question remains as to how well these findings correlate with *in vivo* physiology. Although the authors stained germ cells for SYCP3 to indicate their entry into meiosis at the appropriate timing, as observed *in vivo*, this marker alone is not sufficient to confirm normal entry and progression through meiotic prophase I. The authors may consider conducting a detailed chromatin spread experiment to compare the progression of meiotic prophase I in the germ cells cultured *in vitro* with those at the same developmental stage *in vivo*. Otherwise, readers may view the statement that 'the *ex vivo* culture system faithfully recapitulates *in vivo* development' as an overstatement. 10x genomics single cell RNA-seq is another choice, but it may take a while.

Response:

We thank the reviewer for this thoughtful and constructive comment. We understand the reviewer's central concern that our conclusions rely primarily on observations made in an *ex vivo*

system, and that stronger validation is needed to demonstrate how closely this system reflects *in vivo* physiology. We agree that assessing meiotic progression based on chromosome axis formation alone is not sufficient to fully establish faithful meiotic progression and developmental equivalence.

To address this concern, we have expanded our analyses to more comprehensively evaluate meiotic progression and oocyte differentiation in the *ex vivo* system. As detailed in our response to Referee #2 (comment 1), we have now included additional cytological and transcriptional assessments (new Fig. 1E–G). Specifically, we examined synapsis and crossover designation by immunostaining for SYCP3 and SYCP1, as well as MLH1 foci, in stage-matched *in vivo* and *ex vivo* samples. *Ex vivo* oocytes exhibited robust SYCP3/SYCP1 colocalization along fully synapsed chromosome axes and comparable MLH1 foci frequency and distribution, indicating normal pachytene progression and crossover designation. In addition, we performed RT–qPCR analyses of representative stage-specific genes spanning transcription factors (Nobox, Figla, Sohlh1), meiotic regulators (Spo11, Hormad1), cohesin components (Rec8, Smc1b), zona pellucida genes (Zp3), and DNA methylation factors (Dnmt3a, Dnmt3b). These expression profiles closely mirrored those observed in stage-matched *in vivo* ovaries. Together, these analyses provide a substantially more rigorous evaluation of meiotic progression and oocyte differentiation across the developmental window examined in this study.

In addition, as described in our responses to Referee #1 and Referee #2, we performed several complementary validations to assess the physiological relevance of the *ex vivo* system. These include: (1) quantitative comparison of germ cell size between *in vivo* and *ex vivo* samples (Referee #2, comment 2), (2) validation of blebbing in freshly isolated ovaries (Referee #2, comment 4), and (3) validation of synchronous germ cell degeneration in freshly isolated ovaries (Referee #2, comment 5). Together, these experiments demonstrate that key cytological, morphometric, and dynamic features observed *ex vivo* are also detectable under near-physiological conditions.

We have revised the manuscript accordingly and have moderated statements regarding the extent to which the *ex vivo* system recapitulates *in vivo* development to avoid overstatement.

Some minor concerns and suggestions:

1. The title could be revised to more accurately reflect the key findings of the paper.

Response:

We thank the reviewer for this helpful suggestion. We have revised the title to more accurately reflect the key findings of our study: “Dynamic blebbing and absence of organelle transfer during mouse oocyte formation.”

2. Consider including only the fetal ovary data in the results, as the sex of the cultured gonads could eventually be determined either by genotyping or later morphological analysis.

Response:

We thank the reviewer for this important point. We took multiple steps to ensure that male gonads were not inadvertently included in our analyses. At E12.5, gonadal sex can be reliably distinguished morphologically, as male gonads exhibit prominent testis cord structures that are absent in female gonads; this comparison has been added to new Fig. 1B (shown below). At E11.5, when morphological identification is not yet reliable, we performed PCR-based genotyping prior to culture to confirm sex (new Fig. EV5A). Furthermore, for E12.5 samples, subsequent culture revealed characteristic features of ovarian development, including germ cell cyst formation and meiotic entry, providing additional confirmation of ovarian identity. Therefore, all data presented in this study were derived exclusively from female gonads. We have clarified these procedures in the revised manuscript to avoid ambiguity (lines 124-126; 298-300).

(B) Representative images of female and male gonads at E12.5. The male gonad exhibits distinct testis cord structures, whereas the female gonad lacks cord formation and displays a uniformly unstructured morphology. M, mesonephros; G, gonad. Scale bar, 500 μ m.

3. On Page 25, the section titled "Generation of mKikGR-CETN2 Mice" could be moved to the "Mice" section, following the first paragraph on Page 19.

Response:

We have revised the manuscript accordingly and moved the section titled "Generation of mKikGR-CETN2 Mice" to the "Mice" section (lines 600-626).

4. On Page 6, line 20, "we used Stella-ECFP reporter mice, in which the germline-specific gene Stella (also known as 21 Dppa3)" - please switch the order of the words "Stella" and "Dppa3," as Dppa3 is the standard nomenclature for this gene.

Response:

We have revised the text accordingly, listing Dppa3 first as the standard gene nomenclature, followed by Stella (line 148).

5. On Page 16, second paragraph, the phrasing seems ambiguous. Do the authors intend to convey that intercellular bridges are essential for meiotic entry and progression, or are they mediators of germ cell death in the female fetal ovaries?

Response:

We thank the reviewer for pointing out this ambiguity. Our intention was not to suggest that intercellular bridges are essential for meiotic entry or progression. Rather, we aimed to propose a potential role for intercellular bridges in mediating the synchronous germ cell degeneration observed in this study. We have revised the Discussion to clarify that this role remains speculative and context-dependent, and that intercellular bridges may contribute to coordinated degeneration without being required for meiotic progression itself (lines 511-525).

6. On Page 17, lines 3-9, this section needs revision. Did the authors actually observe the coexistence of the two types of germ cell elimination in their system?

Response:

In our analyses, we observed both synchronous and asynchronous germ cell elimination events. As shown in new Fig. 3E-F, multicellular, synchronous degeneration accounted for the majority of events, particularly between E15.5 and E18.5, while single-cell or smaller cluster degeneration was also present. We have revised the Discussion to clearly state this observation and to explicitly reference the corresponding data (lines 537-546).

7. The blebbing phenomenon is an intriguing and unusual observation. The authors should discuss its relevance to oocyte formation in more detail. Does it relate to exocytosis and possibly play a role in germ cell-somatic cell interactions?

Response:

We thank the reviewer for this suggestion. In response, we have expanded the Discussion (lines 478-502) to further elaborate on the potential relevance of germ cell blebbing during oocyte formation. We now discuss its preferential occurrence at germ cell–somatic interfaces, its temporal coincidence with the mitosis-to-meiosis transition, and the possibility that blebbing modulates heterotypic cell–cell interactions during this developmental window. While we did not directly assess exocytosis, we have included discussion of how blebbing could relate to local membrane organization or signaling at germ cell–somatic contacts.

8. On Page 10, lines 9-11, this appears to be an incomplete sentence.

Response:

To enhance clarity, we have revised the sentence accordingly in the revised manuscript (lines 304-307)

9. On Page 23, lines 8-10, the vendors of the reagents should be provided.

Response:

We have now included vendor information for the relevant reagents in the revised manuscript (lines 805-809).

10. In Figure 1G, the Y-axis label should be "distance between the nucleus and the cell surface center." Additionally, it should clearly indicate the two groups that are statistically significantly different.

Response:

We thank the reviewer for this suggestion. The Y-axis label has been revised to "Distance between the nucleus and the cell surface center" as suggested. We have also clarified the statistical comparisons in the revised Figure 2E (formerly Figure 1G). Statistical analyses were performed between consecutive developmental stages within each cell type (germ cells and somatic cells), and the figure now clearly indicates which comparisons reached statistical significance. As

described in the revised text (lines 179-181), germ cells showed significant stage-dependent changes, whereas somatic cells did not.

11. In Figure 3K, a higher-quality image of the SYCP3 staining is needed.

Response:

The dim SYCP3 staining in the original Figure 3K reflects the fact that the oocytes were from E12.5 + 3d cultures, corresponding to a stage before full chromosome axis formation. To provide a clearer representation of SYCP3 localization, we have replaced this panel with images from E12.5 + 5d cultures (new Fig. 4L; shown below), in which many oocytes are at the pachytene stage and exhibit more prominent chromosome axis staining.

(L) Representative immunostaining of SYCP3 in cultured gonads following ROCKi treatment. E12.5 + 5d gonads were stained with antibodies against SYCP3 and MVH, followed by DAPI counterstaining. SYCP3-positive chromosome axes were detected in germ cells irrespective of ROCKi treatment. Scale bar, 10 μ m.

Dr. Tomoya S Kitajima
RIKEN Center for Biosystems Dynamics Research
Laboratory for Chromosome Segregation
2-2-3 Minatojima-minamimachi, Chuo-ku
Kobe
650-0047
Japan

25th Mar 2026

Re: EMBOJ-2025-122016R
Dynamic blebbing and absence of organelle transfer during mouse oocyte formation

Dear Tomoya,

Thank you for submitting your revised manuscript on live imaging of mouse oocyte formation to The EMBO Journal. Two of the original referees have now assessed it once more, and were both fully satisfied with the revision. We shall therefore be happy to accept the manuscript for publication, following incorporation of a few remaining editorial points as follows:

- Please adjust the order of the manuscript sections, and also make sure to use the correct section headers: Single title page with complete author information, Abstract, Introduction, Results, Discussion, Methods, Data Availability, Acknowledgements, Disclosure and Competing Interests Statement, References, Main Figure Legends, Tables, Expanded Figure Legends.
- Please remove the ORCID listings from the title page - we can only display and link them to the publication if each author by themselves links their ORCID to their author profile in our system (please contact our office if you should need help with this).
- To facilitate transferring and handling of the figure files, please provide them in less extensive/better compiled format, with more modest file sizes.
- As we are switching from a free-text author contribution statement towards a more formal statement based on Contributor Role Taxonomy (CRediT) terms, please remove the present Author Contribution section and instead specify each author's contribution(s) directly in the Author Information page of our submission system during upload of the final manuscript. See <https://casrai.org/credit/> for more information.
- Please carefully go through the reference list and make sure that all references have their complete citation information - including year, journal name & volume, and page/locator numbers - such information is currently missing for several of them.
- Please provide suggestions for a short 'blurb' text prefacing and summing up the conceptual aspect of the study in two sentences (max. 250 characters), followed by 3-5 one-sentence 'bullet points' with brief factual statements of key results of the paper; they will form the basis of an editor-written 'Synopsis' accompanying the online version of the article. Please also upload a synopsis image, which can be used as a "visual title" for the synopsis section of your paper. The image (maybe based on a more compacted version of Figure 7?) should be ideally in JPG format, and please make sure that it remains in the modest dimensions of (exactly) 550 pixels wide and between 300-500 pixels high.
- Finally, during routine pre-acceptance checks, our data editors have raised the following queries regarding figures, data, and legends; I would appreciate if you briefly answered to them in the cover letter of your final submission, and made the requested text modifications with changes/additions highlighted via the "Track changes" option, to facilitate our final checking":
 1. Please note that the exact p values are not provided in the legends of figures 1G; 2E; 4D,G,I,J; EV-6B,C,H.
 2. Please note that the error bars are not defined in the legends of figures EV-3C.

I am returning the manuscript to you for a final round of minor revision, solely to allow you to make these modifications and upload the revised files. Once we will have received them, we should be ready to swiftly proceed with formal acceptance and production of the manuscript.

With kind regards,

Hartmut

Hartmut Vodermaier, PhD

*** PLEASE NOTE: All revised manuscripts are subject to initial checks for completeness and adherence to our formatting guidelines. Revisions may be returned to the authors and delayed in their editorial re-evaluation if they fail to comply with the following requirements. As a first step please read our guidelines for revised submissions:
<https://link.springer.com/journal/44318/submission-guidelines#cms-Revised-submissions>

1) Every manuscript requires a Data Availability section (even if only stating that no deposited datasets are included). Primary datasets or computer code produced in the current study have to be deposited in appropriate public repositories prior to resubmission, and reviewer access details provided in case that public access is not yet allowed.

- size of the scale bars that are mandatory for all micrograph panels
- the statistical test used to generate error bars and P-values
- the type of error bars (e.g., S.E.M., S.D.)
- the number (n) and nature (biological or technical replicate) of independent experiments underlying each data point
- Figures may not include error bars for experiments with $n < 3$; scatter plots showing individual data points should be used instead.

3) Revised manuscript text (including main tables, and figure legends for main and EV figures) has to be submitted as an editable text file (e.g., .docx format). We encourage highlighting of changes (e.g., via text color) for the referees' reference.

4) Each main and each Expanded View (EV) figure should be uploaded as individual production-quality files (preferably in .eps, .tif, .jpg formats). For suggestions on figure preparation/layout, please refer to our Figure Preparation Guidelines:
<https://media.springernature.com/original/springer-cms/rest/v1/content/27825798/data/v1>

5) Point-by-point response letters should include the original referee comments in full together with your detailed responses to them (and to specific editor requests if applicable), and also be uploaded as an editable (e.g., .docx) text file.

6) Please complete our Author Checklist, and make sure that information entered into the checklist is also reflected in the manuscript; the checklist will be available to readers as part of the Review Process File.

8) Please note that supplementary information at EMBO Press has been superseded by the 'Expanded View' for inclusion of additional figures, tables, movies or datasets; with up to five EV Figures being typeset and directly accessible in the HTML version of the article.

9) To facilitate reproducibility and cross-laboratory adoption of methodologies, please structure the Materials & Methods section as outlined in our guide to authors, including a completed Reagents and Tools Table.

10) Digital image enhancement is acceptable practice, as long as it accurately represents the original data and conforms to community standards. If a figure has been subjected to significant electronic manipulation, this must be clearly noted in the figure legend and/or the 'Materials and Methods' section. The editors reserve the right to request original versions of figures and the original images that were used to assemble the figure. Finally, we generally encourage uploading of numerical as well as gel/blot image source data.

In the interest of ensuring the conceptual advance provided by the work, we recommend submitting a revision within 3 months (23rd Jun 2026). Please discuss the revision progress ahead of this time with the editor if you require more time to complete the revisions. Use the link below to submit your revision:

Link Not Available

Referee #2:

My previous concerns have been satisfactorily addressed in this revision - I have no remaining concerns.

Referee #3:

I am satisfied with the revisions made by the authors. They have thoroughly addressed all the concerns I raised in my previous review, including the additional experiments, the clarified methodology, and the revised discussion. This manuscript is now significantly improved and is suitable for publication. I recommend acceptance without further review.